

# MME-Emotion: A Holistic Evaluation Benchmark for Emotional Intelligence in Multimodal Large Language Models

**Fan Zhang**[*,1,2]**, Zebang Cheng**[*,3]**, Chong Deng**[*,2]**, Haoxuan Li**[4]**, Zheng Lian**[5]**,**
**Qian Chen**[2]**, Huadai Liu**[2]**, Wen Wang**[2]**, Yi-Fan Zhang**[6]**, Renrui Zhang**[1]**, Ziyu Guo**[1]**,**
**Zhihong Zhu**[7]**, Hao Wu**[7]**, Haixin Wang**[8]**, Yefeng Zheng**[9]**, Xiaojiang Peng**[†,3]**,**
**Xian Wu**[7]**, Kun Wang**[†,10]**, Xiangang Li**[2]**, Jieping Ye**[2]**, Pheng-Ann Heng**[†,1]

[1]The Chinese University of Hong Kong [2]Tongyi Lab [3]SZTU [4]Peking University
[5]Tongji University [6]CASIA [7]Tencent [8]UCLA [9]Westlake University [10]NTU
fzhang@link.cuhk.edu.hk, pheng@cse.cuhk.edu.hk

## ABSTRACT

Recent advances in multimodal large language models (MLLMs) have catalyzed transformative progress in affective computing, enabling models to exhibit emergent emotional intelligence. Despite substantial methodological progress, current emotional benchmarks remain limited, as it is still unknown: (a) the generalization abilities of MLLMs across distinct scenarios, and (b) their reasoning capabilities to identify the triggering factors behind emotional states. To bridge these gaps, we present **MME-Emotion**, a systematic benchmark that assesses both emotional understanding and reasoning capabilities of MLLMs, enjoying *scalable capacity*, *diverse settings*, and *unified protocols*. As the largest emotional intelligence benchmark for MLLMs, MME-Emotion contains over 6,000 curated video clips with task-specific questioning-answering (QA) pairs, spanning broad scenarios to formulate eight emotional tasks. It further incorporates a holistic evaluation suite with hybrid metrics for emotion recognition and reasoning, analyzed through a multi-agent system framework. Through a rigorous evaluation of 20 advanced MLLMs, we uncover both their strengths and limitations, yielding several key insights: ❶ Current MLLMs exhibit unsatisfactory emotional intelligence, with the best-performing model achieving only 39.3% recognition score and 56.0% Chain-of-Thought (CoT) score on our benchmark. ❷ Generalist models (*e.g.*, Gemini-2.5-Pro and GPT-4o) derive emotional intelligence from generalized multimodal understanding capabilities, while specialist models (*e.g.*, R1-Omni and AffectGPT) can achieve comparable performance through domain-specific post-training adaptation. By introducing MME-Emotion, we hope that it can serve as a foundation for advancing the emotional intelligence of MLLMs in the future. Project Page: https://mme-emotion.github.io/.

## 1 INTRODUCTION

Affective computing (Picard, 2000; Tao & Tan, 2005; Zhang et al., 2024c; 2025a; Mao et al., 2025) is a fascinating interdisciplinary field seeking to bridge the gap between human emotional intelligence and machine capabilities, with significant applications in education (Wu et al., 2016; Yadegaridehkordi et al., 2019), healthcare (Yannakakis, 2018; Liu et al., 2024b; Zhang et al., 2025b; Zhu et al., 2025), and human-robot interaction (Filippini et al., 2020; Gervasi et al., 2023). Catalyzed by emergent multimodal large language models (MLLMs) (Wu et al., 2023; Zhang et al., 2024b), research focus has increasingly shifted from exploring emotion recognition capability (Kosti et al., 2017) to investigating the interpretability of clues behind emotional states (Lian et al., 2024c).

---

[*]Equal contribution.
[†]Corresponding authors.

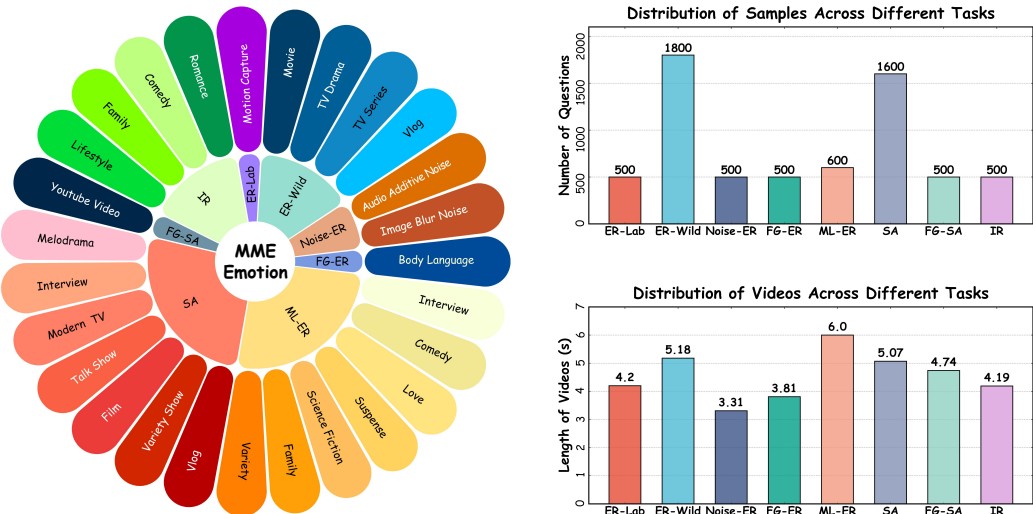

Figure 1: **Overview of MME-Emotion Statistics.** *Left*: Task Types. MME-Emotion encompasses eight emotional tasks across 27 distinct scenario types, enabling fine-grained analysis of diverse video contexts. *Right*: Data Distributions. MME-Emotion features balanced distributions of question volume and video duration, facilitating comprehensive evaluation of temporal understanding.

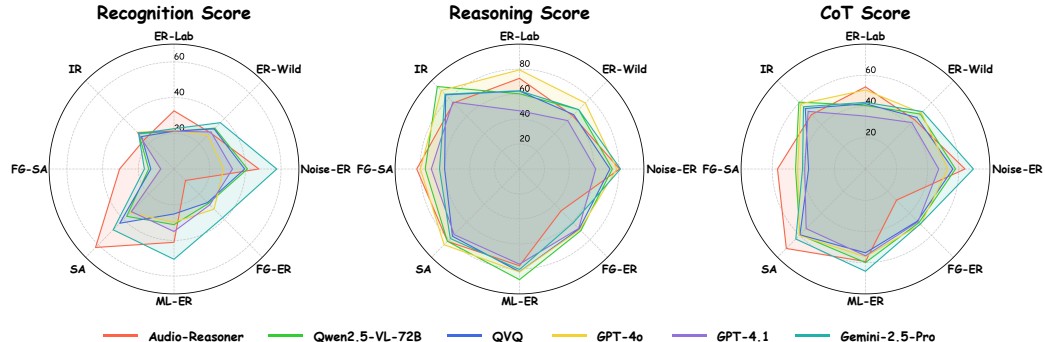

Figure 2: **Performance Comparison of Leading MLLMs on MME-Emotion.** Our evaluation suite assesses MLLMs using three unified metrics (Rec-S, Rea-S, and CoT-S) across eight emotional tasks.

While a growing body of methodological research (Cheng et al., 2024a; Zhao et al., 2025b; Lian et al., 2024c) explores the use of multimodal (audio, visual, and textual) data to boost emotional intelligence in MLLMs, existing evaluation benchmarks (Liu et al., 2024a; Hu et al., 2025) suffer from notable limitations, including (a) *inadequate scenario coverage* and (b) *inconsistent evaluation protocols*. Consequently, the extent to which current advanced MLLMs demonstrate emotional intelligence remains unclear under open and fair settings, as we are still unknown: ❶ how effectively they generalize across different real-world scenarios, ❷ how accurately they can recognize emotional states, and ❸ how well they are capable of identifying the triggering factors behind those emotions.

Towards this end, we present MME-Emotion, the first-ever comprehensive emotional intelligence benchmark for MLLMs, featuring *scalable capacity*, *diverse settings*, and *unified protocols*. As shown in Figure 1, MME-Emotion consists of 6,500 video clips associated with task-specific question-answering (QA) pairs across 27 distinct scenario types to formulate eight emotional tasks, including emotion recognition in the lab (ER-Lab), emotion recognition in the wild (ER-Wild), emotion recognition under noise (Noise-ER), fine-grained emotion recognition (FG-ER), multi-label emotion recognition (ML-ER), sentiment analysis (SA), fine-grained sentiment analysis (FG-SA), and intent recognition (IR). The distributions of question volume and video duration are balanced across all tasks, with each task containing a minimum of 500 QA pairs and video clips averaging >3.3 seconds.

Going beyond this, we provide a holistic evaluation suite for assessing the capabilities of MLLMs in emotion recognition and reasoning using unified protocols across all sub-tasks within MME-Emotion. For each question, we employ a multi-agent system framework to enable automated evaluation of MLLMs' responses with an MLLM-as-judge strategy. The visual clues, extracted audio clues, ground-truth emotion labels, and partitioned answer steps of a specific MLLM are fed into a GPT-based judge agent to evaluate the performance using three metrics: recognition score, reasoning score, and Chain-of-Thought (CoT) score. To further validate our evaluation approach, we also ask five human experts to cross-evaluate the performance of MLLMs on sampled data and manually annotated scores at each answer step. The comparison between GPT and expert scores demonstrates high consistency across multiple statistical metrics, confirming the effectiveness of our automated evaluation strategy.

Applying our evaluation suite to 20 state-of-the-art MLLMs, we uncover both their strengths and limitations, yielding the following key insights: ❶ The overall emotional intelligence of current MLLMs remains far from satisfactory (Figure 2). Even the top-performing model (Gemini-2.5-Pro) achieves merely 39.3% recognition score and 56.0% CoT score on our benchmark, respectively. The average performance across all evaluated MLLMs (29.4% recognition score, 49.5% reasoning score, and 39.5% CoT score) indicates there is still substantial room for improvement. ❷ While generalist models (*e.g.*, Gemini-2.5-Pro (Google, 2025c) and GPT-4o (OpenAI, 2024)) derive emotional intelligence from generalized multimodal understanding capabilities, specialist models (*e.g.*, R1-Omni (Zhao et al., 2025a) and Audio-Reasoner (Xie et al., 2025)) can achieve comparable performance through emotion-specific post-training adaptation techniques, such as supervised fine-tuning (SFT) (Zhang et al., 2023b; Liu et al., 2023b) and human preference alignment (Guo et al., 2025; Liu et al., 2025). ❸ Generally, response step count positively correlates with model performance, underscoring the necessity for equipping MLLMs with emotion reasoning capabilities in future development.

The main contributions of this paper can be summarized as follows:

- *Comprehensive Benchmark*: We introduce MME-Emotion, the largest benchmark for emotional intelligence in MLLMs that encompasses eight emotional tasks and 27 distinct scenarios, enabling a comprehensive evaluation of MLLMs' generalization capabilities across diverse settings.

- *Holistic Evaluation Suite*: We provide a multi-agent system framework that can automatically assess the emotion recognition and reasoning abilities of MLLMs using three unified metrics across all tasks. The validity of our evaluation strategy is also fully verified by five human experts.

- *Empirical Analysis*: Through a rigorous evaluation of 20 advanced open-source and closed-source MLLMs on MME-Emotion, along with in-depth analysis, we reveal their strengths and limitations, paving the way for future research efforts to advance emotional intelligence in MLLMs.

## 2 RELATED WORK

### 2.1 EMOTIONAL INTELLIGENCE IN MULTIMODAL LARGE LANGUAGE MODELS

The success of MLLMs (Liu et al., 2023a; Zhu et al., 2023; Luo et al., 2025b; Wang et al., 2025) has advanced affective computing, prompting growing interest in equipping these models with emotional intelligence (Cheng et al., 2024a; Zhao et al., 2025b; Lian et al., 2025a; Zhu et al.; Lian et al., 2025b). On the one hand, general MLLMs (such as the Gemini (Team et al., 2023), GPT (Achiam et al., 2023), and Qwen-VL (Bai et al., 2023) series) have demonstrated strong multimodal understanding capabilities. Thanks to their powerful LLM backbones and extensive pretraining on diverse datasets, emotional intelligence emerges as a byproduct of these general models. On the other hand, MLLMs with relatively small parameter sizes (ranging from 0.5B to 7B) have not yet shown strong general multimodal understanding capabilities, primarily due to limitations in data scale and model capacity. However, with post-training adaptation strategies, it is still possible to elicit strong performance on emotion-related tasks. For example, Emotion-LLaMA (Cheng et al., 2024a) employs emotion-specific encoders to project video, audio, and text modalities into a unified LLM space, followed by instruction tuning to activate emotional intelligence. R1-Omni (Zhao et al., 2025a) enhances emotional reasoning capabilities in MLLMs through reinforcement learning with verified feedback (RLVR). AffectGPT (Lian et al., 2024c), by training on a large-scale emotional dataset and introducing a pre-fusion projector to integrate multimodal emotional signals, builds a powerful emotion-specialized model. However, these models are often evaluated only on specific tasks and limited scenarios, leaving it unclear how they perform under open and fair evaluation settings.

Table 1: **Comparison of MME-Emotion with other Benchmarks related to Emotional Intelligence.** Rec-A and Rea-Q are short for recognition accuracy and reasoning quality, respectively.

| Benchmark | Task | Modality | Rec-A | Rea-Q | QA | LLM Eval | Human Assist |
|---|---|---|---|---|---|---|---|
| EmotionBench (Huang et al., 2023) | 1 | Text | ✓ | ✗ | ✓ | ✗ | ✓ |
| EmoBench (Sabour et al., 2024) | 1 | Text | ✓ | ✗ | ✓ | ✗ | ✗ |
| MOSABench (Song et al., 2024) | 1 | Image | ✓ | ✗ | ✓ | ✓ | ✗ |
| MM-InstructEval (Yang et al., 2025) | 6 | Image | ✓ | ✗ | ✓ | ✗ | ✗ |
| IEMOCAP (Busso et al., 2008) | 1 | Video | ✓ | ✗ | ✗ | ✗ | ✗ |
| MC-EIU (Liu et al., 2024a) | 2 | Video | ✓ | ✗ | ✗ | ✗ | ✗ |
| OV-MER (Lian et al., 2024a) | 1 | Video | ✓ | ✗ | ✓ | ✗ | ✓ |
| EmoBench-M (Hu et al., 2025) | 3 | Video | ✓ | ✗ | ✓ | ✓ | ✗ |
| MER-UniBench (Lian et al., 2024c) | 3 | Video | ✓ | ✗ | ✓ | ✗ | ✗ |
| MME-Emotion (Ours) | 8 | Video | ✓ | ✓ | ✓ | ✓ | ✓ |

## 2.2 DATASETS AND BENCHMARKS FOR EMOTIONAL INTELLIGENCE

Numerous efforts (Busso et al., 2008; Li et al., 2017) have been made to establish an open and fair environment for evaluating emotional intelligence in artificial intelligence (AI) models. However, most existing datasets and benchmarks were developed before the emergence of LLMs and MLLMs, primarily aimed at assessing the performance of traditional machine learning and deep learning models. Although some recent studies (Lian et al., 2024c; Hu et al., 2025) have attempted to adapt these datasets for MLLMs using prompt templates, a common limitation still remains: they focus solely on evaluating emotion recognition capabilities, while overlooking the models' abilities to reason about emotional cues. To address this gap, we propose an automated evaluation strategy, along with a newly introduced metric termed reasoning score, to quantify the reasoning abilities of MLLMs.

## 3 MME-EMOTION

In this section, we first introduce the data curation process, followed by an elaboration on our evaluation suite, including a multi-agent system framework and three unified metrics. As shown in Table 1, compared with existing emotion-related benchmarks, MME-Emotion stands out as the only one that simultaneously accounts for both recognition accuracy and reasoning quality.

### 3.1 BENCHMARK CONSTRUCTION

To curate data and construct our benchmark while reducing annotation costs, we collect videos and their corresponding emotion labels from publicly available resources. For longer videos that exhibit emotional shifts over time, we segment them into shorter video clips based on timestamps and the consistent emotion labels within specific intervals. We then convert various tasks into a QA format using prompt templates. Given that current MLLMs still lack sufficient emotional intelligence to handle open-ended tasks, we include all candidate emotion labels as a predefined label set within the prompt, enabling the model to make predictions in a closed-set setting. By aggregating and resampling data from multiple public datasets (Busso et al., 2008; Poria et al., 2018; Jiang et al., 2020; Liu et al., 2022b; Lee et al., 2019; Liu et al., 2024a; Feng et al., 2024; Luo et al., 2020; Lian et al., 2024b; Zadeh et al., 2018; 2016; Liu et al., 2022a), we compile a total of 6,500 QA pairs along with their corresponding video clips. To prevent potential data leakage, all involved samples are exclusively drawn from the test sets. These samples span eight emotion-related tasks across 27 distinct scenarios, including emotion recognition in the lab (ER-Lab), emotion recognition in the wild (ER-Wild), emotion recognition under noise (Noise-ER), fine-grained emotion recognition (FG-ER), multi-label emotion recognition (ML-ER), sentiment analysis (SA), fine-grained sentiment analysis (FG-SA), and intent recognition (IR). By constructing the MME-Emotion benchmark, we provide a comprehensive and systematic environment for assessing emotional intelligence in MLLMs.

### 3.2 EVALUATION SUITE

**Evaluation Strategy.** Evaluating the reasoning capabilities of MLLMs' emotional intelligence is a challenging task. To address this, we propose an annotation-free strategy based on a multi-agent

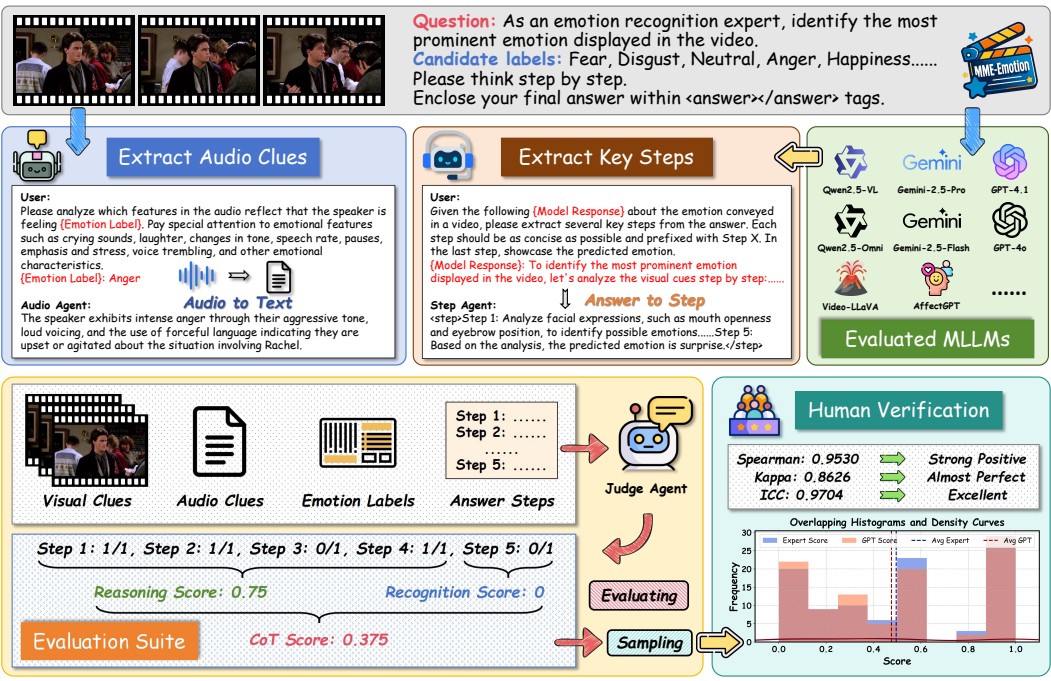

Figure 3: **Illustration of Our Evaluation Strategy.** We leverage a multi-agent system framework to assess the recognition and reasoning capabilities of MLLMs across different tasks with three unified metrics. To validate the effectiveness of our MLLM-as-judge strategy, we further compare the results of the judge agent on sampled data against results cross-evaluated by five human experts.

system (Figure 3). This approach allows us to assess the reasoning performance of MLLMs without the need for manually annotated ground-truth reasoning steps. Specifically, we first obtain the MLLM's response to a given question, and then employ a unimodal step agent to automatically extract key reasoning steps from the model's answer. This process can be formulated as:

$$A = \text{MLLM}(Q, V), \quad S = \text{Step-LLM}(P_s, A), \tag{1}$$

where $Q$, $V$, and $A$ are short for question, video, and answer, respectively. $P_s$ and $S$ denote the prompt for the step agent and the corresponding answer steps. Here, we opt for GPT-4.1 (OpenAI, 2025) as our step agent. Note that we refrain from incorporating additional elements—such as the original question or the video content—during the step partition process, as we aim to avoid introducing external biases that could influence the identification of reasoning steps. Afterward, we employ another powerful multimodal agent as the judge to rate the performance of the extracted answer steps. To ensure an accurate and comprehensive assessment, it is essential to provide the judge agent with complete multimodal information, thereby minimizing the risk of misjudgment due to missing context. However, mainstream multimodal agents such as GPT-4o (OpenAI, 2024) are currently unable to process all modalities simultaneously. To address this limitation, we propose a divide-and-conquer perspective. Specifically, for visual clues, we can directly convert the original video into frames and feed them into the judge agent. For audio clues, we utilize a separate, powerful audio-language model as an audio agent to extract relevant audio information. Once we obtain the visual and audio clues, we combine them with the answer steps and the ground-truth emotion labels and input them into the judge agent for final evaluation. This process can be formulated as:

$$C_v = \text{Convert}(V), \quad C_a = \text{Audio-LLM}(P_a, V), \tag{2}$$

$$\text{Rec-S}, \text{Rea-S} = \text{Judge-MLLM}(P_j, C_v, C_a, Y, S), \tag{3}$$

where $C_v$, $C_a$, and $Y$ denote visual clues, audio clues, and emotion labels, respectively. $P_a$ and $P_j$ are the prompts for extracting audio clues and evaluating the performance. Here, we employ Qwen2-Audio (Chu et al., 2024) as the audio agent and GPT-4o (OpenAI, 2024) as the judge agent. We adopt three unified evaluation metrics across all tasks: recognition score (Rec-S), reasoning score (Rea-S), and Chain-of-Thought score (CoT-S), each of which will be detailed in the following section.

Table 2: **Overall Performance Comparison** (%) **on MME-Emotion.** The top three performing results are highlighted in red (1st), blue (2nd), and yellow (3rd) backgrounds, respectively.

| Model | LLM Size | A | V | T | Avg Step | Avg Token | Rec-S | Rea-S | CoT-S |
|---|---|---|---|---|---|---|---|---|---|
| *Open-source MLLMs* | | | | | | | | | |
| Qwen2-Audio (Chu et al., 2024) | 7B | ✓ | | ✓ | 3.0 | 40.3 | 34.1 | 50.4 | 42.3 |
| Audio-Reasoner (Xie et al., 2025) | 7B | ✓ | | ✓ | 5.0 | 356.8 | 38.1 | 71.6 | 54.8 |
| Qwen2-VL-7B (Wang et al., 2024) | 7B | | ✓ | ✓ | 2.9 | 68.0 | 29.2 | 38.1 | 33.7 |
| Qwen2-VL-72B (Wang et al., 2024) | 72B | | ✓ | ✓ | 1.5 | 24.8 | 31.1 | 10.5 | 20.8 |
| Qwen2.5-VL-7B (Bai et al., 2025) | 7B | | ✓ | ✓ | 4.8 | 169.7 | 28.4 | 64.8 | 46.6 |
| Qwen2.5-VL-72B (Bai et al., 2025) | 72B | | ✓ | ✓ | 4.8 | 266.3 | 31.3 | 75.7 | 53.5 |
| QVQ (Team, 2024) | 72B | | ✓ | ✓ | 5.5 | 899.9 | 31.4 | 70.1 | 50.8 |
| Video-LLaVA (Lin et al., 2023) | 7B | | ✓ | ✓ | 2.3 | 19.1 | 25.8 | 32.8 | 29.3 |
| Video-LLaMA (Zhang et al., 2023a) | 7B | ✓ | ✓ | ✓ | 4.5 | 122.5 | 26.1 | 48.5 | 37.3 |
| Video-LLaMA2 (Cheng et al., 2024b) | 7B | ✓ | ✓ | ✓ | 2.6 | 37.7 | 29.2 | 27.7 | 28.4 |
| Qwen2.5-Omni (Xu et al., 2025) | 7B | ✓ | ✓ | ✓ | 3.7 | 78.6 | 17.4 | 59.3 | 38.4 |
| Emotion-LLaMA (Cheng et al., 2024a) | 7B | ✓ | ✓ | ✓ | 1.0 | 2.3 | 25.1 | 0.4 | 12.8 |
| HumanOmni (Zhao et al., 2025b) | 7B | ✓ | ✓ | ✓ | 1.0 | 1.3 | 36.0 | 0.3 | 18.1 |
| R1-Omni (Zhao et al., 2025a) | 0.5B | ✓ | ✓ | ✓ | 5.0 | 156.2 | 26.3 | 58.6 | 42.4 |
| AffectGPT (Lian et al., 2024c) | 7B | ✓ | ✓ | ✓ | 4.9 | 122.8 | 11.9 | 50.6 | 31.2 |
| *Closed-source MLLMs* | | | | | | | | | |
| GPT-4o (OpenAI, 2024) | — | | ✓ | ✓ | 4.4 | 169.4 | 27.8 | 79.8 | 53.8 |
| GPT-4.1 (OpenAI, 2025) | — | | ✓ | ✓ | 5.2 | 141.2 | 28.8 | 65.2 | 47.0 |
| Gemini-2.0-Flash (Google, 2025a) | — | | ✓ | ✓ | 4.1 | 64.7 | 36.3 | 60.0 | 48.1 |
| Gemini-2.5-Flash (Google, 2025b) | — | | ✓ | ✓ | 4.3 | 261.8 | 34.7 | 52.7 | 43.7 |
| Gemini-2.5-Pro (Google, 2025c) | — | | ✓ | ✓ | 5.1 | 538.6 | 39.3 | 72.7 | 56.0 |

**Evaluation Metric.** Next, we provide a detailed introduction to the evaluation metrics employed in this paper. During the earlier step extraction process from MLLM answers, we consistently place the task-specific prediction in the final step, while treating all other steps as the reasoning process. To compute the recognition score, we compare the final prediction step against the ground-truth emotion labels. For tasks involving a single emotion label, we adopt standard accuracy as the recognition score, following conventional evaluation practices (Li & Deng, 2020; Tzirakis et al., 2017). For tasks involving multiple emotion labels, the recognition score is defined as the ratio between the number of correctly predicted emotions and the total number of ground-truth emotions. We then derive the reasoning score from the extracted reasoning steps. Each step is treated as a binary classification problem, where the judge agent determines whether the reasoning is correct or not. To eliminate potential bias caused by varying step lengths, we compute the reasoning score as the average correctness across all reasoning steps. Finally, we obtain the CoT score by taking a weighted combination of the recognition score and the reasoning score, providing a holistic measure of both recognition accuracy and reasoning quality. The CoT score is defined as:

$$\text{CoT-S} = \alpha \times \text{Rec-S} + (1 - \alpha) \times \text{Rea-S}, \tag{4}$$

where $\alpha \in [0, 1]$ is a hyperparameter that controls the relative contribution of the recognition score and reasoning score in the overall evaluation process. By default, we set $\alpha = 0.5$.

**Human Verification.** To evaluate the reliability of our MLLM-as-judge evaluation strategy, we recruited five experts to perform manual validation. Specifically, the human experts annotated 373 reasoning steps drawn from 100 randomly sampled questions and their corresponding answers from evaluated MLLMs. Then we assessed the consistency between the GPT scores and the expert scores. As shown in the bottom-right panel of Figure 3, the Spearman's rank correlation coefficient (Spearman, 1961) between the two sets of scores is 0.9530, indicating a strong positive correlation. In terms of inter-rater reliability, the two sets of scores achieve a Cohen's Kappa coefficient (McHugh, 2012) of 0.8626, which falls into the category of almost perfect agreement. Additionally, the intra-class correlation coefficient (ICC) (Koch, 2004) of 0.9704 further confirms excellent reliability between the two scoring methods. These results collectively demonstrate that our MLLM-as-judge strategy aligns closely with human judgment and serves as a highly reliable evaluation approach.

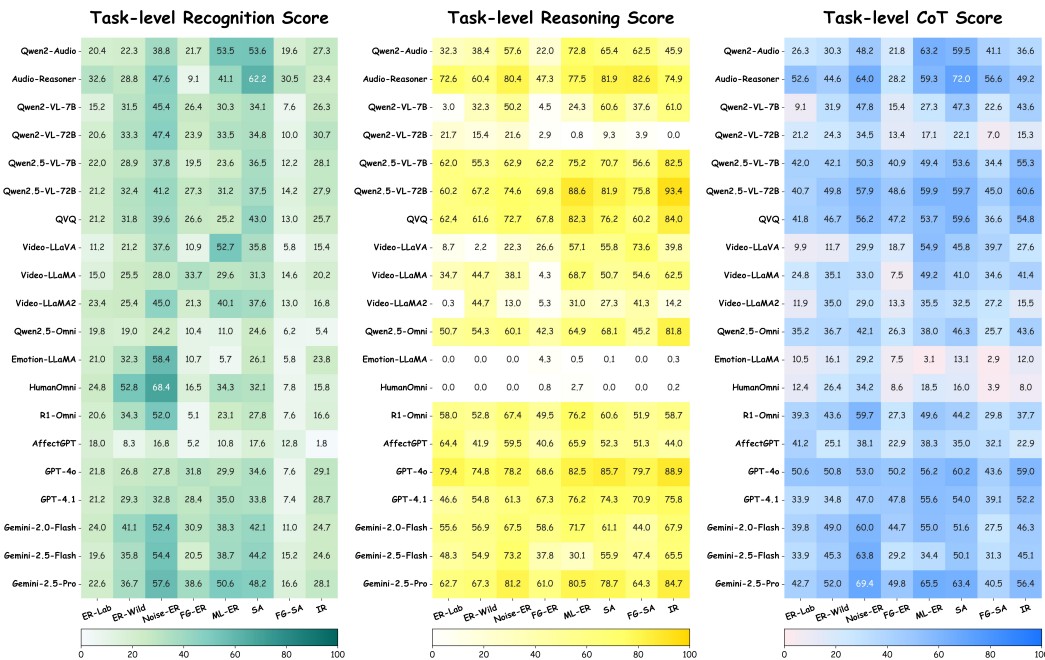

Figure 4: **Task-level Performance Comparison** (%) **on MME-Emotion.** We showcase fine-grained comparison results of 20 MLLMs using Rec-S, Rea-S, and CoT-S across 8 emotional tasks.

## 4  EXPERIMENTS

### 4.1  EXPERIMENTAL SETUP

We evaluate the performance on a total of 20 cutting-edge MLLMs, including Qwen2-Audio (Chu et al., 2024), Audio-Reasoner (Xie et al., 2025), Qwen2-VL-7B/72B (Wang et al., 2024), Qwen2.5-VL-7B/72B (Bai et al., 2025), QVQ (Team, 2024), Video-LLaVA (Lin et al., 2023), Video-LLaMA (Zhang et al., 2023a), Video-LLaMA2 (Cheng et al., 2024b), Qwen2.5-Omni (Xu et al., 2025), Emotion-LLaMA (Cheng et al., 2024a), HumanOmni (Zhao et al., 2025b), R1-Omni (Zhao et al., 2025a), AffectGPT (Lian et al., 2024c), GPT-4o (OpenAI, 2024), GPT-4.1 (OpenAI, 2025), Gemini-2.0-Flash (Google, 2025a), Gemini-2.5-Flash (Google, 2025b), and Gemini-2.5-Pro (Google, 2025c). For most of the open-sourced MLLMs, we use the vLLM (Kwon et al., 2023) framework to perform inference on our benchmark. For models that are not compatible with the vLLM framework, we conduct experiments using the official open-source code provided by the authors. For closed-sourced MLLMs, inference is conducted directly via their official APIs. All the experiments are conducted under zero-shot settings. To ensure a fair comparison across models, we maintain consistent prompting wherever possible, with only minor modifications when necessary. For example, for audio-language models, the word "video" in prompts is replaced with "audio"; for vision-language models that do not support video input but accept multiple images, "video" is replaced with "video frames".

### 4.2  MAIN RESULTS

We showcase the overall performance comparison on MME-Emotion, covering three performance metrics: recognition score (Rec-S), reasoning score (Rea-S), and Chain-of-Thought score (CoT-S) as well as two response length metrics: average step count (Avg Step) and average token count (Avg Token), as shown in Table 2. Notably, even the top-performing models, Gemini-2.5-Pro (Google, 2025c), Audio-Reasoner (Xie et al., 2025), and GPT-4o (OpenAI, 2024), achieve CoT scores of only 56.0%, 54.8%, and 53.8%, respectively. All evaluated MLLMs report recognition scores below 40%, and most closed-source MLLMs also score below 40% in terms of CoT quality. This performance landscape highlights the challenging nature of the MME-Emotion benchmark. It also suggests that even state-of-the-art MLLMs are still in the early stages of developing emotional intelligence.

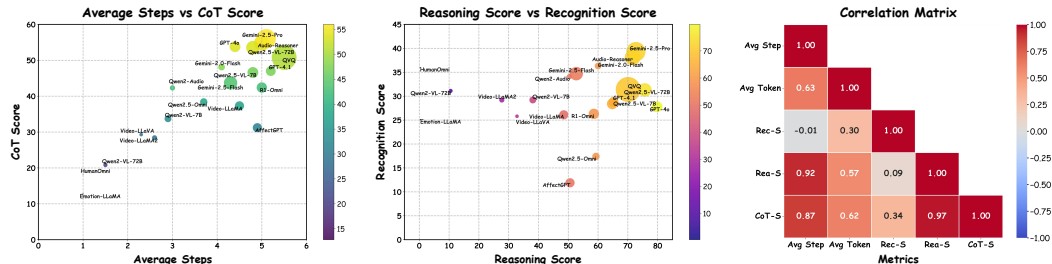

Figure 5: **Relationships among Model Evaluation Metrics.** *Left Panel*: Relationship between average steps and CoT scores. *Center Panel*: Relationship between reasoning and recognition scores. *Right Panel*: Pearson correlation quantifying inter-dependencies among five evaluation metrics.

We also observe that current MLLMs have yet to fully and effectively leverage multimodal information. As shown in Table 2, Audio-Reasoner, which processes only audio and textual modalities, achieves a strong CoT score of 54.8%. Similarly, many closed-source vision-language models, utilizing only visual and textual information, also demonstrate competitive performance. In contrast, omnimodal models, which are designed to process audio, visual, and textual information simultaneously, often exhibit noticeable performance drops. This phenomenon points to two potential issues: ❶ Multimodal data often contains redundant or inconsistent emotional clues across different modalities. ❷ Existing omnimodal models still lack effective strategies for robust multimodal emotional clues fusion.

In addition, Figure 4 presents a fine-grained comparison of 20 MLLMs across eight distinct emotional tasks, evaluated using three performance metrics. The results reveal notable differences in task difficulty. For more challenging tasks such as FG-SA and IR, even the top-performing MLLMs achieve recognition scores of only around 30%. When comparing the results of two related tasks, ER-Wild and ER-Lab, we observe a clear performance gap across different MLLMs. This discrepancy suggests that most existing MLLMs are primarily trained on in-the-wild data and therefore struggle to generalize effectively to controlled, in-the-lab settings, leading to noticeable performance degradation. These findings further underscore the discriminative capability of the MME-Emotion benchmark.

## 4.3 OBSERVATIONS AND INSIGHTS

Delving into the experimental results, we can derive the following key observations and insights.

**Obs. 1: Emotional intelligence in MLLMs can be bootstrapped through diverse approaches.**
On the one hand, generalist MLLMs, such as the Gemini and GPT series, are equipped with large model capacities and trained on massive datasets. As a result, they exhibit strong general understanding capabilities and perform well across various tasks (Li et al., 2024; Fu et al., 2023; Zhang et al., 2024d; Fu et al., 2025; Jiang et al., 2025). Although these models have not been explicitly adapted to emotion-specific domains (Fu et al., 2024), our experimental results show that they can generalize from general intelligence to emotional intelligence, illustrating one viable path forward. On the other hand, specialist models, such as AffectGPT and R1-Omni, may not match generalist models in broad understanding domains due to the limitation of scaling laws (Kaplan et al., 2020; Zhang et al., 2024a). However, by constructing large-scale emotional data and applying post-training adaptation, they can achieve comparable performance against generalist models. These findings demonstrate that there is no one-size-fits-all solution: emotional intelligence can be bootstrapped through a variety of strategies, opening the door to a broad range of future research directions.

**Obs. 2: Encouraging deeper reasoning can enhance emotional intelligence in MLLMs.** In Figure 5, there is a general positive correlation between the average number of answer steps and overall performance. While certain non-reasoning MLLMs, such as Emotion-LLaMA, HumanOmni, and Qwen2-VL-72B, can generate direct answers without explicit reasoning through instruction tuning, most reasoning-capable MLLMs demonstrate a strong synergy between reasoning and recognition: improvements in one metric are often accompanied by gains in the other. These findings suggest a promising research direction for advancing the emotional intelligence of MLLMs: encouraging deeper reasoning by incentivizing models to reflect and identify the underlying triggers of emotions.

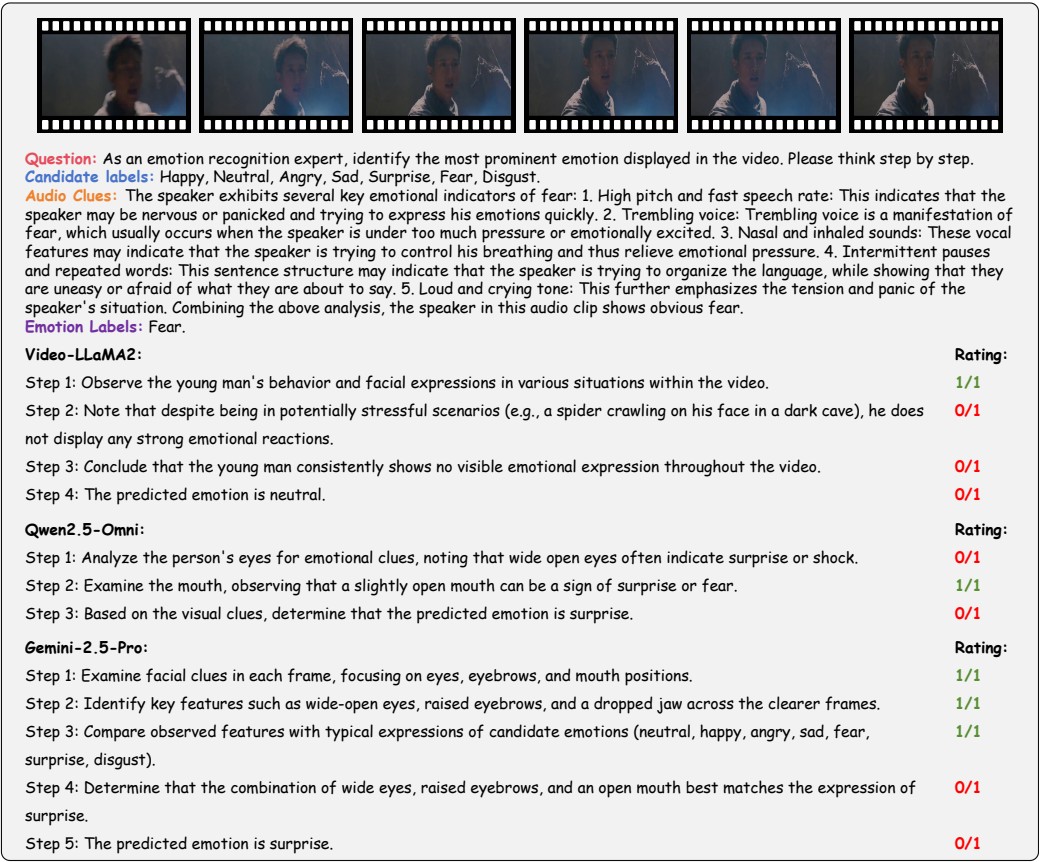

Figure 6: **Cases of Error Responses.** We showcase some typical error types of cutting-edge MLLMs.

**Obs. 3: Limited visual perception capabilities constrain the emotional intelligence in MLLMs.**
As illustrated in Figure 6, the failure cases of Video-LLaMA2 and Qwen2.5-Omni reveal that both models struggle primarily due to insufficient visual perception capabilities. In this example, Video-LLaMA2 fails to accurately capture subtle facial expression changes required and thus incorrectly classifies the emotion as neutral. Qwen2.5-Omni demonstrates notable progress by narrowing down the candidate emotions to surprise and fear, eliminating many irrelevant options. However, when it comes to making a fine-grained distinction between these two emotions, it still fails to recognize the subject's fearful expression, leading to an incorrect prediction. These findings highlight the importance of improving the fine-grained visual perception capabilities of MLLMs, with particular emphasis on the accurate interpretation of facial expressions and body movements. Such advancements are essential for the continued development of emotional intelligence in MLLMs.

**Obs. 4: Incomplete multimodal information limits the emotional intelligence in MLLMs.** The failure case of Gemini-2.5-Pro in Figure 6 reveals that the error primarily stems from the absence of audio information. In this example, the character's fearful emotion is clearly conveyed through audio clues. However, the model fails to recognize it due to its inability to jointly process both audio and visual modalities from the video clip. Most current generalist MLLMs are still limited in their capacity to integrate audio and visual information simultaneously, leading to misinterpretations when critical emotional signals are missing. These findings suggest that although current leading models are primarily vision-language models, the future of emotional intelligence lies in advancing more powerful omnimodal models capable of jointly processing audio, visual, and textual information.

## 5 CONCLUSION

In this paper, we introduced MME-Emotion, a comprehensive multi-task benchmark for evaluating emotional intelligence in MLLMs, accompanied by a holistic evaluation suite. The assessment

process was fully automated within a multi-agent system framework and thoroughly validated by human experts. Through rigorous evaluation using three unified metrics and in-depth empirical analysis, we uncovered both the strengths and limitations of cutting-edge MLLMs, laying a solid foundation for future research aimed at advancing emotional intelligence in MLLMs. In future work, we intend to build upon the observations and insights derived from this paper and systematically address the existing shortcomings of MLLMs, developing a powerful emotion-specialist model.

## ACKNOWLEDGMENT

The work described in this paper was supported in part by the Research Grants Council of the Hong Kong Special Administrative Region, China, under Project T45-401/22-N and Project CUHK 14202125.

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

CONTENTS

## A  THE USE OF LARGE LANGUAGE MODELS (LLMS)

LLMs were not involved in the research ideation or the writing of this paper.

## B  LIMITATIONS AND SOCIAL IMPACT

While MME-Emotion represents a significant step forward in the field of affective computing and multimodal large language models (MLLMs), several limitations remain worth acknowledging:

- First, although we provide a multi-task and multi-scenario benchmark for evaluating emotional intelligence in MLLMs, it would be beneficial to incorporate task difficulty classification. Similar to the categorization of mathematical problems (Hendrycks et al., 2021) into primary school, high school, college, and Olympiad levels, assigning a difficulty rating or level to each sample in our benchmark could offer a more nuanced understanding of MLLMs' capabilities in handling affective computing tasks across varying levels of complexity.

- Second, since our benchmark is constructed from publicly available video datasets, the included data covers a wide range of multilingual scenarios. However, we do not currently distinguish between different languages in the benchmark, nor do we analyze model performance across different linguistic contexts. In future work, we aim to extend our evaluation from a multilingual perspective, enabling a more comprehensive assessment of emotional intelligence in MLLMs.

Overall, we present a systematic benchmark for evaluating both the recognition and reasoning capabilities of MLLMs in emotion-related tasks, which we believe will serve as a valuable resource for both the MLLM and affective computing communities. We hope this work will inspire growing research efforts aimed at bootstrapping emotional intelligence in MLLMs, ultimately advancing downstream applications in education, healthcare, and human–robot interaction.

## C  MORE RELATED WORKS

### C.1  DATASETS AND BENCHMARKS FOR MULTIMODAL LARGE LANGUAGE MODELS

With the rise of large language models (LLMs) and the growing popularity of multimodal large language models (MLLMs), an increasing number of studies have focused on evaluating MLLM performance across a wide range of domains. By collecting data and employing human annotations, MME (Fu et al., 2023) evaluates the general understanding capabilities of MLLMs across 14 subtasks, ranging from perception tasks to cognition tasks. Video-MME (Fu et al., 2025) assesses the video analysis capabilities of MLLMs using a diverse and high-quality collection of video data. MME-RealWorld (Zhang et al., 2024d) aggregates a large set of high-resolution images, combined with manual annotations, to construct a suite of practical and challenging tasks, including optical character recognition (OCR) in the wild, remote sensing, and autonomous driving. FinMME (Luo et al., 2025a) leverages annotations from 20 domain experts to comprehensively evaluate MLLMs' understanding capabilities in the financial domain. These benchmarks have significantly accelerated progress in MLLM development, enabling improved performance in both general-purpose and domain-specific scenarios, ultimately contributing to a wide array of real-world applications.

## D  IMPLEMENTATION DETAILS

### D.1  EVALUATED MLLMS

We evaluated a total of 20 MLLMs, comprising 15 closed-source and 5 open-source models. A brief introduction to each model is provided below:

- Qwen2-Audio (Chu et al., 2024) is the latest open-source large audio-language model developed by Alibaba Group. It is designed to handle a wide range of audio inputs and can perform detailed audio analysis or generate direct textual responses based on spoken

instructions. Qwen2-Audio achieves state-of-the-art performance across multiple audio-related tasks, including automatic speech recognition (ASR), speech-to-text translation (S2TT), speech emotion recognition (SER), and vocal sound classification (VSC).

- Audio-Reasoner (Xie et al., 2025) is a large audio-language model built upon Qwen2-Audio, enhanced with advanced reasoning capabilities. By curating structured chain-of-thought (CoT) data and applying supervised fine-tuning, it achieves superior performance and sets a new state-of-the-art in audio reasoning tasks.

- Qwen2-VL (Wang et al., 2024) is a newly developed series of large vision-language models from Alibaba Group. A key innovation in Qwen2-VL is the incorporation of a naive dynamic resolution mechanism, which allows the model to flexibly adapt to images of different resolutions by converting them into variable numbers of visual tokens. This design enhances both the efficiency and precision of visual representations, in a manner that closely mirrors human visual perception. Consequently, Qwen2-VL series achieve cutting-edge results in visual understanding tasks and exhibits strong capabilities in complex reasoning and decision-making.

- Qwen2.5-VL (Bai et al., 2025) series, developed by Alibaba Group, represents one of the most advanced large vision-language model families to date. Leveraging a streamlined and efficient vision encoder alongside dynamic resolution and frame rate training strategies, the Qwen2.5-VL models set new state-of-the-art performance on a wide range of visual tasks. In particular, they demonstrate exceptional capabilities in fine-grained video understanding.

- QVQ (Team, 2024) is a large vision-language model developed by Alibaba Group based on the Qwen2-VL architecture, with a particular focus on strengthening visual reasoning capabilities. It represents a significant advancement in AI's ability to perform visual understanding and solve complex problems. QVQ achieves remarkable performance, especially in reasoning-intensive tasks such as mathematical problem solving, marking a notable breakthrough in the field.

- Video-LLaVA (Lin et al., 2023) is a large vision-language model capable of processing both images and videos. Its core innovation lies in projecting image and video features into a unified representation space, enabling the language model to learn cross-modal interactions from a consistent visual embedding. Video-LLaVA demonstrates strong performance across both image understanding and video analysis tasks.

- Video-LLaMA (Zhang et al., 2023a) is a large omnimodal model designed to process both visual and audio information through dedicated visual and audio branches. Unlike many existing models that are limited to a single modality, Video-LLaMA enables simultaneous understanding of both visual and acoustic content within videos. By integrating multimodal data, it addresses critical limitations in current video understanding approaches.

- Video-LLaMA2 (Cheng et al., 2024b) builds upon its predecessor by integrating a customized spatio-temporal convolution connector (STC), which effectively captures the intricate spatial and temporal dynamics inherent in video data. Furthermore, the model incorporates an audio branch through joint training, allowing it to seamlessly fuse audio clues and significantly enhance its multimodal understanding capabilities.

- Qwen2.5-Omni (Xu et al., 2025) is a flagship large omnimodal model developed by Alibaba Group. It adopts the Thinker-Talker architecture, specifically designed for comprehensive multimodal perception. The model can seamlessly process a wide range of input modalities, including text, images, audio, and video, while also supporting streaming text generation and natural speech synthesis outputs. Qwen2.5-Omni achieves strong performance across both unimodal and multimodal understanding tasks.

- Emotion-LLaMA (Cheng et al., 2024a) is an omnimodal emotion-specialist model designed for multimodal emotion recognition. It leverages audio, visual, and textual encoders to capture comprehensive multimodal representations and employs a two-stage training strategy to enhance learning effectiveness. As a result, Emotion-LLaMA achieves strong performance across various emotion recognition tasks.

- HumanOmni (Zhao et al., 2025b) is an omnimodal specialist model tailored for human-centric tasks, including emotion recognition. It utilizes three distinct visual projectors—face-related, body-related, and interaction-related—combined with a learnable gating mechanism to capture fine-grained, task-relevant visual representations. This design enables

HumanOmni to achieve strong performance across a wide range of human-centric down-stream tasks.

- R1-Omni (Zhao et al., 2025a) is an omnimodal emotion-specialist model built upon the foundation of HumanOmni. By combining reinforcement learning with verifiable reward (RLVR) and rule-based rewards, R1-Omni effectively enhances the model's reasoning capabilities on emotional tasks.

- AffectGPT (Lian et al., 2024c) is an omnimodal emotion-specialist model. Leveraging a model-based crowdsourcing strategy, the authors curated a large-scale emotion under-standing dataset rich in emotional cues. Leveraging this dataset for training, alongside the incorporation of pre-fusion operations to enhance multimodal integration, AffectGPT demonstrates robust and advanced capabilities in emotion understanding.

- GPT-4o (OpenAI, 2024) is one of the latest MLLMs developed by OpenAI, providing convenient APIs for processing textual, visual, and audio modalities. It demonstrates state-of-the-art performance across a wide range of benchmarks, with significant advancements in visual perception, understanding, and reasoning. Equipped with a unified architecture that facilitates seamless cross-modal interaction, GPT-4o is both highly efficient and adaptable, making it well-suited for real-world multimodal applications.

- GPT-4.1 (OpenAI, 2025) is a powerful and cost-effective MLLM released by OpenAI. It demonstrates significant improvements in programming capabilities and instruction fol-lowing. Additionally, GPT-4.1 features an expanded context window supporting up to one million tokens, enabling enhanced long-context understanding that better leverages information to improve both efficiency and performance.

- Gemini-2.0-Flash (Google, 2025a) supports input and output across multiple modalities, including images, videos, and audio, while maintaining efficient inference speed. By leveraging advanced reasoning capabilities along with extended context understanding, Gemini-2.0-Flash achieves strong performance on a variety of multimodal reasoning tasks.

- Gemini-2.5-Flash (Google, 2025b) is a multimodal hybrid reasoning model that strikes a balance between performance and efficiency. By allowing users to selectively enable reasoning, it offers a flexible trade-off among performance, speed, and cost. Through fine-grained control over reasoning tokens, Gemini-2.5-Flash achieves outstanding performance across a variety of multimodal understanding tasks.

- Gemini-2.5-Pro (Google, 2025c), a recent release from Google DeepMind, advances mul-timodal large language modeling with enhanced capabilities in visual understanding. It supports longer context windows and improves the efficiency of cross-modal alignment. Equipped with robust reasoning skills, Gemini-2.5 Pro excels across a wide spectrum of tasks, such as programming, mathematics, and scientific problem solving.

## D.2 PROMPT FOR EVALUATED MLLMS

We adopt unified prompts across all evaluated MLLMs to ensure a fair comparison, with slight modifications made for audio-language models, vision-language models, and omnimodal models to accommodate their specific input modalities. Examples of the prompts are provided below.

---

**Prompt for ER-Lab, ER-Wild, and Noise-ER**

As an emotion recognition expert, identify the most prominent emotion displayed in the video.
Candidate labels: ......
Please think step by step. Enclose your final answer within <answer></answer> tags.

---

**Prompt for FG-ER**

As an emotion recognition expert, identify one or several emotions displayed in the video.
Candidate labels: ......
Please think step by step. Enclose your final answer within <answer></answer> tags.

---

> **Prompt for ML-ER**
>
> As an emotion recognition expert, identify multiple emotions displayed in the video.
> Candidate labels: ......
> Please think step by step. Enclose your final answer within <answer></answer> tags.

> **Prompt for SA**
>
> As an emotion recognition expert, identify the most prominent sentiment displayed in the video.
> Candidate labels: ......
> Please think step by step. Enclose your final answer within <answer></answer> tags.

> **Prompt for FG-SA**
>
> As an emotion recognition expert, identify the most prominent fine-grained sentiment displayed in the video.
> Candidate labels: ......
> Please think step by step. Enclose your final answer within <answer></answer> tags.

> **Prompt for IR**
>
> As an emotion recognition expert, identify the most prominent intent displayed in the video.
> Candidate labels: ......
> Please think step by step. Enclose your final answer within <answer></answer> tags.

## D.3 PROMPT FOR EXTRACTING AUDIO CLUES

An example prompt for extracting audio clues is shown below. Please note that minor adjustments are needed depending on the specific task requirements.

> **Prompt for Extracting Audio Clues**
>
> You are a speech emotion analysis expert, specializing in analyzing the tone and intonation of input audio. Please analyze which features in the audio reflect that the speaker is feeling {Emotion labels}. Pay special attention to emotional features such as crying sounds, laughter, changes in tone, speech rate, pauses, emphasis and stress, voice trembling, and other emotional characteristics.
> Emotion labels: ......

## D.4 PROMPT FOR EXTRACTING KEY STEPS

An example prompt for extracting key answer steps is shown below. Please note that minor adjustments are needed depending on the specific task requirements.

---

**Prompt for Extracting Kye Steps**

You are an expert in affective computing and very good at handling tasks related to emotion recognition. Given the following answer about the emotion conveyed in a video, please help me extract several key steps from the answer. Each step should be as concise as possible and prefixed with Step X. In the last step, showcase the predicted emotion. If there are no reasoning steps, please showcase the predicted emotion at Step 1. Enclose your result within <step></step> tags.
Answer: ......

Example 1:
Answer:
In the video, an elderly man wearing a green shirt is in an outdoor nighttime setting. He appears focused and serious, with slightly furrowed brows and a serious expression. His eyes are scanning the other person, suggesting he wants to convey something important or ask about a certain thing. As time passes, his gaze shifts from scanning to direct engagement, eventually relaxing and showing a hint of happiness, indicating fluctuations in his emotions. Overall, the elderly man experiences a moderate intensity of neutral emotion, mixed with slight positive fluctuations.
Result:
<step>Step 1: Observe the setting and characters in the video, noting any relevant details such as location and participants' appearance. Step 2: Analyze the facial expressions and movements, such as furrowed brows and downturned eyes, looking for indicators of specific emotions. Step 3: Consider verbal and non-verbal cues, including mouth movements and gaze direction, that signify communication and emotional responses. Step 4: Synthesize observations to understand the overall emotional state conveyed by the participant. Step 5: Based on the analysis, determine that the predicted emotion is neutral.</step>

Example 2:
Answer:
happy
Result:
<step>Step 1: The predicted emotion is happy.</step>

---

## D.5 PROMPT FOR RATING PERFORMANCE

An example prompt for evaluating the performance is shown below. Please note that minor adjustments are needed depending on the specific task requirements.

---

**Prompt for Evaluation the Performance**

You are an expert in affective computing and very good at handling tasks related to emotion recognition. I will first give you some ground truth information about the emotion in a video: visual clue, audio clue, and emotion label. I will also give you a model prediction. Please help me rate the performance of the prediction.

Rating Requirements:
1. Rate each step using 0 or 1. (0=wrong, 1=correct)
2. For the last step, your rating should be 1 if the predicted emotion matches the ground truth emotion label, otherwise 0. Don't consider visual/audio clues at this step.
3. For each other step, only consider predictions clearly contradicting visual/audio clues as incorrect.
4. Ensure the number of steps in your rating is equal to that in the model prediction.
5. Output format: <score>Step 1: 0/1, Step 2: 1/1,...</score>.

Input Data:
- Visual clue: The video frames (images)
- Audio clue: ......
- Emotion label: ......
- Model prediction: ......

Example Output:
<score>Step 1: 1/1, Step 2: 0/1, Step 3: 0/1</score>

---

# E ADDITIONAL EXPERIMENTAL RESULTS

## E.1 ADDITIONAL EXPERIMENTAL RESULTS ACROSS TASKS

As shown in Table 3, Table 4, Table 5, Table 6, Table 7, Table 8, Table 9, and Table 10, we present a fine-grained performance comparison of cutting-edge MLLMs across various emotional tasks.

## E.2 ADDITIONAL EXPERIMENTAL RESULTS ACROSS CATEGORIES

As shown in Figure 7, Figure 8, Figure 9, Figure 10, Figure 11, and Figure 12, we present a fine-grained performance comparison of cutting-edge MLLMs across different categories of emotions, sentiments, and intents. We include only single-label tasks and exclude the results of multi-label comparisons.

## E.3 ADDITIONAL EXPERIMENTAL RESULTS ACROSS METRICS

As shown in Figure 13, we provide additional results to showcase the relationships among model evaluation metrics. The results are consistent with our previous observations.

## E.4 ADDITIONAL CASE STUDY

As shown in Figure 14, Figure 15, Figure 16, Figure 17, Figure 18, Figure 19, Figure 20, and Figure 21, we present additional examples of responses from three representative MLLMs across a range of emotional tasks.

Table 3: **Overall Performance Comparison** (%) **on ER-Lab.** The top three performing results are highlighted in red (1st), blue (2nd), and yellow (3rd) backgrounds, respectively.

| Model | LLM Size | A | V | T | Avg Step | Avg Token | Rec-S | Rea-S | CoT-S |
|---|---|---|---|---|---|---|---|---|---|
| *Open-source MLLMs* | | | | | | | | | |
| Qwen2-Audio (Chu et al., 2024) | 7B | ✓ | | ✓ | 1.7 | 13.4 | 20.4 | 32.3 | 26.3 |
| Audio-Reasoner (Xie et al., 2025) | 7B | ✓ | | ✓ | 5.2 | 358.5 | 32.6 | 72.6 | 52.6 |
| Qwen2-VL-7B (Wang et al., 2024) | 7B | | ✓ | ✓ | 1.2 | 13.1 | 15.2 | 3.0 | 9.1 |
| Qwen2-VL-72B (Wang et al., 2024) | 72B | | ✓ | ✓ | 2.1 | 58.5 | 20.6 | 21.7 | 21.2 |
| Qwen2.5-VL-7B (Bai et al., 2025) | 7B | | ✓ | ✓ | 4.6 | 180.1 | 22.0 | 62.0 | 42.0 |
| Qwen2.5-VL-72B (Bai et al., 2025) | 72B | | ✓ | ✓ | 4.5 | 220.8 | 21.2 | 60.2 | 40.7 |
| QVQ (Team, 2024) | 72B | | ✓ | ✓ | 5.4 | 928.4 | 21.2 | 62.4 | 41.8 |
| Video-LLaVA (Lin et al., 2023) | 7B | | ✓ | ✓ | 1.6 | 14.7 | 11.2 | 8.7 | 9.9 |
| Video-LLaMA (Zhang et al., 2023a) | 7B | ✓ | ✓ | ✓ | 4.4 | 94.8 | 15.0 | 34.7 | 24.8 |
| Video-LLaMA2 (Cheng et al., 2024b) | 7B | ✓ | ✓ | ✓ | 1.0 | 5.4 | 23.4 | 0.3 | 11.9 |
| Qwen2.5-Omni (Xu et al., 2025) | 7B | ✓ | ✓ | ✓ | 4.2 | 109.6 | 19.8 | 50.7 | 35.2 |
| Emotion-LLaMA (Cheng et al., 2024a) | 7B | ✓ | ✓ | ✓ | 1.0 | 1.0 | 21.0 | 0.0 | 10.5 |
| HumanOmni (Zhao et al., 2025b) | 7B | ✓ | ✓ | ✓ | 1.0 | 1.1 | 24.8 | 0.0 | 12.4 |
| R1-Omni (Zhao et al., 2025a) | 0.5B | ✓ | ✓ | ✓ | 4.8 | 151.4 | 20.6 | 58.0 | 39.3 |
| AffectGPT (Lian et al., 2024c) | 7B | ✓ | ✓ | ✓ | 4.6 | 185.2 | 18.0 | 64.4 | 41.2 |
| *Closed-source MLLMs* | | | | | | | | | |
| GPT-4o (OpenAI, 2024) | — | | ✓ | ✓ | 4.8 | 242.4 | 21.8 | 79.4 | 50.6 |
| GPT-4.1 (OpenAI, 2025) | — | | ✓ | ✓ | 5.6 | 150.7 | 21.2 | 46.6 | 33.9 |
| Gemini-2.0-Flash (Google, 2025a) | — | | ✓ | ✓ | 4.1 | 60.8 | 24.0 | 55.6 | 39.8 |
| Gemini-2.5-Flash (Google, 2025b) | — | | ✓ | ✓ | 4.5 | 363.4 | 19.6 | 48.3 | 33.9 |
| Gemini-2.5-Pro (Google, 2025c) | — | | ✓ | ✓ | 5.2 | 585.9 | 22.6 | 62.7 | 42.7 |

Table 4: **Overall Performance Comparison** (%) **on ER-Wild.** The top three performing results are highlighted in red (1ˢᵗ), blue (2ⁿᵈ), and yellow (3ʳᵈ) backgrounds, respectively.

| Model | LLM Size | A | V | T | Avg Step | Avg Token | Rec-S | Rea-S | CoT-S |
|---|---|---|---|---|---|---|---|---|---|
| *Open-source MLLMs* | | | | | | | | | |
| Qwen2-Audio (Chu et al., 2024) | 7B | ✓ | | ✓ | 2.8 | 32.1 | 22.3 | 38.4 | 30.3 |
| Audio-Reasoner (Xie et al., 2025) | 7B | ✓ | | ✓ | 5.1 | 387.5 | 28.8 | 60.4 | 44.6 |
| Qwen2-VL-7B (Wang et al., 2024) | 7B | | ✓ | ✓ | 2.7 | 53.8 | 31.5 | 32.3 | 31.9 |
| Qwen2-VL-72B (Wang et al., 2024) | 72B | | ✓ | ✓ | 1.9 | 30.7 | 33.3 | 15.4 | 24.3 |
| Qwen2.5-VL-7B (Bai et al., 2025) | 7B | | ✓ | ✓ | 4.8 | 162.7 | 28.9 | 55.3 | 42.1 |
| Qwen2.5-VL-72B (Bai et al., 2025) | 72B | | ✓ | ✓ | 4.9 | 293.7 | 32.4 | 67.2 | 49.8 |
| QVQ (Team, 2024) | 72B | | ✓ | ✓ | 5.4 | 826.8 | 31.8 | 61.6 | 46.7 |
| Video-LLaVA (Lin et al., 2023) | 7B | | ✓ | ✓ | 1.0 | 5.4 | 21.2 | 2.2 | 11.7 |
| Video-LLaMA (Zhang et al., 2023a) | 7B | ✓ | ✓ | ✓ | 4.3 | 94.2 | 25.5 | 44.7 | 35.1 |
| Video-LLaMA2 (Cheng et al., 2024b) | 7B | ✓ | ✓ | ✓ | 4.3 | 94.2 | 25.4 | 44.7 | 35.0 |
| Qwen2.5-Omni (Xu et al., 2025) | 7B | ✓ | ✓ | ✓ | 3.7 | 78.1 | 19.0 | 54.3 | 36.7 |
| Emotion-LLaMA (Cheng et al., 2024a) | 7B | ✓ | ✓ | ✓ | 1.0 | 1.1 | 32.3 | 0.0 | 16.1 |
| HumanOmni (Zhao et al., 2025b) | 7B | ✓ | ✓ | ✓ | 1.0 | 1.5 | 52.8 | 0.0 | 26.4 |
| R1-Omni (Zhao et al., 2025a) | 0.5B | ✓ | ✓ | ✓ | 5.1 | 156.3 | 34.3 | 52.8 | 43.6 |
| AffectGPT (Lian et al., 2024c) | 7B | ✓ | ✓ | ✓ | 4.8 | 115.3 | 8.3 | 41.9 | 25.1 |
| *Closed-source MLLMs* | | | | | | | | | |
| GPT-4o (OpenAI, 2024) | — | | ✓ | ✓ | 4.6 | 172.5 | 26.8 | 74.8 | 50.8 |
| GPT-4.1 (OpenAI, 2025) | — | | ✓ | ✓ | 5.3 | 148.3 | 29.3 | 54.8 | 42.1 |
| Gemini-2.0-Flash (Google, 2025a) | — | | ✓ | ✓ | 4.2 | 62.1 | 41.1 | 56.9 | 49.0 |
| Gemini-2.5-Flash (Google, 2025b) | — | | ✓ | ✓ | 4.6 | 279.9 | 35.8 | 54.9 | 45.3 |
| Gemini-2.5-Pro (Google, 2025c) | — | | ✓ | ✓ | 5.1 | 527.5 | 36.7 | 67.3 | 52.0 |

Table 5: **Overall Performance Comparison** (%) **on Noise-ER.** The top three performing results are highlighted in red (1ˢᵗ), blue (2ⁿᵈ), and yellow (3ʳᵈ) backgrounds, respectively.

| Model | LLM Size | A | V | T | Avg Step | Avg Token | Rec-S | Rea-S | CoT-S |
|---|---|---|---|---|---|---|---|---|---|
| *Open-source MLLMs* | | | | | | | | | |
| Qwen2-Audio (Chu et al., 2024) | 7B | ✓ | | ✓ | 3.1 | 29.0 | 38.8 | 57.6 | 48.2 |
| Audio-Reasoner (Xie et al., 2025) | 7B | ✓ | | ✓ | 4.9 | 353.7 | 47.6 | 80. 4 | 64.0 |
| Qwen2-VL-7B (Wang et al., 2024) | 7B | | ✓ | ✓ | 3.4 | 57.1 | 45.4 | 50.2 | 47.8 |
| Qwen2-VL-72B (Wang et al., 2024) | 72B | | ✓ | ✓ | 1.9 | 38.0 | 47.4 | 21.6 | 34.5 |
| Qwen2.5-VL-7B (Bai et al., 2025) | 7B | | ✓ | ✓ | 4.7 | 147.4 | 37.8 | 62.9 | 50.3 |
| Qwen2.5-VL-72B (Bai et al., 2025) | 72B | | ✓ | ✓ | 4.9 | 284.7 | 41.2 | 74.6 | 57.9 |
| QVQ (Team, 2024) | 72B | | ✓ | ✓ | 5.4 | 718.3 | 39.6 | 72.7 | 56.2 |
| Video-LLaVA (Lin et al., 2023) | 7B | | ✓ | ✓ | 1.7 | 11.5 | 37.6 | 22.3 | 29.9 |
| Video-LLaMA (Zhang et al., 2023a) | 7B | ✓ | ✓ | ✓ | 4.2 | 70.4 | 28.0 | 38.1 | 33.0 |
| Video-LLaMA2 (Cheng et al., 2024b) | 7B | ✓ | ✓ | ✓ | 1.5 | 8.5 | 45.0 | 13.0 | 29.0 |
| Qwen2.5-Omni (Xu et al., 2025) | 7B | ✓ | ✓ | ✓ | 3.7 | 82.8 | 24.2 | 60.1 | 42.1 |
| Emotion-LLaMA (Cheng et al., 2024a) | 7B | ✓ | ✓ | ✓ | 1.0 | 1.1 | 58.4 | 0.0 | 29.2 |
| HumanOmni (Zhao et al., 2025b) | 7B | ✓ | ✓ | ✓ | 1.0 | 1.3 | 68.4 | 0.0 | 34.2 |
| R1-Omni (Zhao et al., 2025a) | 0.5B | ✓ | ✓ | ✓ | 5.3 | 185.7 | 52.0 | 67.4 | 59.7 |
| AffectGPT (Lian et al., 2024c) | 7B | ✓ | ✓ | ✓ | 4.9 | 121.3 | 16.8 | 59.5 | 38.1 |
| *Closed-source MLLMs* | | | | | | | | | |
| GPT-4o (OpenAI, 2024) | — | | ✓ | ✓ | 4.4 | 144.1 | 27.8 | 78.2 | 53.0 |
| GPT-4.1 (OpenAI, 2025) | — | | ✓ | ✓ | 5.2 | 138.3 | 32.8 | 61.3 | 47.0 |
| Gemini-2.0-Flash (Google, 2025a) | — | | ✓ | ✓ | 4.2 | 60.2 | 52.4 | 67.5 | 60.0 |
| Gemini-2.5-Flash (Google, 2025b) | — | | ✓ | ✓ | 5.0 | 301.6 | 54.4 | 73.2 | 63.8 |
| Gemini-2.5-Pro (Google, 2025c) | — | | ✓ | ✓ | 5.3 | 503.6 | 57.6 | 81.2 | 69.4 |

Table 6: **Overall Performance Comparison** (%) **on FG-ER.** The top three performing results are highlighted in red (1st), blue (2nd), and yellow (3rd) backgrounds, respectively.

| Model | LLM Size | A | V | T | Avg Step | Avg Token | Rec-S | Rea-S | CoT-S |
|---|---|---|---|---|---|---|---|---|---|
| *Open-source MLLMs* | | | | | | | | | |
| Qwen2-Audio (Chu et al., 2024) | 7B | ✓ | | ✓ | 2.7 | 51.2 | 21.7 | 22.0 | 21.8 |
| Audio-Reasoner (Xie et al., 2025) | 7B | ✓ | | ✓ | 5.0 | 585.1 | 9.1 | 47.3 | 28.2 |
| Qwen2-VL-7B (Wang et al., 2024) | 7B | | ✓ | ✓ | 1.1 | 10.9 | 26.4 | 4.5 | 15.4 |
| Qwen2-VL-72B (Wang et al., 2024) | 72B | | ✓ | ✓ | 1.0 | 9.2 | 23.9 | 2.9 | 13.4 |
| Qwen2.5-VL-7B (Bai et al., 2025) | 7B | | ✓ | ✓ | 4.8 | 197.4 | 19.5 | 62.2 | 40.9 |
| Qwen2.5-VL-72B (Bai et al., 2025) | 72B | | ✓ | ✓ | 4.9 | 279.7 | 27.3 | 69.8 | 48.6 |
| QVQ (Team, 2024) | 72B | | ✓ | ✓ | 5.6 | 1044.8 | 26.6 | 67.8 | 47.2 |
| Video-LLaVA (Lin et al., 2023) | 7B | | ✓ | ✓ | 2.7 | 26.0 | 10.9 | 26.6 | 18.7 |
| Video-LLaMA (Zhang et al., 2023a) | 7B | ✓ | ✓ | ✓ | 5.1 | 236.7 | 33.7 | 34.6 | 34.1 |
| Video-LLaMA2 (Cheng et al., 2024b) | 7B | ✓ | ✓ | ✓ | 1.2 | 20.0 | 21.3 | 5.3 | 13.3 |
| Qwen2.5-Omni (Xu et al., 2025) | 7B | ✓ | ✓ | ✓ | 3.3 | 72.9 | 10.4 | 42.3 | 26.3 |
| Emotion-LLaMA (Cheng et al., 2024a) | 7B | ✓ | ✓ | ✓ | 1.5 | 15.4 | 10.7 | 4.3 | 7.5 |
| HumanOmni (Zhao et al., 2025b) | 7B | ✓ | ✓ | ✓ | 1.0 | 1.5 | 16.5 | 0.8 | 8.6 |
| R1-Omni (Zhao et al., 2025a) | 0.5B | ✓ | ✓ | ✓ | 4.7 | 135.1 | 5.1 | 49.5 | 27.3 |
| AffectGPT (Lian et al., 2024c) | 7B | ✓ | ✓ | ✓ | 4.8 | 115.5 | 5.2 | 40.6 | 22.9 |
| *Closed-source MLLMs* | | | | | | | | | |
| GPT-4o (OpenAI, 2024) | — | | ✓ | ✓ | 4.4 | 198.0 | 31.8 | 68.6 | 50.2 |
| GPT-4.1 (OpenAI, 2025) | — | | ✓ | ✓ | 5.2 | 145.3 | 28.4 | 67.3 | 47.8 |
| Gemini-2.0-Flash (Google, 2025a) | — | | ✓ | ✓ | 4.4 | 104.2 | 30.9 | 58.6 | 44.7 |
| Gemini-2.5-Flash (Google, 2025b) | — | | ✓ | ✓ | 3.5 | 354.1 | 20.5 | 37.8 | 29.2 |
| Gemini-2.5-Pro (Google, 2025c) | — | | ✓ | ✓ | 5.1 | 473.7 | 38.6 | 61.0 | 49.8 |

Table 7: **Overall Performance Comparison** (%) **on ML-ER.** The top three performing results are highlighted in red (1st), blue (2nd), and yellow (3rd) backgrounds, respectively.

| Model | LLM Size | A | V | T | Avg Step | Avg Token | Rec-S | Rea-S | CoT-S |
|---|---|---|---|---|---|---|---|---|---|
| *Open-source MLLMs* | | | | | | | | | |
| Qwen2-Audio (Chu et al., 2024) | 7B | ✓ | | ✓ | 3.9 | 101.0 | 53.5 | 72.8 | 63.2 |
| Audio-Reasoner (Xie et al., 2025) | 7B | ✓ | | ✓ | 5.2 | 447.6 | 41.1 | 77.5 | 59.3 |
| Qwen2-VL-7B (Wang et al., 2024) | 7B | | ✓ | ✓ | 2.1 | 92.0 | 30.3 | 24.3 | 27.3 |
| Qwen2-VL-72B (Wang et al., 2024) | 72B | | ✓ | ✓ | 1.0 | 8.4 | 33.5 | 0.8 | 17.1 |
| Qwen2.5-VL-7B (Bai et al., 2025) | 7B | | ✓ | ✓ | 4.8 | 193.4 | 23.6 | 75.2 | 49.4 |
| Qwen2.5-VL-72B (Bai et al., 2025) | 72B | | ✓ | ✓ | 4.9 | 279.4 | 31.2 | 88.6 | 59.9 |
| QVQ (Team, 2024) | 72B | | ✓ | ✓ | 5.7 | 892.3 | 25.2 | 82.3 | 53.7 |
| Video-LLaVA (Lin et al., 2023) | 7B | | ✓ | ✓ | 3.7 | 58.5 | 52.7 | 57.1 | 54.9 |
| Video-LLaMA (Zhang et al., 2023a) | 7B | ✓ | ✓ | ✓ | 5.0 | 231.7 | 29.6 | 68.7 | 49.2 |
| Video-LLaMA2 (Cheng et al., 2024b) | 7B | ✓ | ✓ | ✓ | 2.2 | 23.3 | 40.1 | 31.0 | 35.5 |
| Qwen2.5-Omni (Xu et al., 2025) | 7B | ✓ | ✓ | ✓ | 3.4 | 83.8 | 11.0 | 64.9 | 38.0 |
| Emotion-LLaMA (Cheng et al., 2024a) | 7B | ✓ | ✓ | ✓ | 1.0 | 1.2 | 5.7 | 0.5 | 3.1 |
| HumanOmni (Zhao et al., 2025b) | 7B | ✓ | ✓ | ✓ | 1.0 | 1.5 | 34.3 | 2.7 | 18.5 |
| R1-Omni (Zhao et al., 2025a) | 0.5B | ✓ | ✓ | ✓ | 5.1 | 149.8 | 23.1 | 76.2 | 49.6 |
| AffectGPT (Lian et al., 2024c) | 7B | ✓ | ✓ | ✓ | 4.9 | 118.5 | 10.8 | 65.9 | 38.3 |
| *Closed-source MLLMs* | | | | | | | | | |
| GPT-4o (OpenAI, 2024) | — | | ✓ | ✓ | 4.2 | 162.9 | 29.9 | 82.5 | 56.2 |
| GPT-4.1 (OpenAI, 2025) | — | | ✓ | ✓ | 5.5 | 163.5 | 35.0 | 76.2 | 55.6 |
| Gemini-2.0-Flash (Google, 2025a) | — | | ✓ | ✓ | 4.3 | 81.8 | 38.3 | 71.7 | 55.0 |
| Gemini-2.5-Flash (Google, 2025b) | — | | ✓ | ✓ | 2.5 | 140.6 | 38.7 | 30.1 | 34.4 |
| Gemini-2.5-Pro (Google, 2025c) | — | | ✓ | ✓ | 5.1 | 650.2 | 50.6 | 80.5 | 65.5 |

Table 8: **Overall Performance Comparison** (%) **on SA.** The top three performing results are highlighted in red (1st), blue (2nd), and yellow (3rd) backgrounds, respectively.

| Model | LLM Size | A | V | T | Avg Step | Avg Token | Rec-S | Rea-S | CoT-S |
|---|---|---|---|---|---|---|---|---|---|
| *Open-source MLLMs* | | | | | | | | | |
| Qwen2-Audio (Chu et al., 2024) | 7B | ✓ | | ✓ | 3.2 | 35.3 | 53.6 | 65.4 | 59.5 |
| Audio-Reasoner (Xie et al., 2025) | 7B | ✓ | | ✓ | 4.9 | 230.2 | 62.2 | 81.9 | 72.0 |
| Qwen2-VL-7B (Wang et al., 2024) | 7B | | ✓ | ✓ | 4.0 | 95.6 | 34.1 | 60.6 | 47.3 |
| Qwen2-VL-72B (Wang et al., 2024) | 72B | | ✓ | ✓ | 1.4 | 22.2 | 34.8 | 9.3 | 22.1 |
| Qwen2.5-VL-7B (Bai et al., 2025) | 7B | | ✓ | ✓ | 4.9 | 161.0 | 36.5 | 70.7 | 53.6 |
| Qwen2.5-VL-72B (Bai et al., 2025) | 72B | | ✓ | ✓ | 4.8 | 227.7 | 37.5 | 81.9 | 59.7 |
| QVQ (Team, 2024) | 72B | | ✓ | ✓ | 5.7 | 823.1 | 43.0 | 76.2 | 59.6 |
| Video-LLaVA (Lin et al., 2023) | 7B | | ✓ | ✓ | 3.0 | 18.4 | 35.8 | 55.8 | 45.8 |
| Video-LLaMA (Zhang et al., 2023a) | 7B | ✓ | ✓ | ✓ | 4.5 | 96.2 | 31.3 | 50.7 | 41.0 |
| Video-LLaMA2 (Cheng et al., 2024b) | 7B | ✓ | ✓ | ✓ | 2.1 | 15.9 | 37.6 | 27.3 | 32.5 |
| Qwen2.5-Omni (Xu et al., 2025) | 7B | ✓ | ✓ | ✓ | 3.7 | 73.8 | 24.6 | 68.1 | 46.3 |
| Emotion-LLaMA (Cheng et al., 2024a) | 7B | ✓ | ✓ | ✓ | 1.0 | 1.0 | 26.1 | 0.1 | 13.1 |
| HumanOmni (Zhao et al., 2025b) | 7B | ✓ | ✓ | ✓ | 1.0 | 1.1 | 32.1 | 0.0 | 16.0 |
| R1-Omni (Zhao et al., 2025a) | 0.5B | ✓ | ✓ | ✓ | 5.1 | 155.7 | 27.8 | 60.6 | 44.2 |
| AffectGPT (Lian et al., 2024c) | 7B | ✓ | ✓ | ✓ | 4.9 | 118.4 | 17.6 | 52.3 | 35.0 |
| *Closed-source MLLMs* | | | | | | | | | |
| GPT-4o (OpenAI, 2024) | — | | ✓ | ✓ | 4.2 | 144.3 | 34.6 | 85.7 | 60.2 |
| GPT-4.1 (OpenAI, 2025) | — | | ✓ | ✓ | 4.9 | 119.5 | 33.8 | 74.3 | 54.0 |
| Gemini-2.0-Flash (Google, 2025a) | — | | ✓ | ✓ | 4.0 | 52.5 | 42.1 | 61.1 | 51.6 |
| Gemini-2.5-Flash (Google, 2025b) | — | | ✓ | ✓ | 4.2 | 208.0 | 44.2 | 55.9 | 50.1 |
| Gemini-2.5-Pro (Google, 2025c) | — | | ✓ | ✓ | 5.0 | 477.5 | 48.2 | 78.7 | 63.4 |

Table 9: **Overall Performance Comparison** (%) **on FG-SA.** The top three performing results are highlighted in red (1st), blue (2nd), and yellow (3rd) backgrounds, respectively.

| Model | LLM Size | A | V | T | Avg Step | Avg Token | Rec-S | Rea-S | CoT-S |
|---|---|---|---|---|---|---|---|---|---|
| *Open-source MLLMs* | | | | | | | | | |
| Qwen2-Audio (Chu et al., 2024) | 7B | ✓ | | ✓ | 3.1 | 38.7 | 19.6 | 62.5 | 41.1 |
| Audio-Reasoner (Xie et al., 2025) | 7B | ✓ | | ✓ | 4.8 | 310.9 | 30.5 | 82.6 | 56.6 |
| Qwen2-VL-7B (Wang et al., 2024) | 7B | | ✓ | ✓ | 3.1 | 90.2 | 7.6 | 37.6 | 22.6 |
| Qwen2-VL-72B (Wang et al., 2024) | 72B | | ✓ | ✓ | 1.3 | 17.1 | 10.0 | 3.9 | 7.0 |
| Qwen2.5-VL-7B (Bai et al., 2025) | 7B | | ✓ | ✓ | 4.8 | 165.0 | 12.2 | 56.6 | 34.4 |
| Qwen2.5-VL-72B (Bai et al., 2025) | 72B | | ✓ | ✓ | 5.0 | 241.9 | 14.2 | 75.8 | 45.0 |
| QVQ (Team, 2024) | 72B | | ✓ | ✓ | 5.4 | 907.6 | 13.0 | 60.2 | 36.6 |
| Video-LLaVA (Lin et al., 2023) | 7B | | ✓ | ✓ | 3.0 | 13.8 | 5.8 | 73.6 | 39.7 |
| Video-LLaMA (Zhang et al., 2023a) | 7B | ✓ | ✓ | ✓ | 4.6 | 134.8 | 14.6 | 54.6 | 34.6 |
| Video-LLaMA2 (Cheng et al., 2024b) | 7B | ✓ | ✓ | ✓ | 2.9 | 27.3 | 13.0 | 41.3 | 27.2 |
| Qwen2.5-Omni (Xu et al., 2025) | 7B | ✓ | ✓ | ✓ | 3.6 | 79.0 | 6.2 | 45.2 | 25.7 |
| Emotion-LLaMA (Cheng et al., 2024a) | 7B | ✓ | ✓ | ✓ | 1.0 | 1.3 | 5.8 | 0.0 | 2.9 |
| HumanOmni (Zhao et al., 2025b) | 7B | ✓ | ✓ | ✓ | 1.0 | 1.0 | 7.8 | 0.0 | 3.9 |
| R1-Omni (Zhao et al., 2025a) | 0.5B | ✓ | ✓ | ✓ | 4.9 | 142.7 | 7.6 | 51.9 | 29.8 |
| AffectGPT (Lian et al., 2024c) | 7B | ✓ | ✓ | ✓ | 4.8 | 120.7 | 12.8 | 51.3 | 32.1 |
| *Closed-source MLLMs* | | | | | | | | | |
| GPT-4o (OpenAI, 2024) | — | | ✓ | ✓ | 4.4 | 144.0 | 7.6 | 79.7 | 43.6 |
| GPT-4.1 (OpenAI, 2025) | — | | ✓ | ✓ | 5.0 | 116.7 | 7.4 | 70.9 | 39.1 |
| Gemini-2.0-Flash (Google, 2025a) | — | | ✓ | ✓ | 3.7 | 50.8 | 11.0 | 44.0 | 27.5 |
| Gemini-2.5-Flash (Google, 2025b) | — | | ✓ | ✓ | 4.5 | 272.6 | 15.2 | 47.4 | 31.3 |
| Gemini-2.5-Pro (Google, 2025c) | — | | ✓ | ✓ | 5.0 | 571.6 | 16.6 | 64.3 | 40.5 |

Table 10: **Overall Performance Comparison** (%) **on IR.** The top three performing results are highlighted in red (1st), blue (2nd), and yellow (3rd) backgrounds, respectively.

| Model | LLM Size | A | V | T | Avg Step | Avg Token | Rec-S | Rea-S | CoT-S |
|---|---|---|---|---|---|---|---|---|---|
| *Open-source MLLMs* | | | | | | | | | |
| Qwen2-Audio (Chu et al., 2024) | 7B | ✓ | | ✓ | 3.0 | 41.6 | 27.3 | 45.9 | 36.6 |
| Audio-Reasoner (Xie et al., 2025) | 7B | ✓ | | ✓ | 4.7 | 361.2 | 23.4 | 74.9 | 49.2 |
| Qwen2-VL-7B (Wang et al., 2024) | 7B | | ✓ | ✓ | 3.8 | 103.6 | 26.3 | 61.0 | 43.6 |
| Qwen2-VL-72B (Wang et al., 2024) | 72B | | ✓ | ✓ | 1.0 | 7.5 | 30.7 | 0.0 | 15.3 |
| Qwen2.5-VL-7B (Bai et al., 2025) | 7B | | ✓ | ✓ | 5.0 | 183.7 | 28.1 | 82.5 | 55.3 |
| Qwen2.5-VL-72B (Bai et al., 2025) | 72B | | ✓ | ✓ | 4.8 | 319.4 | 27.9 | 93.4 | 60.6 |
| QVQ (Team, 2024) | 72B | | ✓ | ✓ | 5.5 | 1417.8 | 25.7 | 84.0 | 54.8 |
| Video-LLaVA (Lin et al., 2023) | 7B | | ✓ | ✓ | 3.1 | 33.5 | 15.4 | 39.8 | 27.6 |
| Video-LLaMA (Zhang et al., 2023a) | 7B | ✓ | ✓ | ✓ | 4.7 | 130.8 | 20.2 | 62.5 | 41.4 |
| Video-LLaMA2 (Cheng et al., 2024b) | 7B | ✓ | ✓ | ✓ | 1.5 | 10.5 | 16.8 | 14.2 | 15.5 |
| Qwen2.5-Omni (Xu et al., 2025) | 7B | ✓ | ✓ | ✓ | 3.4 | 59.4 | 5.4 | 81.8 | 43.6 |
| Emotion-LLaMA (Cheng et al., 2024a) | 7B | ✓ | ✓ | ✓ | 1.0 | 1.9 | 23.8 | 0.3 | 12.0 |
| HumanOmni (Zhao et al., 2025b) | 7B | ✓ | ✓ | ✓ | 1.0 | 1.4 | 15.8 | 0.2 | 8.0 |
| R1-Omni (Zhao et al., 2025a) | 0.5B | ✓ | ✓ | ✓ | 5.2 | 175.6 | 16.6 | 58.7 | 37.7 |
| AffectGPT (Lian et al., 2024c) | 7B | ✓ | ✓ | ✓ | 4.9 | 117.5 | 1.8 | 44.0 | 22.9 |
| *Closed-source MLLMs* | | | | | | | | | |
| GPT-4o (OpenAI, 2024) | — | | ✓ | ✓ | 4.5 | 195.5 | 29.1 | 88.9 | 59.0 |
| GPT-4.1 (OpenAI, 2025) | — | | ✓ | ✓ | 5.3 | 172.6 | 28.7 | 75.8 | 52.2 |
| Gemini-2.0-Flash (Google, 2025a) | — | | ✓ | ✓ | 4.4 | 75.6 | 24.7 | 67.9 | 46.3 |
| Gemini-2.5-Flash (Google, 2025b) | — | | ✓ | ✓ | 4.7 | 277.7 | 24.6 | 65.5 | 45.1 |
| Gemini-2.5-Pro (Google, 2025c) | — | | ✓ | ✓ | 5.2 | 660.5 | 28.1 | 84.7 | 56.4 |

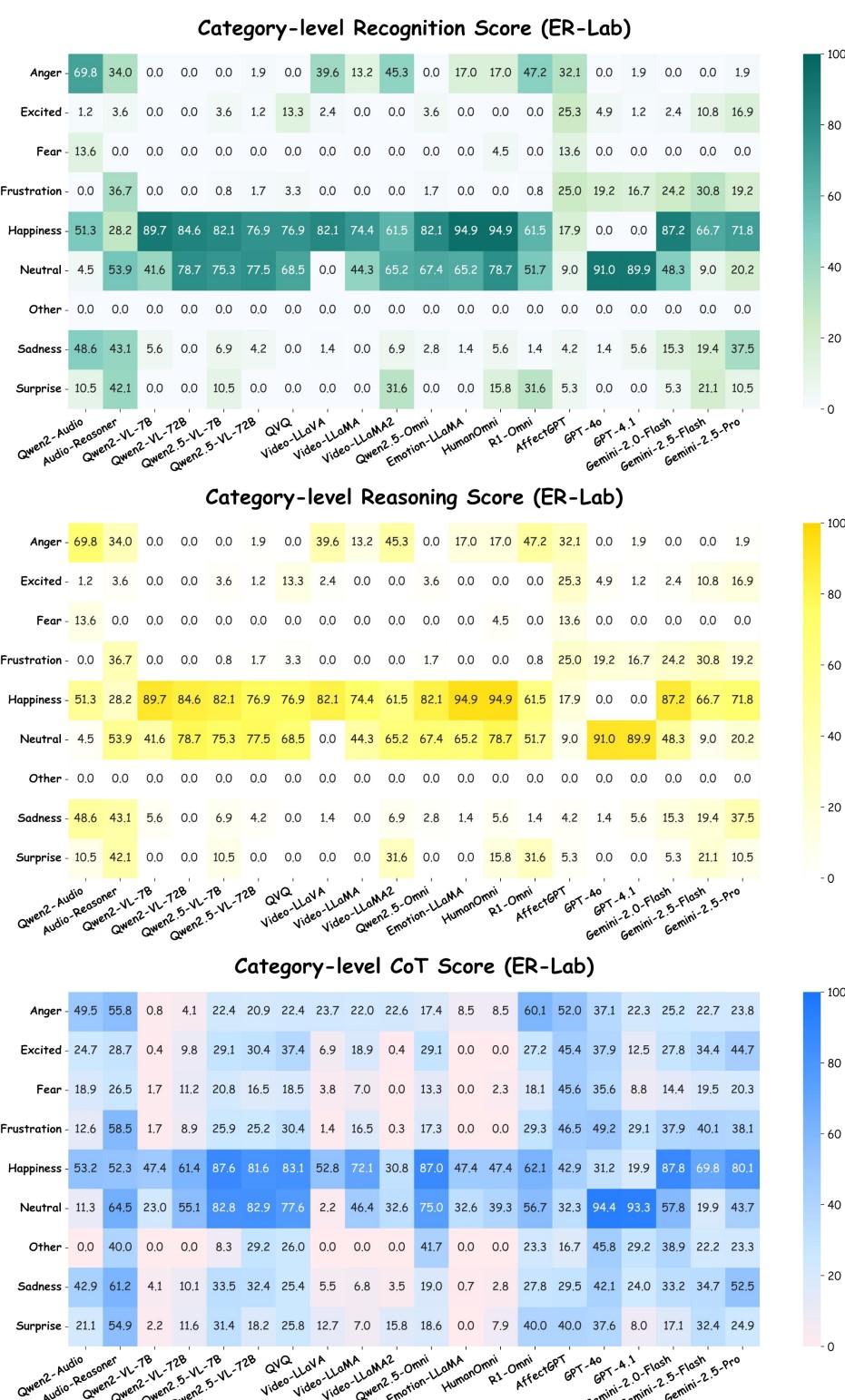

Figure 7: **Category-level Performance Comparison (%) on ER-Lab.** We showcase fine-grained comparison results of 20 MLLMs using Rec-S, Rea-S, and CoT-S across 9 emotion categories.

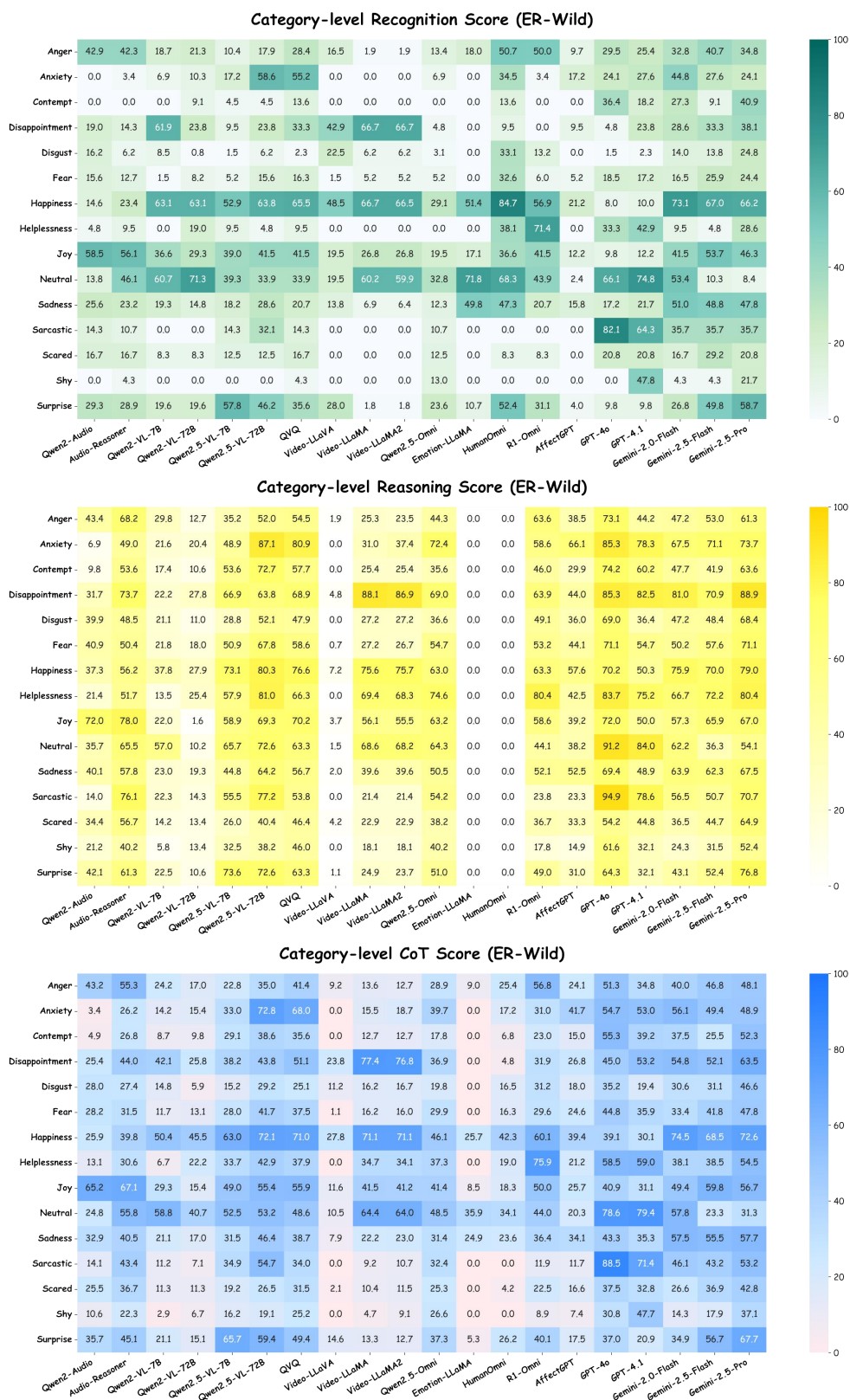

Figure 8: **Category-level Performance Comparison (%) on ER-Wild.** We showcase fine-grained comparison results of 20 MLLMs using Rec-S, Rea-S, and CoT-S across 15 emotion categories.

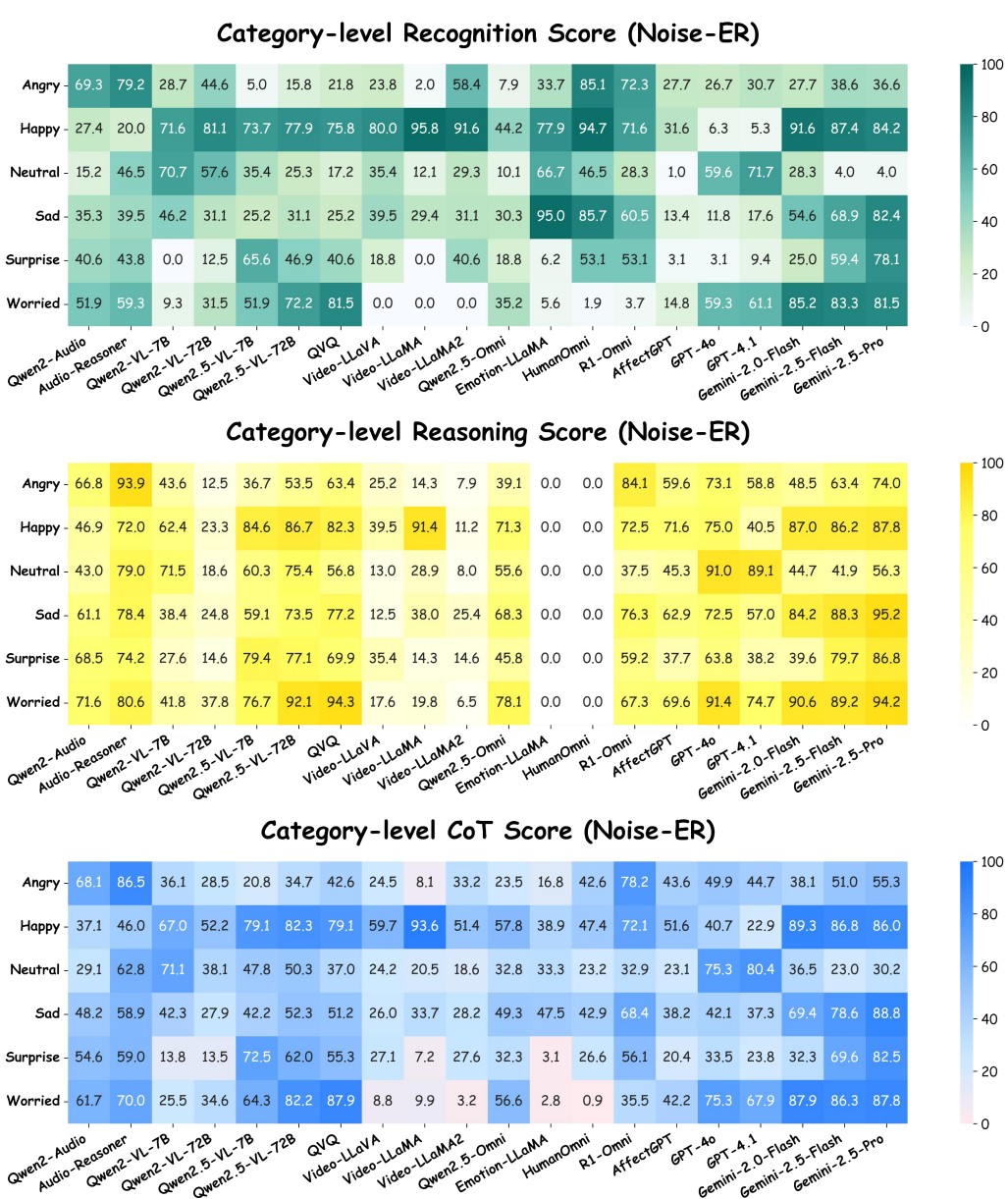

Figure 9: **Category-level Performance Comparison (%) on Noise-ER.** We showcase fine-grained comparison results of 20 MLLMs using Rec-S, Rea-S, and CoT-S across 6 emotion categories.

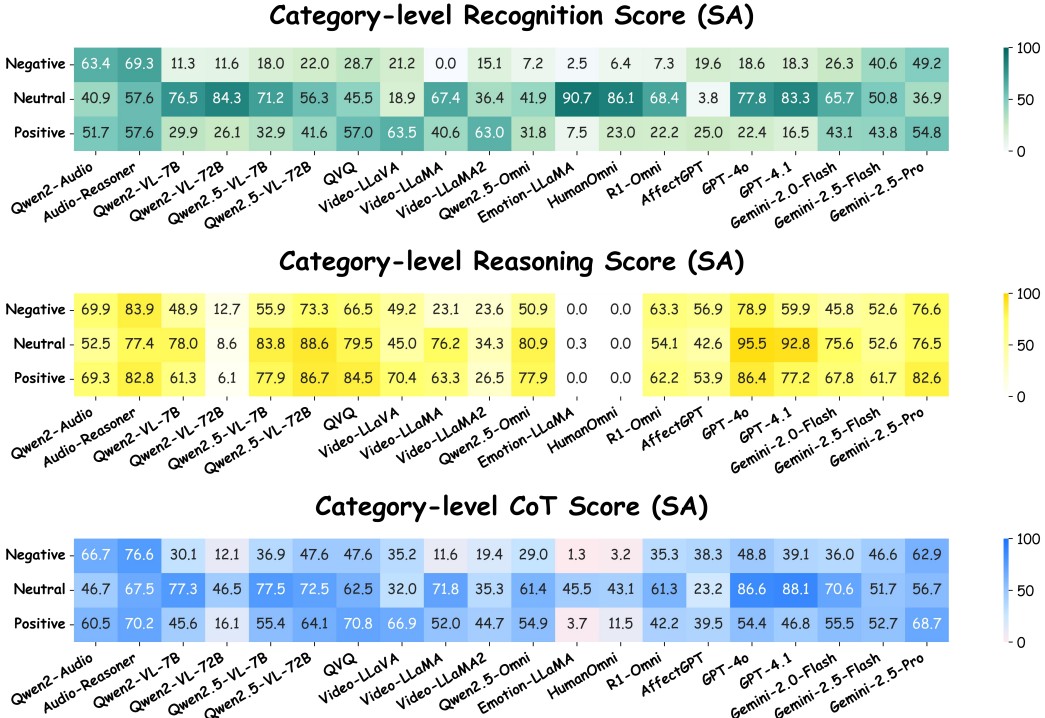

Figure 10: **Category-level Performance Comparison** (%) **on SA.** We showcase fine-grained comparison results of 20 MLLMs using Rec-S, Rea-S, and CoT-S across 3 sentiment categories.

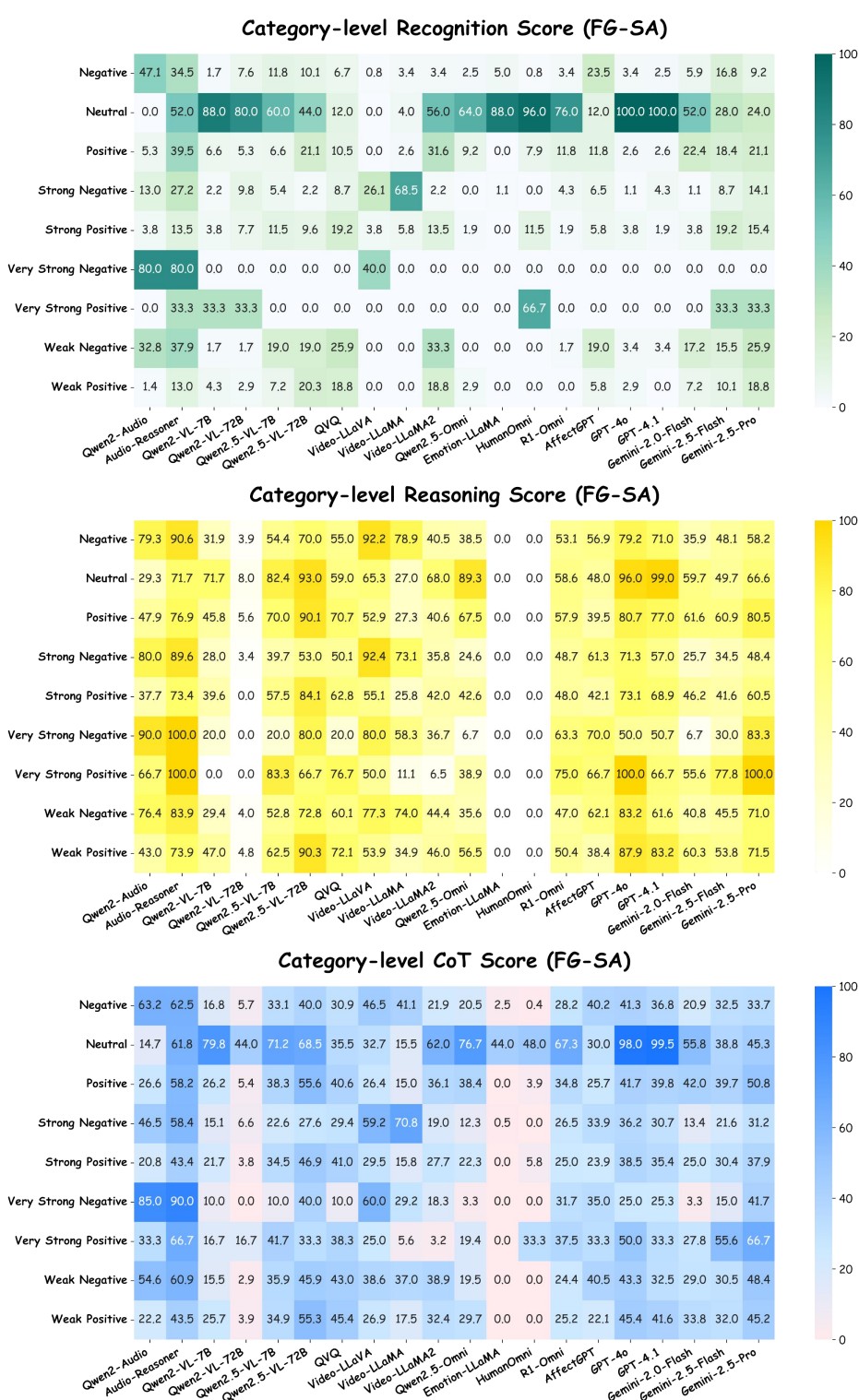

Figure 11: **Category-level Performance Comparison** (%) **on FG-SA.** We showcase fine-grained comparison results of 20 MLLMs using Rec-S, Rea-S, and CoT-S across 9 sentiment categories.

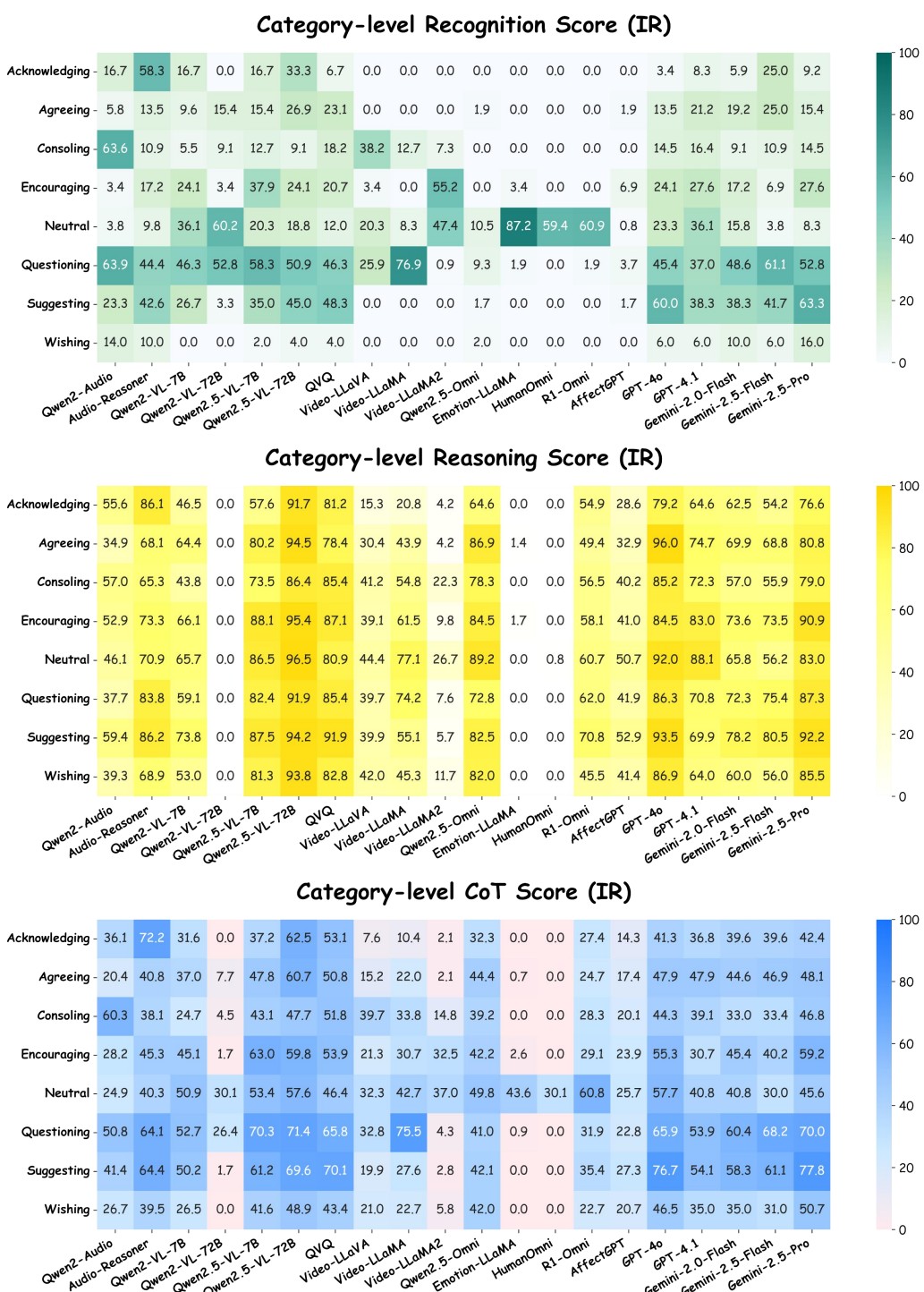

Figure 12: **Category-level Performance Comparison** (%) **on IR.** We showcase fine-grained comparison results of 20 MLLMs using Rec-S, Rea-S, and CoT-S across 9 intent categories.

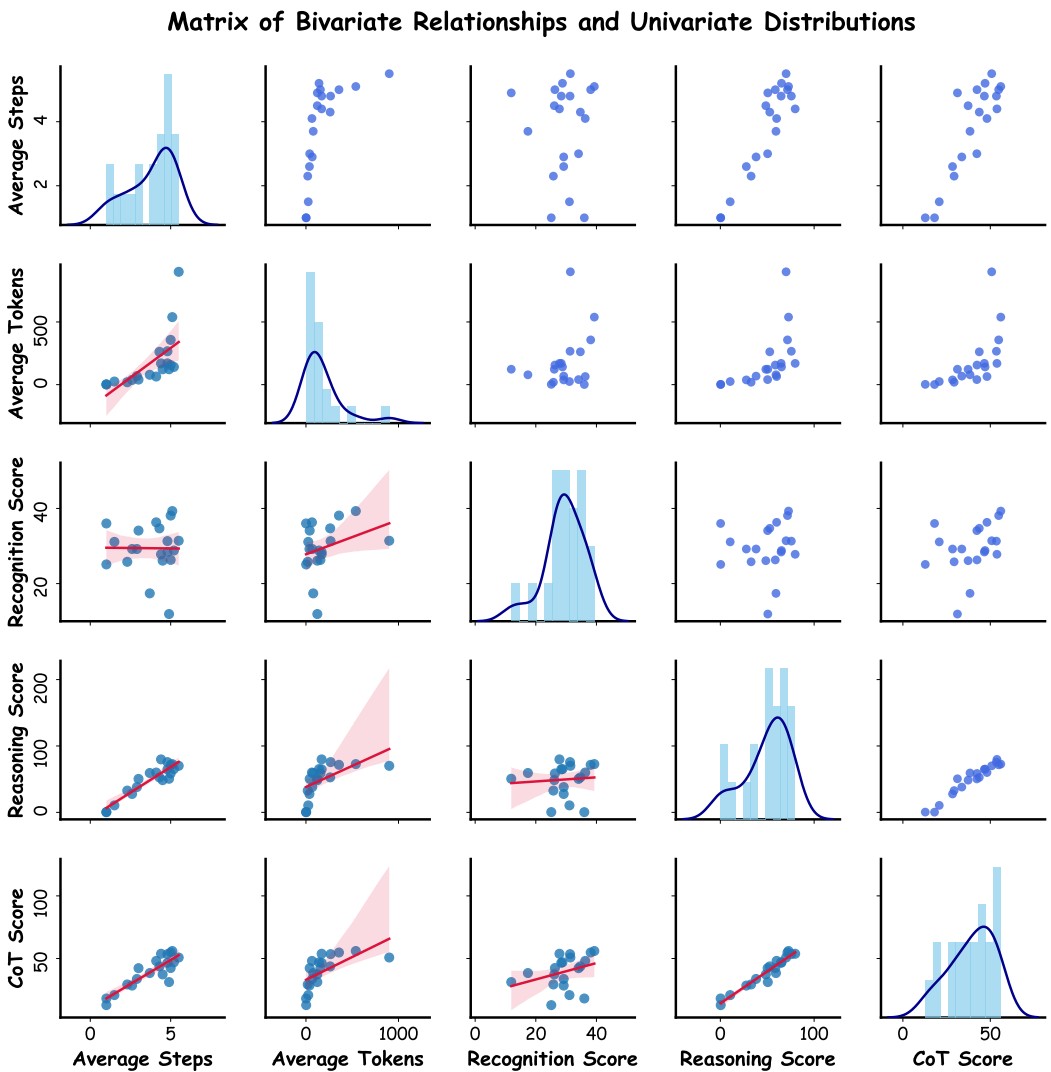

Figure 13: **Multivariate Analysis of Evaluation Metrics.** We showcase pairwise scatter plots with distribution diagnostics across five different evaluation metrics on 20 state-of-the-art MLLMs.

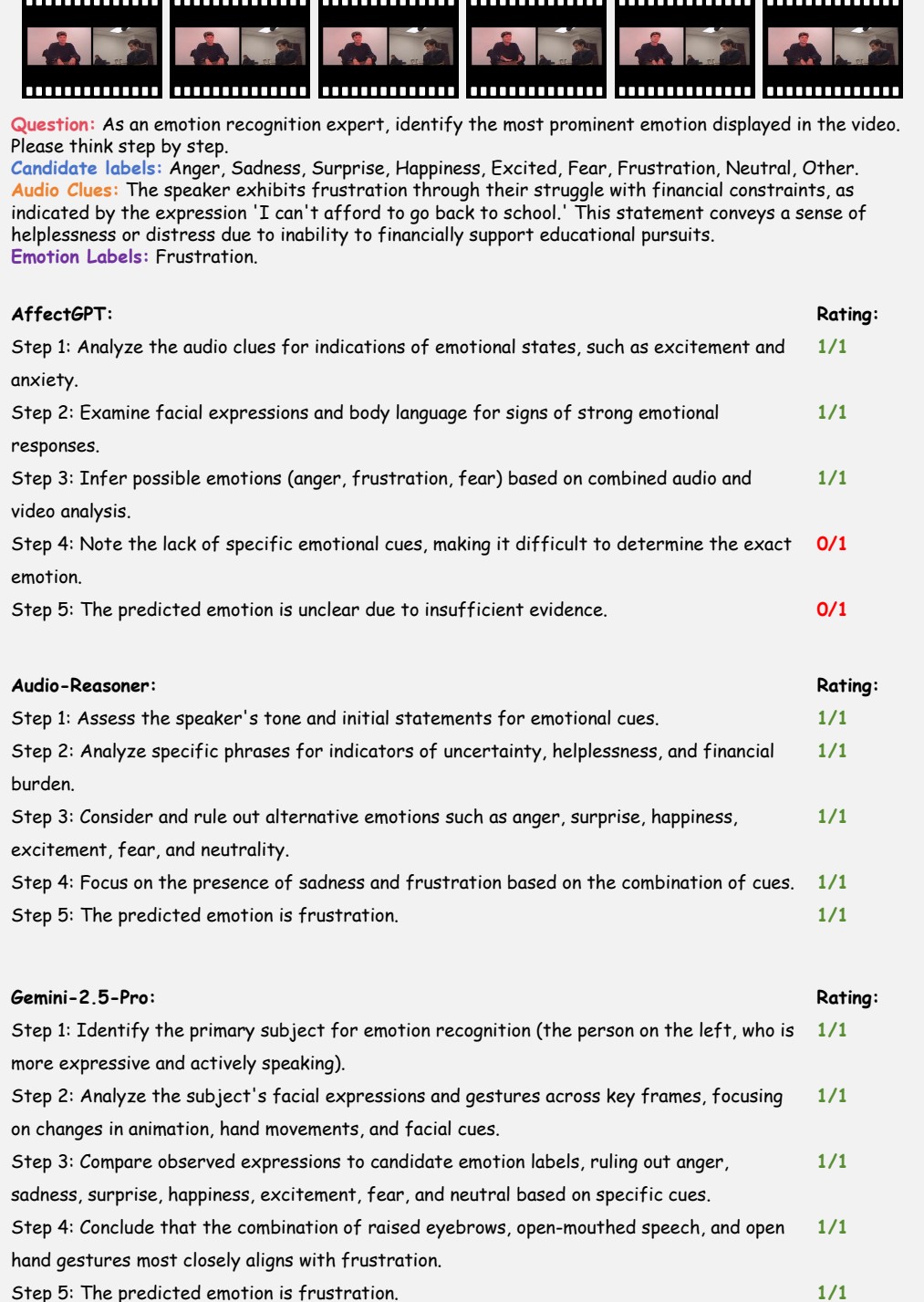

Figure 14: **Case Study on ER-Lab.** We showcase the answers and ratings of three MLLMs.

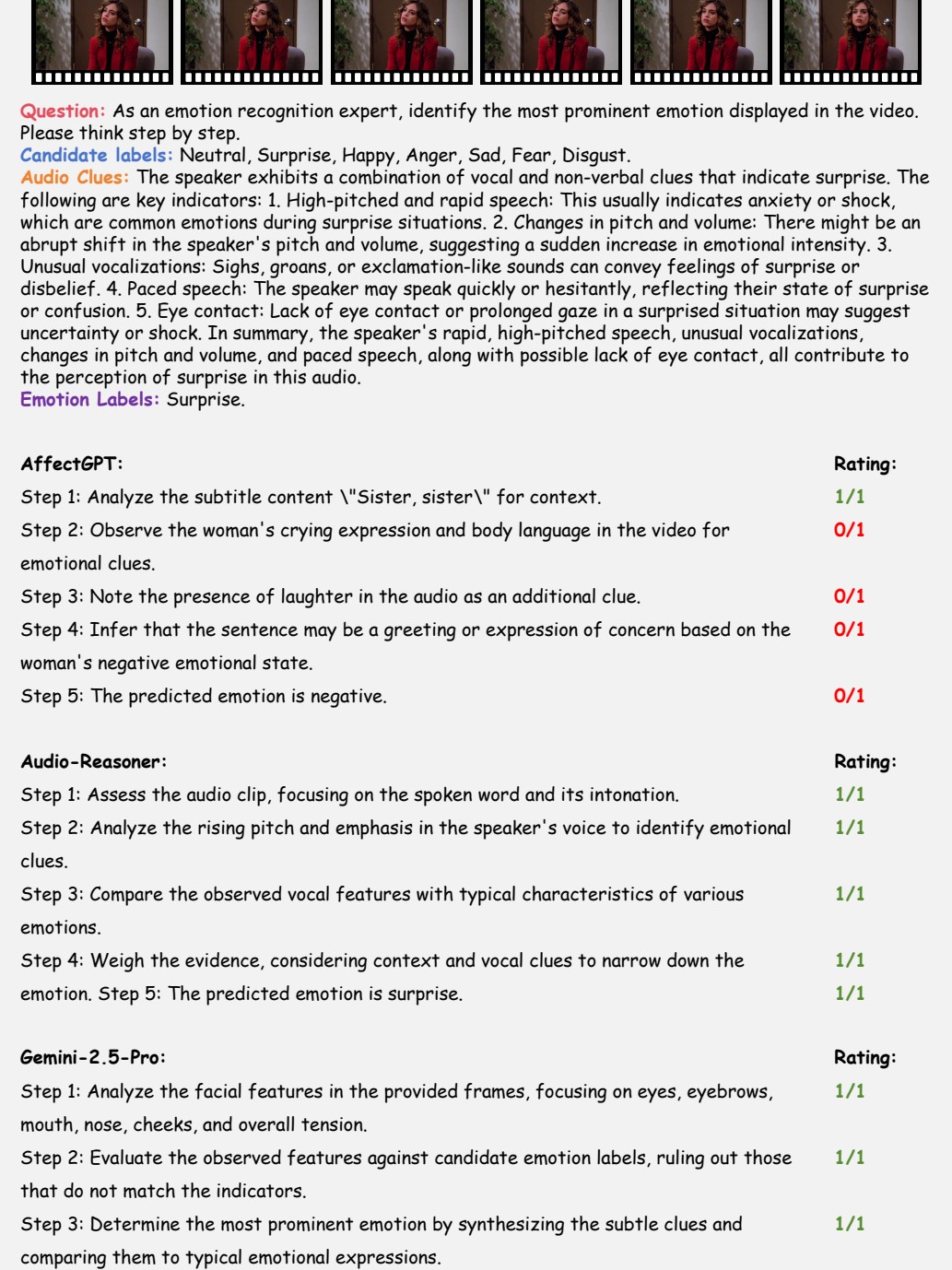

**Question:** As an emotion recognition expert, identify the most prominent emotion displayed in the video. Please think step by step.
**Candidate labels:** Neutral, Surprise, Happy, Anger, Sad, Fear, Disgust.
**Audio Clues:** The speaker exhibits a combination of vocal and non-verbal clues that indicate surprise. The following are key indicators: 1. High-pitched and rapid speech: This usually indicates anxiety or shock, which are common emotions during surprise situations. 2. Changes in pitch and volume: There might be an abrupt shift in the speaker's pitch and volume, suggesting a sudden increase in emotional intensity. 3. Unusual vocalizations: Sighs, groans, or exclamation-like sounds can convey feelings of surprise or disbelief. 4. Paced speech: The speaker may speak quickly or hesitantly, reflecting their state of surprise or confusion. 5. Eye contact: Lack of eye contact or prolonged gaze in a surprised situation may suggest uncertainty or shock. In summary, the speaker's rapid, high-pitched speech, unusual vocalizations, changes in pitch and volume, and paced speech, along with possible lack of eye contact, all contribute to the perception of surprise in this audio.
**Emotion Labels:** Surprise.

| **AffectGPT:** | **Rating:** |
|---|---|
| Step 1: Analyze the subtitle content \"Sister, sister\" for context. | 1/1 |
| Step 2: Observe the woman's crying expression and body language in the video for emotional clues. | 0/1 |
| Step 3: Note the presence of laughter in the audio as an additional clue. | 0/1 |
| Step 4: Infer that the sentence may be a greeting or expression of concern based on the woman's negative emotional state. | 0/1 |
| Step 5: The predicted emotion is negative. | 0/1 |

| **Audio-Reasoner:** | **Rating:** |
|---|---|
| Step 1: Assess the audio clip, focusing on the spoken word and its intonation. | 1/1 |
| Step 2: Analyze the rising pitch and emphasis in the speaker's voice to identify emotional clues. | 1/1 |
| Step 3: Compare the observed vocal features with typical characteristics of various emotions. | 1/1 |
| Step 4: Weigh the evidence, considering context and vocal clues to narrow down the emotion. Step 5: The predicted emotion is surprise. | 1/1 |
| | 1/1 |

| **Gemini-2.5-Pro:** | **Rating:** |
|---|---|
| Step 1: Analyze the facial features in the provided frames, focusing on eyes, eyebrows, mouth, nose, cheeks, and overall tension. | 1/1 |
| Step 2: Evaluate the observed features against candidate emotion labels, ruling out those that do not match the indicators. | 1/1 |
| Step 3: Determine the most prominent emotion by synthesizing the subtle clues and comparing them to typical emotional expressions. | 1/1 |
| Step 4: The predicted emotion is neutral. | 0/1 |

Figure 15: **Case Study on ER-Wild.** We showcase the answers and ratings of three MLLMs.

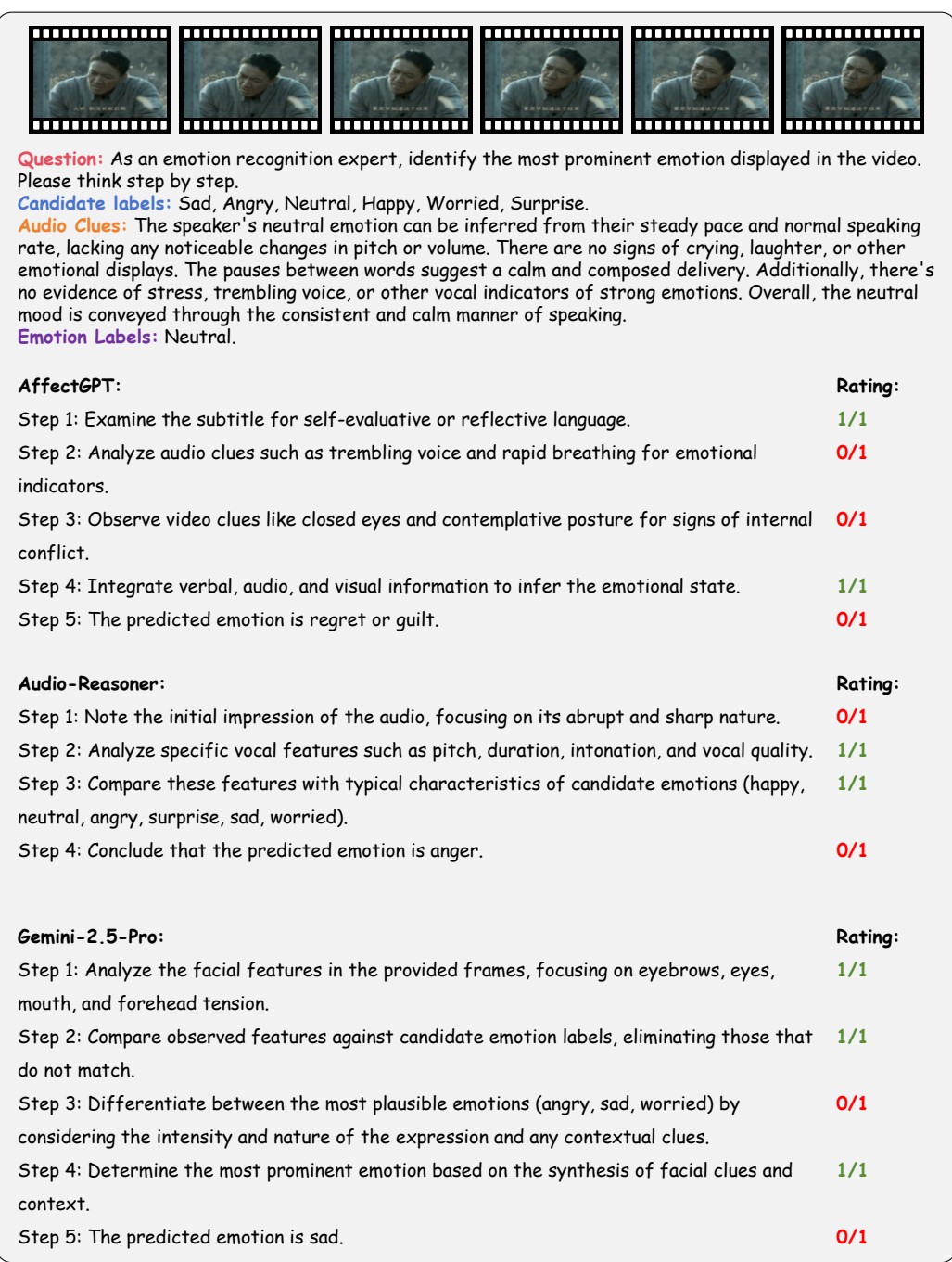

Figure 16: **Case Study on Noise-ER.** We showcase the answers and ratings of three MLLMs.

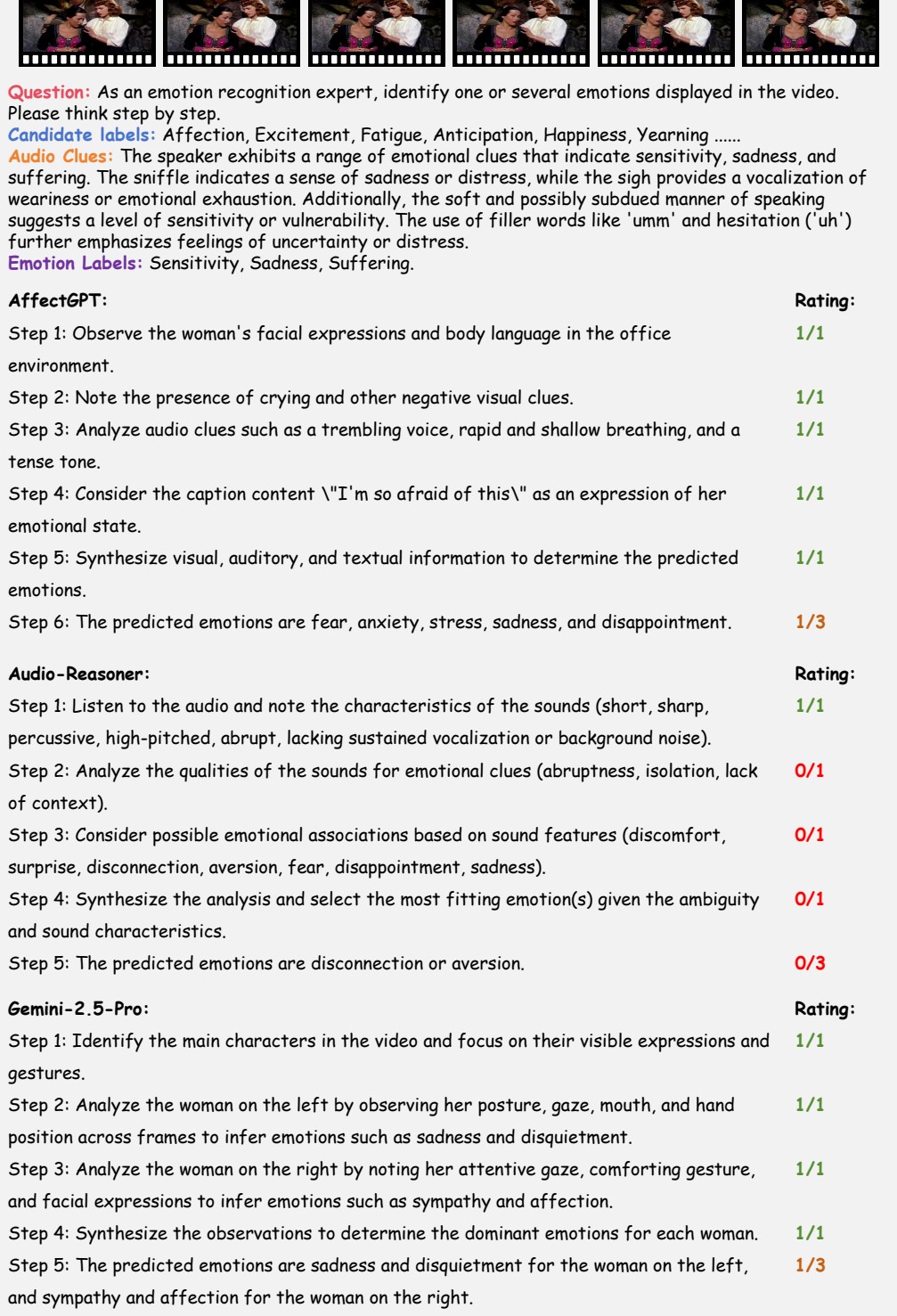

**Question:** As an emotion recognition expert, identify one or several emotions displayed in the video. Please think step by step.
**Candidate labels:** Affection, Excitement, Fatigue, Anticipation, Happiness, Yearning ......
**Audio Clues:** The speaker exhibits a range of emotional clues that indicate sensitivity, sadness, and suffering. The sniffle indicates a sense of sadness or distress, while the sigh provides a vocalization of weariness or emotional exhaustion. Additionally, the soft and possibly subdued manner of speaking suggests a level of sensitivity or vulnerability. The use of filler words like 'umm' and hesitation ('uh') further emphasizes feelings of uncertainty or distress.
**Emotion Labels:** Sensitivity, Sadness, Suffering.

| **AffectGPT:** | **Rating:** |
|---|---|
| Step 1: Observe the woman's facial expressions and body language in the office environment. | **1/1** |
| Step 2: Note the presence of crying and other negative visual clues. | **1/1** |
| Step 3: Analyze audio clues such as a trembling voice, rapid and shallow breathing, and a tense tone. | **1/1** |
| Step 4: Consider the caption content \"I'm so afraid of this\" as an expression of her emotional state. | **1/1** |
| Step 5: Synthesize visual, auditory, and textual information to determine the predicted emotions. | **1/1** |
| Step 6: The predicted emotions are fear, anxiety, stress, sadness, and disappointment. | **1/3** |

| **Audio-Reasoner:** | **Rating:** |
|---|---|
| Step 1: Listen to the audio and note the characteristics of the sounds (short, sharp, percussive, high-pitched, abrupt, lacking sustained vocalization or background noise). | **1/1** |
| Step 2: Analyze the qualities of the sounds for emotional clues (abruptness, isolation, lack of context). | **0/1** |
| Step 3: Consider possible emotional associations based on sound features (discomfort, surprise, disconnection, aversion, fear, disappointment, sadness). | **0/1** |
| Step 4: Synthesize the analysis and select the most fitting emotion(s) given the ambiguity and sound characteristics. | **0/1** |
| Step 5: The predicted emotions are disconnection or aversion. | **0/3** |

| **Gemini-2.5-Pro:** | **Rating:** |
|---|---|
| Step 1: Identify the main characters in the video and focus on their visible expressions and gestures. | **1/1** |
| Step 2: Analyze the woman on the left by observing her posture, gaze, mouth, and hand position across frames to infer emotions such as sadness and disquietment. | **1/1** |
| Step 3: Analyze the woman on the right by noting her attentive gaze, comforting gesture, and facial expressions to infer emotions such as sympathy and affection. | **1/1** |
| Step 4: Synthesize the observations to determine the dominant emotions for each woman. | **1/1** |
| Step 5: The predicted emotions are sadness and disquietment for the woman on the left, and sympathy and affection for the woman on the right. | **1/3** |

Figure 17: **Case Study on FG-ER.** We showcase the answers and ratings of three MLLMs.

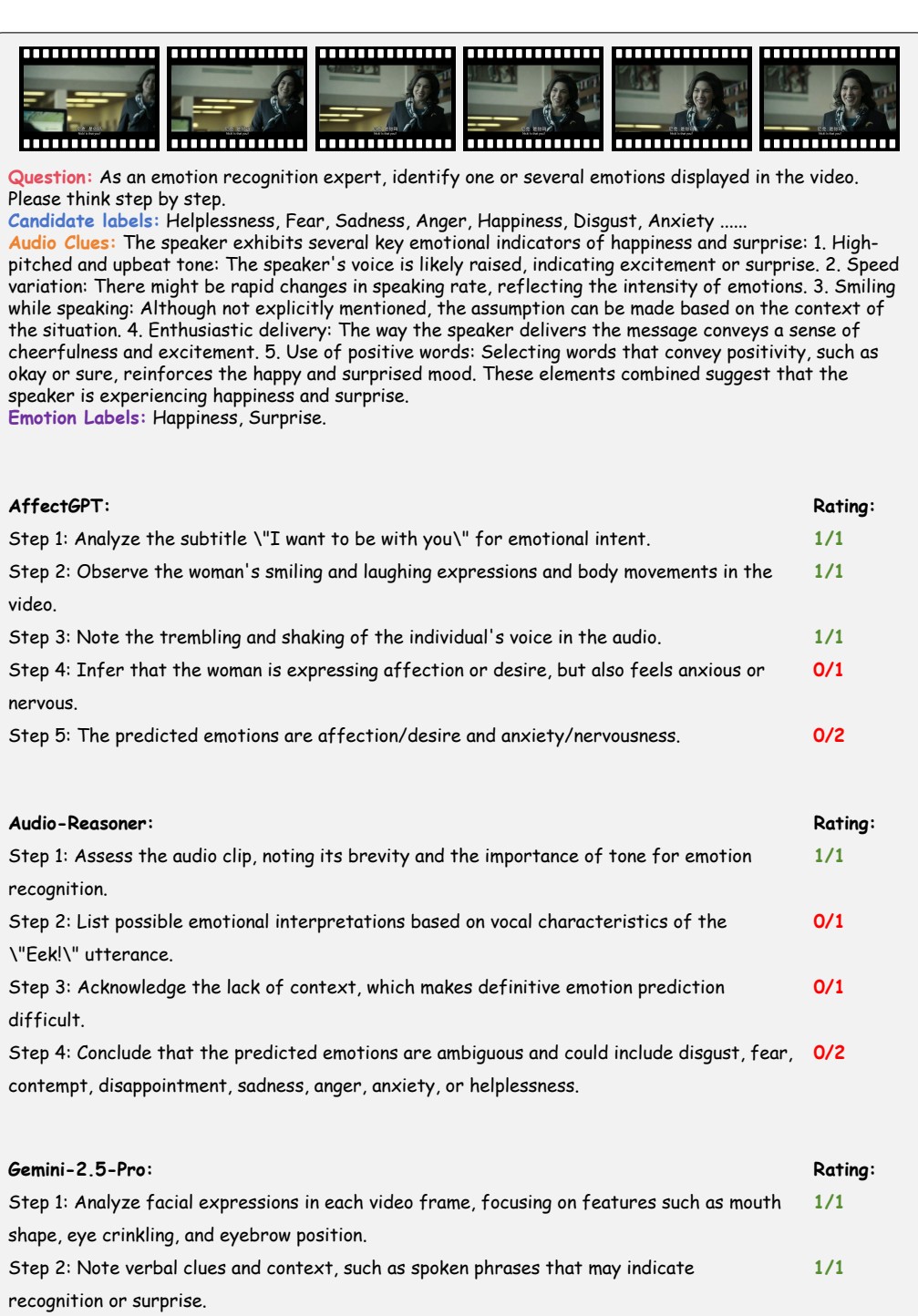

**Question:** As an emotion recognition expert, identify one or several emotions displayed in the video. Please think step by step.
**Candidate labels:** Helplessness, Fear, Sadness, Anger, Happiness, Disgust, Anxiety ......
**Audio Clues:** The speaker exhibits several key emotional indicators of happiness and surprise: 1. High-pitched and upbeat tone: The speaker's voice is likely raised, indicating excitement or surprise. 2. Speed variation: There might be rapid changes in speaking rate, reflecting the intensity of emotions. 3. Smiling while speaking: Although not explicitly mentioned, the assumption can be made based on the context of the situation. 4. Enthusiastic delivery: The way the speaker delivers the message conveys a sense of cheerfulness and excitement. 5. Use of positive words: Selecting words that convey positivity, such as okay or sure, reinforces the happy and surprised mood. These elements combined suggest that the speaker is experiencing happiness and surprise.
**Emotion Labels:** Happiness, Surprise.

| **AffectGPT:** | **Rating:** |
|---|---|
| Step 1: Analyze the subtitle \"I want to be with you\" for emotional intent. | **1/1** |
| Step 2: Observe the woman's smiling and laughing expressions and body movements in the video. | **1/1** |
| Step 3: Note the trembling and shaking of the individual's voice in the audio. | **1/1** |
| Step 4: Infer that the woman is expressing affection or desire, but also feels anxious or nervous. | **0/1** |
| Step 5: The predicted emotions are affection/desire and anxiety/nervousness. | **0/2** |

| **Audio-Reasoner:** | **Rating:** |
|---|---|
| Step 1: Assess the audio clip, noting its brevity and the importance of tone for emotion recognition. | **1/1** |
| Step 2: List possible emotional interpretations based on vocal characteristics of the \"Eek!\" utterance. | **0/1** |
| Step 3: Acknowledge the lack of context, which makes definitive emotion prediction difficult. | **0/1** |
| Step 4: Conclude that the predicted emotions are ambiguous and could include disgust, fear, contempt, disappointment, sadness, anger, anxiety, or helplessness. | **0/2** |

| **Gemini-2.5-Pro:** | **Rating:** |
|---|---|
| Step 1: Analyze facial expressions in each video frame, focusing on features such as mouth shape, eye crinkling, and eyebrow position. | **1/1** |
| Step 2: Note verbal clues and context, such as spoken phrases that may indicate recognition or surprise. | **1/1** |
| Step 3: Match observed facial and verbal clues to candidate emotion labels. | **1/1** |
| Step 4: Exclude emotion labels that do not fit the observed clues. | **1/1** |
| Step 5: The predicted emotions are happiness and surprise. | **2/2** |

Figure 18: **Case Study on ML-ER.** We showcase the answers and ratings of three MLLMs.

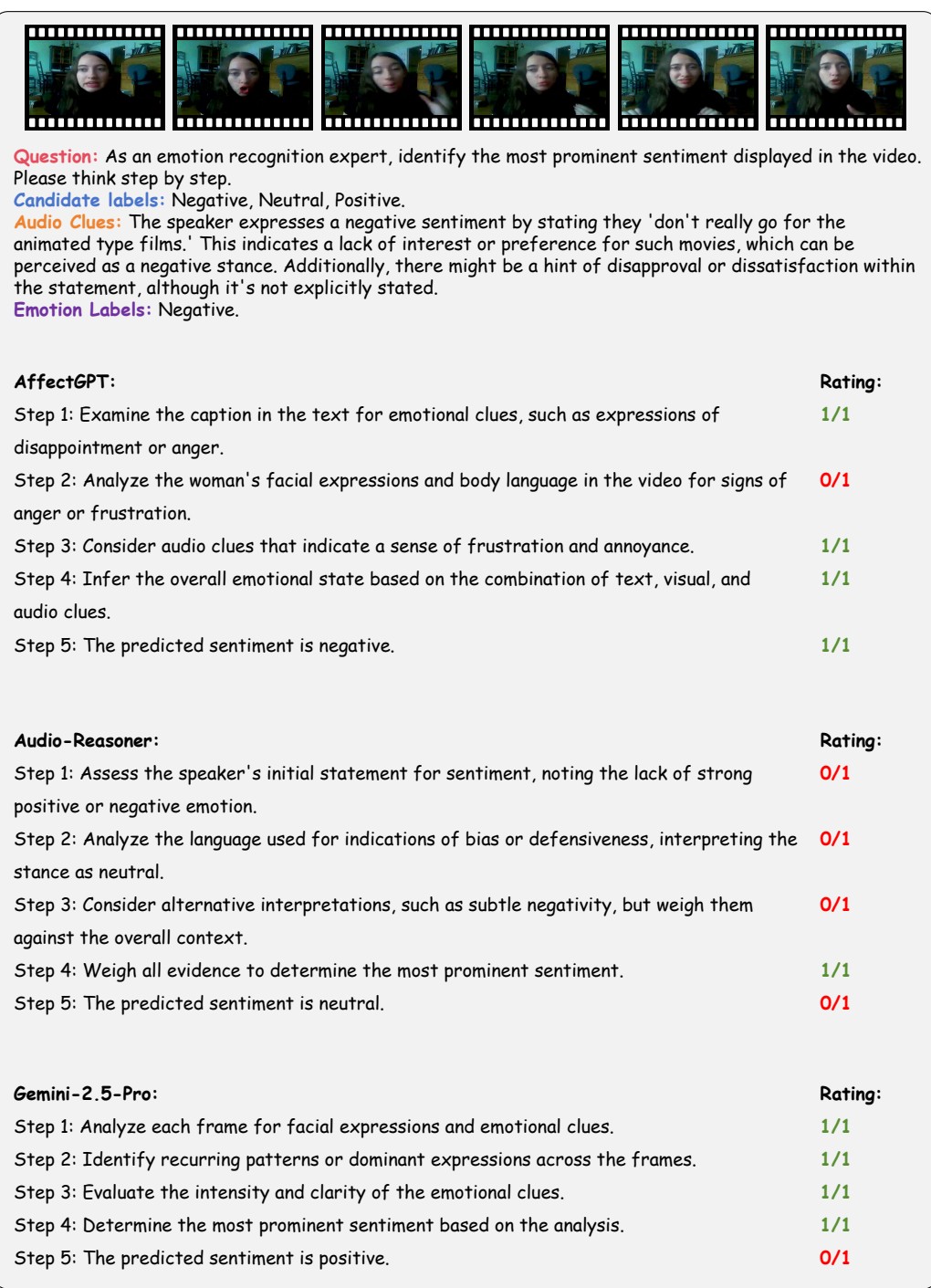

Figure 19: **Case Study on SA.** We showcase the answers and ratings of three MLLMs.

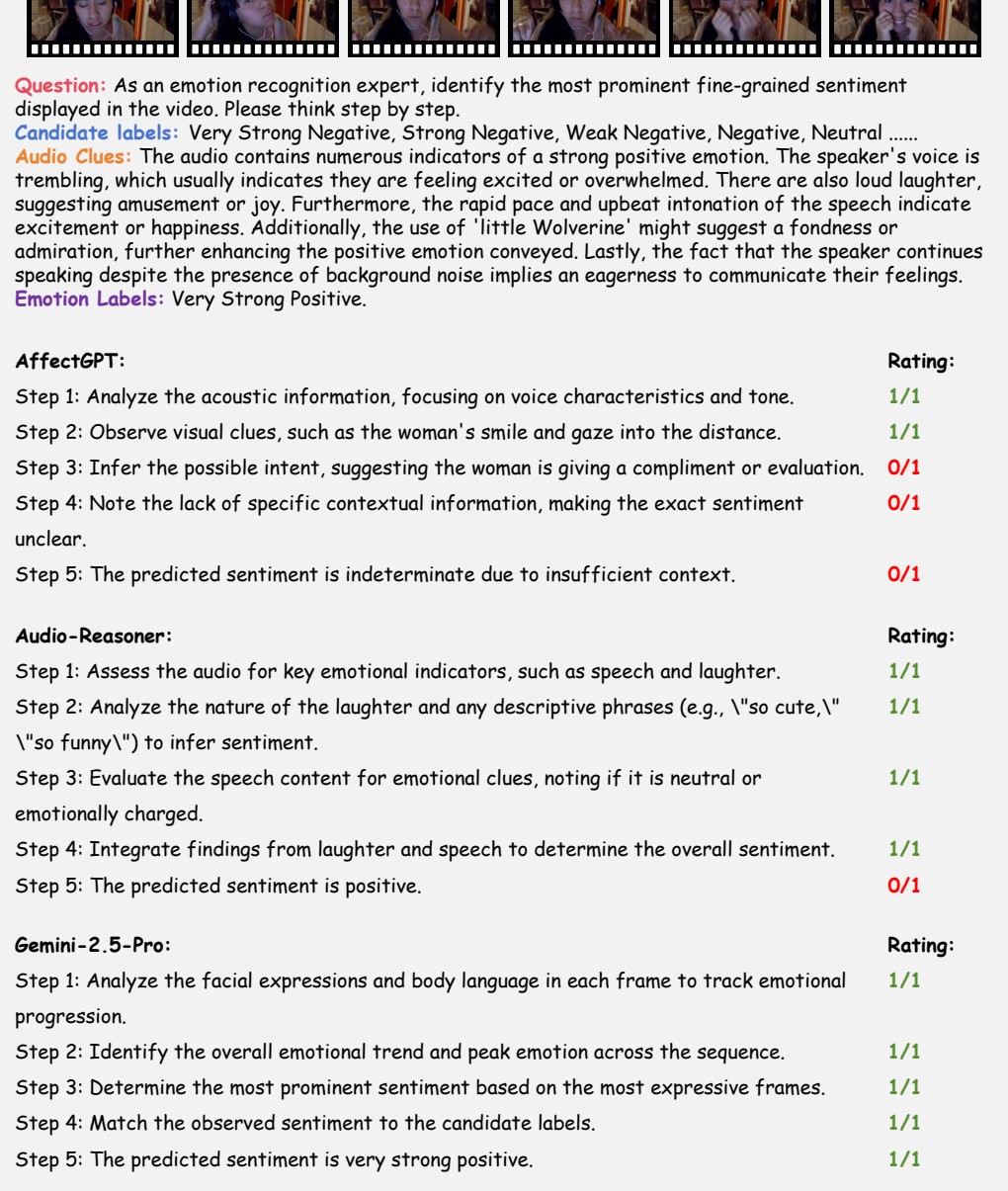

**Question:** As an emotion recognition expert, identify the most prominent fine-grained sentiment displayed in the video. Please think step by step.
**Candidate labels:** Very Strong Negative, Strong Negative, Weak Negative, Negative, Neutral ......
**Audio Clues:** The audio contains numerous indicators of a strong positive emotion. The speaker's voice is trembling, which usually indicates they are feeling excited or overwhelmed. There are also loud laughter, suggesting amusement or joy. Furthermore, the rapid pace and upbeat intonation of the speech indicate excitement or happiness. Additionally, the use of 'little Wolverine' might suggest a fondness or admiration, further enhancing the positive emotion conveyed. Lastly, the fact that the speaker continues speaking despite the presence of background noise implies an eagerness to communicate their feelings.
**Emotion Labels:** Very Strong Positive.

| **AffectGPT:** | **Rating:** |
|---|---|
| Step 1: Analyze the acoustic information, focusing on voice characteristics and tone. | **1/1** |
| Step 2: Observe visual clues, such as the woman's smile and gaze into the distance. | **1/1** |
| Step 3: Infer the possible intent, suggesting the woman is giving a compliment or evaluation. | **0/1** |
| Step 4: Note the lack of specific contextual information, making the exact sentiment unclear. | **0/1** |
| Step 5: The predicted sentiment is indeterminate due to insufficient context. | **0/1** |

| **Audio-Reasoner:** | **Rating:** |
|---|---|
| Step 1: Assess the audio for key emotional indicators, such as speech and laughter. | **1/1** |
| Step 2: Analyze the nature of the laughter and any descriptive phrases (e.g., \"so cute,\" \"so funny\") to infer sentiment. | **1/1** |
| Step 3: Evaluate the speech content for emotional clues, noting if it is neutral or emotionally charged. | **1/1** |
| Step 4: Integrate findings from laughter and speech to determine the overall sentiment. | **1/1** |
| Step 5: The predicted sentiment is positive. | **0/1** |

| **Gemini-2.5-Pro:** | **Rating:** |
|---|---|
| Step 1: Analyze the facial expressions and body language in each frame to track emotional progression. | **1/1** |
| Step 2: Identify the overall emotional trend and peak emotion across the sequence. | **1/1** |
| Step 3: Determine the most prominent sentiment based on the most expressive frames. | **1/1** |
| Step 4: Match the observed sentiment to the candidate labels. | **1/1** |
| Step 5: The predicted sentiment is very strong positive. | **1/1** |

Figure 20: **Case Study on FG-SA.** We showcase the answers and ratings of three MLLMs.

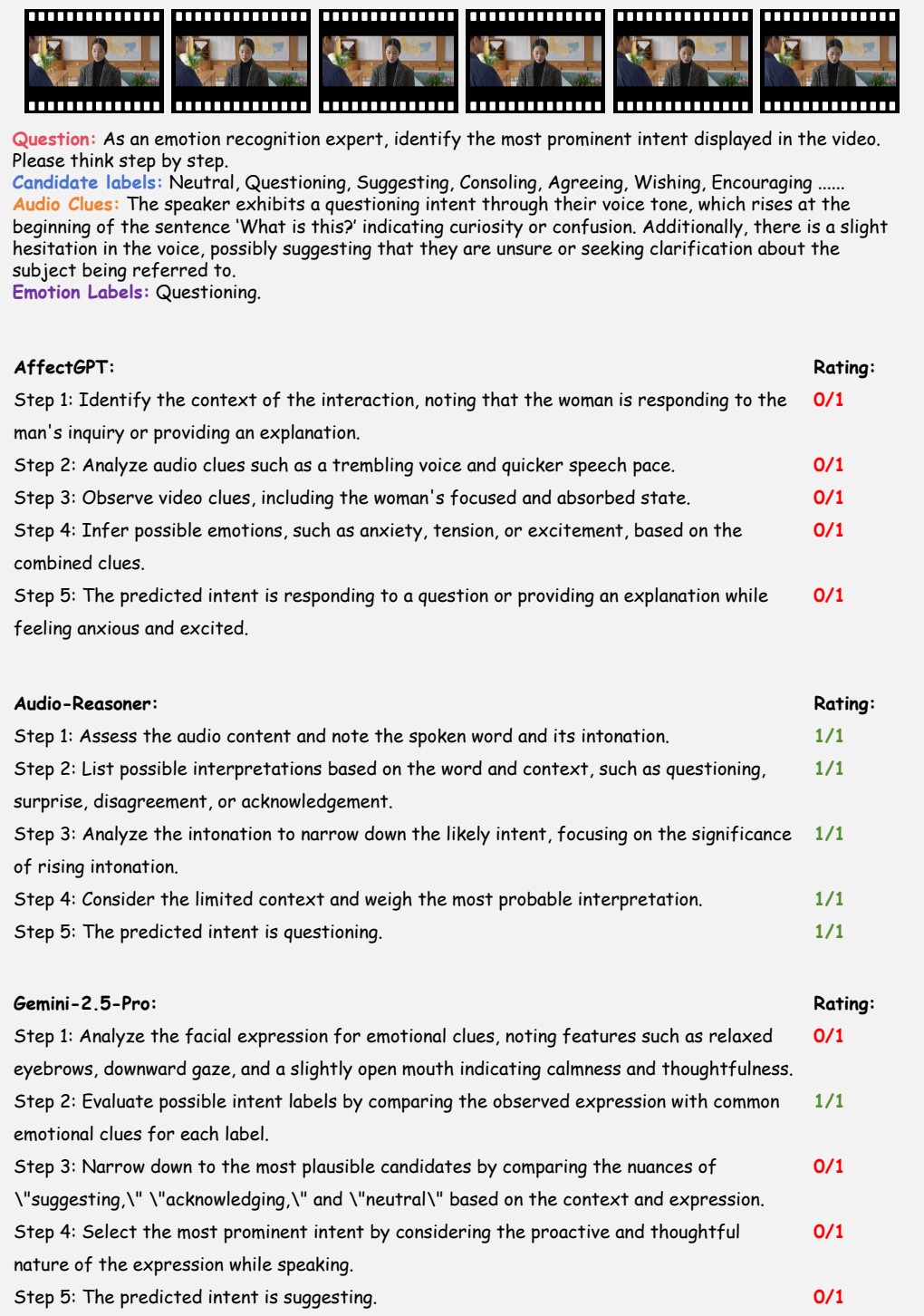

Figure 21: **Case Study on IR.** We showcase the answers and ratings of three MLLMs.

