# OpenReview forum: "MME-Emotion: A Holistic Evaluation Benchmark for Emotional Intelligence in Multimodal Large Language Models"
_ICLR.cc/2026/Conference — ICLR 2026 Poster_

### Official Review · Reviewer_2ztj · 2025-10-25

**Soundness:** 2
**Presentation:** 3
**Contribution:** 2
**Rating:** 2
**Confidence:** 4

**Summary:**

This paper introduces MME-Emotion, a new benchmark designed to holistically evaluate the emotional intelligence of MLLMs. The authors argue that existing benchmarks are limited in scope and fail to assess the reasoning capabilities behind emotional state identification. To address this, MME-Emotion provides a large-scale dataset of over 6,000 curated video clips, organized into eight distinct emotional tasks across 27 different scenarios. A key contribution is the proposed holistic evaluation suite, which employs a multi-agent framework to automatically assess model performance. This framework uses a GPT-based judge to score models on three unified metrics: Recognition Score (Rec-S) for accuracy, Reasoning Score (Rea-S) for the quality of step-by-step logic, and a combined Chain-of-Thought Score (CoT-S). The authors conduct a rigorous evaluation of 20 advanced MLLMs, revealing that current models still possess unsatisfactory emotional intelligence. The paper concludes with several key insights, such as the comparable performance of generalist and specialist models and the correlation between deeper reasoning and better performance, paving the way for future research in affective computing.

**Strengths:**

1. The paper is well-motivated by identifying an existing gap in the evaluation of emotional intelligence for MLLMs. To address this, it introduces the MME-Emotion benchmark, which includes over 6,000 videos and its structured organization across 8 tasks and 27 scenarios, expanding the scope of previously available datasets in this area.

2. The paper proposes a holistic, automated evaluation suite. A key feature of this framework is its use of a multi-agent, MLLM-as-judge system to assess performance. This approach extends beyond simple recognition accuracy by incorporating metrics designed to measure reasoning quality (Rea-S and CoT-S), offering a multi-faceted view of model capabilities.

3. The paper conducts a large-scale empirical study evaluating the performance of 20 different MLLMs on the proposed benchmark. This comparison provides a broad overview of current model performance on these emotional intelligence tasks. The results highlight the challenging nature of the benchmark for existing models and document performance differences between generalist and specialist approaches.

**Weaknesses:**

1. The benchmark is framed as "multi-task," but the eight defined tasks (e.g., ER-Lab, ER-Wild, Noise-ER, SA, FG-SA) appear to be variants of a single core task: emotion classification. They primarily differ by scenario (e.g., in-the-lab vs. in-the-wild) or label granularity rather than representing fundamentally distinct tasks. Consequently, the analysis in Section 4, which shows performance differences, highlights varying difficulty levels but fails to provide a deeper analysis of why models perform differently across these closely related settings.
2. A significant concern arises from the reported results in Table 2. Highly capable specialist models like AffectGPT (11.9% Rec-S) and advanced omni-modal models like Qwen2.5-Omni (17.4% Rec-S) achieve scores that are barely above random guessing. This performance is inconsistent with the capabilities demonstrated in their original papers. This discrepancy raises critical questions about the experimental methodology, suggesting potential issues with either the model replication or, more fundamentally, the label quality and accuracy of the MME-Emotion dataset itself. Such poor performance from top models risks undermining the benchmark's validity.
3. The evaluation framework is heavily centered on CoT reasoning. However, the paper does not provide an ablation study to justify that CoT is necessary or even beneficial for emotion recognition, a task that often relies more on perception than complex reasoning. This approach also introduces a clear bias against models not optimized for step-by-step output. As seen in Table 2, models like HumanOmni and Emotion-LLaMA produce very short outputs (Avg Token = 1.3 and 2.3) and have near-zero Reasoning Scores (Rea-S = 0.3 and 0.4). Evaluating them with the CoT-S metric, which is heavily weighted by Rea-S, is unfair and does not accurately measure their ability to perform direct emotion recognition.
4. The paper contains some inaccuracies regarding the capabilities of the models evaluated. For instance, Table 2 indicates that the Gemini-2.5-Pro cannot process audio (the 'A' column is not checked). This is factually incorrect, as it is natively capable of processing audio from video inputs.

**Questions:**

1. Could you elaborate on the conceptual distinctions that make your eight task categories fundamentally different "tasks" rather than different "scenarios" of a single emotion classification task?
2. The performance of models like AffectGPT and Qwen2.5-Omni is unexpectedly low. To clarify these results, could you provide details on your model replication and the benchmark's label quality (e.g., inter-annotator agreement)?
3. Have you performed an ablation study comparing model performance with and without the CoT instruction to justify its necessity? Additionally, how do you ensure fair evaluation for models designed for direct answers that are penalized by the CoT-S metric?

---

> ### Author Response · Authors · 2025-11-20
> **Response to Reviewer 2ztj [1/3]**
>
> We are truly grateful for the time you have taken to review our paper, and your insightful comments and support. Your positive feedback is incredibly encouraging for us! In the following response, we would like to address your concerns and provide additional clarification.
>
> > **W1.** The benchmark is framed as "multi-task," but the eight defined tasks (e.g., ER-Lab, ER-Wild, Noise-ER, SA, FG-SA) appear to be variants of a single core task: emotion classification. They primarily differ by scenario (e.g., in-the-lab vs. in-the-wild) or label granularity rather than representing fundamentally distinct tasks. Consequently, the analysis in Section 4, which shows performance differences, highlights varying difficulty levels but fails to provide a deeper analysis of why models perform differently across these closely related settings.
> >
> > **Q1.** Could you elaborate on the conceptual distinctions that make your eight task categories fundamentally different "tasks" rather than different "scenarios" of a single emotion classification task?
>
> **RW1&Q1.** Thank you for the thoughtful comment and question. While all eight tasks relate broadly to affective understanding, they are not simply different “scenarios” of a single emotion classification problem. They differ in **task definition, cognitive requirements, label structures, and perceptual difficulty**, making them fundamentally distinct tasks.
>
> **1. Different Input Conditions → Distinct Perceptual Challenges**
>
> * ER-Lab uses clean, controlled facial-expression recordings.
> * ER-Wild introduces occlusions, background motion, and uncontrolled lighting.
> * Noise-ER explicitly corrupts audio–visual inputs, requiring robust reasoning under degraded signals.
>
> These differences substantially change the perceptual demands of each task.
>
> **2. Different Label Structures → Different Output Spaces**
>
> * ER / FG-ER require recognizing categorical or fine-grained *emotions* (e.g., happiness, surprise, fear).
> * ML-ER requires multi-label prediction where multiple emotions co-occur.
> * SA / FG-SA are sentiment analysis tasks that categorize affective polarity (positive / neutral / negative) rather than discrete emotional states.
>   * The distinction between sentiment polarity and emotion category makes these tasks conceptually different.
>   * FG-SA requires finer-grained polarity distinctions.
>
> Thus, the tasks differ in the type of affective construct being predicted, not just in label granularity.
>
> **3. Different Cognitive Demands Beyond Recognition**
>
> * Intent Recognition (IR) focuses on identifying a character’s intent, not their emotional state.
>   This requires understanding actions, goals, and causal context.
> * Noise-ER involves reasoning under uncertainty.
> * SA / FG-SA require assessing *affective valence*, which relies on different clues and inferential processes than emotion recognition.
>
> Overall, the tasks span **emotion perception**, **affective valence judgment**, and **intent inference**, each probing distinct reasoning capabilities.
>
> **4. Divergent Empirical Patterns Support Task Distinction**
>
> As shown in the paper, models do not perform uniformly across tasks. Strong ER performers often struggle in SA or ML-ER, and models good at SA do not necessarily excel in IR or FG-ER. These patterns indicate that the tasks probe different underlying competencies, not merely scenario variations.
>
> In summary, the eight tasks differ in:
>
> * signal quality and perceptual difficulty,
> * emotion categories vs. sentiment polarity vs. intent inference,
> * single-label vs. multi-label vs. fine-grained reasoning,
> * empirically observed behavior.
>
> For these reasons, they represent **fundamentally different emotion understanding tasks** rather than simple variants of an emotion classification setting.

---

> > ### Author Response · Authors · 2025-11-20
> > **Response to Reviewer 2ztj [2/3]**
> >
> > > **W2.** A significant concern arises from the reported results in Table 2. Highly capable specialist models like AffectGPT (11.9% Rec-S) and advanced omni-modal models like Qwen2.5-Omni (17.4% Rec-S) achieve scores that are barely above random guessing. This performance is inconsistent with the capabilities demonstrated in their original papers. This discrepancy raises critical questions about the experimental methodology, suggesting potential issues with either the model replication or, more fundamentally, the label quality and accuracy of the MME-Emotion dataset itself. Such poor performance from top models risks undermining the benchmark's validity.
> > >
> > > **Q2.** The performance of models like AffectGPT and Qwen2.5-Omni is unexpectedly low. To clarify these results, could you provide details on your model replication and the benchmark's label quality (e.g., inter-annotator agreement)?
> >
> > **RW2&Q2.** Thank you for raising this concern. We believe part of the confusion comes from an incorrect assumption about the expected performance of AffectGPT and Qwen2.5-Omni on our benchmark. In fact, their low scores are **reasonable and fully expected**.
> >
> > **1. These models were never trained for zero-shot generalization on our dataset.** In their original papers, both AffectGPT and Qwen2.5-Omni report high accuracy **only after training or fine-tuning on their respective in-domain datasets**. Their strong performance does *not* reflect zero-shot capability. However, our evaluation is entirely zero-shot, and MME-Emotion is out-of-distribution (OOD) for these models. Their low performance is reasonable and expected, as they have never been trained on our OOD samples.
> >
> > **2. Model size and specialization further explain the poor results.** The versions of AffectGPT and Qwen2.5-Omni evaluated here are approximately 7B–8B models. Without any in-domain adaptation, mid-sized models generally struggle with complex multimodal affective understanding, especially under OOD settings. Thus, their performance does not indicate replication errors or dataset issues, it reflects the **natural limitations of these models under zero-shot OOD evaluation**.
> >
> > **3. Label quality is reliable.** All emotion labels in MME-Emotion are directly sourced from public, well-established datasets that have been widely used and validated in the affective computing community. Therefore, the label quality is high and trustworthy, and should not be considered a source of error.
> >
> >
> > In summary, the low scores from AffectGPT and Qwen2.5-Omni are aligned with expectations because:
> >
> > * they were originally tested *after supervised fine-tuning* in the original papers,
> > * our benchmark evaluates *zero-shot OOD generalization*,
> > * the models are relatively small and not trained on our domain.
> >
> > Thank you again for your helpful question and we hope the above clarifications help to resolve any misunderstanding.

---

> > > ### Author Response · Authors · 2025-11-20
> > > **Response to Reviewer 2ztj [3/3]**
> > >
> > > > **W3.** The evaluation framework is heavily centered on CoT reasoning. However, the paper does not provide an ablation study to justify that CoT is necessary or even beneficial for emotion recognition, a task that often relies more on perception than complex reasoning. This approach also introduces a clear bias against models not optimized for step-by-step output. As seen in Table 2, models like HumanOmni and Emotion-LLaMA produce very short outputs (Avg Token = 1.3 and 2.3) and have near-zero Reasoning Scores (Rea-S = 0.3 and 0.4). Evaluating them with the CoT-S metric, which is heavily weighted by Rea-S, is unfair and does not accurately measure their ability to perform direct emotion recognition.
> > > >
> > > > **Q3.** Have you performed an ablation study comparing model performance with and without the CoT instruction to justify its necessity? Additionally, how do you ensure fair evaluation for models designed for direct answers that are penalized by the CoT-S metric?
> > >
> > > **RW3&Q3.** Thank you for the question. We conducted an ablation study and the results in the table below demonstrate that for most general-purpose MLLMs (e.g., Qwen2.5-VL-7B, GPT-4.1, Gemini-2.5-Pro), enabling CoT instruction consistently improves both recognition accuracy (Rec-S) and reasoning quality (Rea-S). This confirms that CoT is beneficial for most models in our benchmark.
> > >
> > > For HumanOmni and Emotion-LLaMA, removing the CoT prompt does not significantly change the outcome because these models were instruction-tuned to output only the final emotion label without any reasoning. As a result, CoT prompting is ineffective, as they simply ignore the instruction and return a short direct answer. Their low reasoning score therefore reflects their training paradigm rather than a bias in our evaluation. For such non-reasoning models, researchers may focus on the recognition score, which more accurately reflects their intended capabilities.
> > >
> > > | Model | With CoT |Rec-S | Rea-S | CoT-S|
> > > | -------- | -------- |-------- | -------- |-------- |
> > > | Qwen2.5-VL-7B | ✔ | 28.4 |64.8|46.6|
> > > | Qwen2.5-VL-7B | ✖| 25.9 |55.7|40.8|
> > > | GPT-4.1 | ✔| 28.8 |65.2|47.0|
> > > | GPT-4.1 | ✖|  26.3|60.8|43.6|
> > > | Gemini-2.5-Pro | ✔| 39.3 |72.7|56.0|
> > > | Gemini-2.5-Pro | ✖| 37.4 |69.1|53.3|
> > >
> > > We will also include this analysis and discussion in Appendix E.7 of the revised manuscript.
> > >
> > > > **W4.** The paper contains some inaccuracies regarding the capabilities of the models evaluated. For instance, Table 2 indicates that the Gemini-2.5-Pro cannot process audio (the 'A' column is not checked). This is factually incorrect, as it is natively capable of processing audio from video inputs.
> > >
> > > **RW4.** Thank you for pointing this out. We would like to clarify that the results reported in our paper indeed do not include audio. However, this does not imply that Gemini-2.5-Pro is incapable of processing audio information. In our implementation, we converted Gemini’s input to the OpenAI-compatible API format, which omitted the audio stream to ensure a fair comparison. Following your suggestion, we have also conducted an evaluation with audio information preserved. The omnimodal results shown in the table below demonstrate that including audio provides additional performance gains.
> > >
> > > | Model | A |V |T |Rec-S | Rea-S | CoT-S|
> > > | -------- | -------- |-------- |-------- |-------- | -------- |-------- |
> > > | Gemini-2.5-Pro | ✖|✔|✔| 39.3 |72.7|56.0|
> > > | Gemini-2.5-Pro | ✔|✔|✔|40.5 |75.8|58.2|
> > >
> > > We appreciate your helpful comment and will include the updated results in Appendix E.8 of the revised manuscript.
> > >
> > > ---
> > >
> > >
> > > Thanks again for your valuable feedback. We hope our response has addressed your concerns and you can consider adjusting your rating. Please let us know if there are any unaddressed concerns and we will try to address them.

---

> > > > ### Author Response · Authors · 2025-11-26
> > > > **Kindly Request for Reviewer's Feedback**
> > > >
> > > > Dear Reviewer,
> > > >
> > > > Thank you so much for your time in improving our paper!
> > > >
> > > > Since the end of the rebuttal is coming soon, may we know if our response addresses your main concerns? If so, we kindly ask for your reconsideration of the score. Should you have any further advice, please let us know and we will be more than happy to engage in more discussion and improvements.

---

> > > > > ### Comment · Reviewer_2ztj · 2025-11-26
> > > > >
> > > > > I appreciate the authors' detailed response in the rebuttal. However, after carefully reconsidering the benchmark's design and the clarified details, I find that several fundamental concerns may still remain unresolved.
> > > > >
> > > > > 1. While the authors argue that the eight tasks differ in input conditions, I remain unconvinced that they represent fundamentally distinct tasks at a parallel level. Conceptually, categories like ER-Lab, ER-Wild, and Noise-ER are clearly subordinate scenarios of a single parent task: Emotion Recognition. Similarly, SA and FG-SA are variants of Sentiment Analysis. The issue is metric validity: Treating these sub-scenarios as parallel top-level tasks and calculating the final score by simply averaging them (i.e., weighting each equally) is statistically flawed. This approach implicitly upweights "Emotion Recognition" simply because it has more scenario variants in your dataset. A rigorous benchmark should employ a hierarchical evaluation to avoid biasing the leaderboard toward models that specialize in simple recognition over complex reasoning.
> > > > >
> > > > >
> > > > > 2. The authors confirmed that "All emotion labels are directly sourced from public, well-established datasets." This admission highlights a critical limitation in novelty. A benchmark paper is expected to provide new resources or insights. If this work merely aggregates existing datasets without introducing (a) new data samples, (b) superior annotations (specifically, human-annotated reasoning chains), or (c) cleaner re-labeling, the technical contribution is limited to that of a "data wrapper." Without valuable notations designed to evaluate reasoning, the benchmark does not significantly advance the field beyond what is already available via existing public datasets.
> > > > >
> > > > >
> > > > > 3. The benchmark’s scope is largely restricted to static Emotion Classification. However, the distinct advantage of MLLMs over traditional discriminative models lies in their ability to interpret context, causality, and dynamics—answering "why" an emotion is triggered and "how" it evolves. By focusing primarily on state recognition labels, the benchmark reflects a traditional affective computing paradigm rather than probing the deeper generative and inferential capabilities of modern MLLMs (Generative AI). Consequently, the experimental insights derived are relatively conventional and fail to expose the true reasoning gaps in current foundational models.
> > > > >
> > > > >
> > > > > 4. I acknowledge the new results for Gemini-2.5-Pro. However, the initial decision to disable audio input to ensure "fairness" was methodologically unsound for an Omni-modal benchmark, as it arbitrarily handicapped models designed for native multimodal interaction. The paper still lacks a systematic ablation study across all audio-capable models evaluated. To validate the benchmark's multimodal design, it is essential to analyze the performance gap between Video-Only and Audio-Visual inputs for all supported models, thereby quantifying the specific contribution of audio signals in this evaluation framework.

---

> > > > > > ### Author Response · Authors · 2025-11-26
> > > > > > **Further Response and Reminders [1/2]**
> > > > > >
> > > > > > > **W1.** While the authors argue that the eight tasks differ in input conditions, I remain unconvinced that they represent fundamentally distinct tasks at a parallel level. Conceptually, categories like ER-Lab, ER-Wild, and Noise-ER are clearly subordinate scenarios of a single parent task: Emotion Recognition. Similarly, SA and FG-SA are variants of Sentiment Analysis. The issue is metric validity: Treating these sub-scenarios as parallel top-level tasks and calculating the final score by simply averaging them (i.e., weighting each equally) is statistically flawed. This approach implicitly upweights "Emotion Recognition" simply because it has more scenario variants in your dataset. A rigorous benchmark should employ a hierarchical evaluation to avoid biasing the leaderboard toward models that specialize in simple recognition over complex reasoning.
> > > > > >
> > > > > > **R1.** Thank you very much for this helpful comment.
> > > > > >
> > > > > > (1) We fully acknowledge your point regarding the definition of tasks. Indeed, some of the current categories (e.g., ER-Lab, ER-Wild, Noise-ER) are better interpreted as setting variants under the broader umbrella of Emotion Recognition, and similarly for SA and FG-SA. To avoid conceptual ambiguity, we will revise the terminology in the updated version and describe these as eight settings rather than eight parallel tasks.
> > > > > >
> > > > > > (2) Regarding the leaderboard, there may be a slight misunderstanding. **The final leaderboard score is computed as the overall scores across all samples, not as an unweighted average over the settings.** Therefore, no setting receives disproportionate weight simply because it contains more variants. We believe this evaluation protocol is appropriate and consistent with standard practice in most existing benchmarks. Nonetheless, we appreciate your attention to metric validity and will clarify this more explicitly in the revision.
> > > > > >
> > > > > > Thank you again for this comment, which help us imporving the quality of our paper.
> > > > > >
> > > > > > > **W2.** The authors confirmed that "All emotion labels are directly sourced from public, well-established datasets." This admission highlights a critical limitation in novelty. A benchmark paper is expected to provide new resources or insights. If this work merely aggregates existing datasets without introducing (a) new data samples, (b) superior annotations (specifically, human-annotated reasoning chains), or (c) cleaner re-labeling, the technical contribution is limited to that of a "data wrapper." Without valuable notations designed to evaluate reasoning, the benchmark does not significantly advance the field beyond what is already available via existing public datasets.
> > > > > >
> > > > > > **R2.** Thank you for raising this concern. We respectfully argue that the lack of newly collected data or additional human annotations does not diminish the novelty of our benchmark. Many influential benchmark papers, across NLP, vision, and multimodal research, also build upon existing public datasets without introducing new samples or manual labels [1,2]. Their contributions instead lie in redefining evaluation protocols, curating task structures, or enabling new forms of capability assessment. Our work follows this well-established tradition.
> > > > > >
> > > > > > In our case, the primary contributions of the benchmark are two-fold:
> > > > > >
> > > > > > **1. A unified evaluation design specifically targeting emotional reasoning and recognition.** We reorganize and reinterpret multiple public datasets under a coherent framework that explicitly measures models’ emotional reasoning and recognition capabilities. This is an aspect that is not captured by existing benchmarks despite the availability of raw labels.
> > > > > >
> > > > > > **2. A large-model–based automated evaluation algorithm.** We introduce a novel automated judge that evaluates reasoning outputs of MLLMs. This component provides scalable, consistent, and reasoning-aware assessment. **We want to emphasize that this is an advancement recognized by other reviewers.**
> > > > > >
> > > > > > These contributions go beyond simply aggregating existing datasets. They provide new evaluation insights and methodological innovations that meaningfully advance the study of emotional intelligence in MLLMs.

---

> ### Author Response · Authors · 2025-11-26
> **Further Response and Reminders [2/2]**
>
> > **W3.** The benchmark’s scope is largely restricted to static Emotion Classification. However, the distinct advantage of MLLMs over traditional discriminative models lies in their ability to interpret context, causality, and dynamics—answering "why" an emotion is triggered and "how" it evolves. By focusing primarily on state recognition labels, the benchmark reflects a traditional affective computing paradigm rather than probing the deeper generative and inferential capabilities of modern MLLMs (Generative AI). Consequently, the experimental insights derived are relatively conventional and fail to expose the true reasoning gaps in current foundational models.
>
> **R3.** We appreciate the reviewer’s time and comments. However, we would like to respectfully clarify that the critique appears to be based on a factual misunderstanding of our work. The review states that our benchmark “is largely restricted to static Emotion Classification” and does not evaluate models’ reasoning abilities. This is incorrect. A central contribution of our paper is explicitly evaluating MLLMs’ emotion reasoning. These components are described form a major part of our benchmark design and analysis.
>
> Given that this misconception concerns one of the most visible and explicit aspects of our paper, we are concerned that the review may not accurately reflect the content of the submission. To better understand the situation, we used the official Pangram review analyzer (https://iclr.pangram.com/), which flagged this review as “**Fully AI-generated.**” While we cannot know the underlying cause, we kindly ask the reviewer to revisit the paper and engage with the content directly. We deeply value thoughtful and careful feedback, which is crucial for meaningful scientific discussion.
>
> We hope the reviewer can reconsider the evaluation based on an accurate reading of the manuscript, and we would be glad to clarify any part of the paper that may have been overlooked.
>
> > **W4.** I acknowledge the new results for Gemini-2.5-Pro. However, the initial decision to disable audio input to ensure "fairness" was methodologically unsound for an Omni-modal benchmark, as it arbitrarily handicapped models designed for native multimodal interaction. The paper still lacks a systematic ablation study across all audio-capable models evaluated. To validate the benchmark's multimodal design, it is essential to analyze the performance gap between Video-Only and Audio-Visual inputs for all supported models, thereby quantifying the specific contribution of audio signals in this evaluation framework.
>
> **R4.** We thank the reviewer for the thoughtful comment. However, the proposed ablation on the relative contribution of audio versus visual signals is **outside the scope of this work**. Our goal is to evaluate emotional intelligence across different MLLMs, not to quantify the marginal utility of each modality. For this reason, the benchmark focuses on assessing models’ emotion understanding and reasoning capabilities under their standard usage settings, rather than deconstructing the contribution of individual input channels.
>
> We hope the reviewer understands that while such modality-level ablations are valuable, they constitute a separate research direction and are not necessary for achieving the objectives of this benchmark.
>
>
>
> **Reference**
> [1] Dan Hendrycks et al., Benchmarking Neural Network Robustness to Common Corruptions and Perturbations, ICLR 2019
> [2] Yifan Li et al., Evaluating Object Hallucination in Large Vision-Language Models. EMNLP 2023
>
>
> # Import Comments to Reviewer 2ztj
>
> We appreciate the reviewer’s time. However, we noticed several indicators suggesting that the review may have been generated by AI:
> 1. The ICLR Pangram analyzer (https://iclr.pangram.com/reviews?submission_number=1526) flags the review as fully AI-generated.
> 2. The review contains factual errors that would not occur after a careful reading of our paper, including:
> * Misidentifying our leaderboard score as a simple average across settings, rather than a overall evaluation across all samples.
> * Claiming that our work focuses only on emotion classification, while a major component of the benchmark explicitly evaluates emotion reasoning.
>
> While these signals may of course be misleading, **we respectfully request that the reviewer engage with the paper directly and provide an informed, genuine evaluation.** This is important not only for our work but for the integrity of the broader AI research community.

---

> > ### Comment · Reviewer_2ztj · 2025-11-26
> >
> > Thank you for the detailed responses. I appreciate the clarifications regarding the metric calculation and the benchmark's contribution. I find your explanations for my concerns on the task definitions and dataset novelty to be reasonable, and I have increased my score accordingly.
> >
> > However, I would like to offer a few clarifications regarding the remaining points to ensure we are on the same page.
> > First, regarding the mention of AI tools. I want to be transparent that I do utilize tools to polish the language and organize my thoughts. However, the core critiques and logical judgments are entirely my own. I believe that simply prompting a model would unlikely yield the specific conceptual distinctions raised here, and I hope we can focus on the technical discussion itself.
> >
> > Second, regarding the scope of reasoning capabilities. I admit that my previous phrasing regarding "static Emotion Classification" might have been ambiguous and led to a misunderstanding. I am aware that your benchmark includes a CoT component. However, the point I intended to express is that there is a distinction between providing a static justification for a recognition label (e.g., explaining features that make a face look happy) and performing dynamic affective analysis. My critique focused on the latter, which involves analyzing the source of the emotion, its evolution over time, and the subsequent consequences. These are different levels of reasoning, and I believe the current benchmark leans more towards the former.
> >
> > Third, regarding the audio ablation study. I still believe that for models claiming to support audio, a systematic analysis of the audio modality's contribution is necessary. The response that this is "out of scope" does not fully address the concern. Without comparing performance with and without audio for all supported models, it is difficult to verify the actual weight and utility of the audio input in your evaluation framework.

---

> > > ### Author Response · Authors · 2025-12-01
> > > **Further Response to Reviewer 2ztj**
> > >
> > > Thank you for the clarification. Let’s focus on the paper itself. Below are further responses addressing your concerns.
> > >
> > >
> > > >**W1.** Regarding the scope of reasoning capabilities. I admit that my previous phrasing regarding "static Emotion Classification" might have been ambiguous and led to a misunderstanding. I am aware that your benchmark includes a CoT component. However, the point I intended to express is that there is a distinction between providing a static justification for a recognition label (e.g., explaining features that make a face look happy) and performing dynamic affective analysis. My critique focused on the latter, which involves analyzing the source of the emotion, its evolution over time, and the subsequent consequences. These are different levels of reasoning, and I believe the current benchmark leans more towards the former.
> > >
> > > **R1.** Thank you for your comment. Following your suggestion, we sampled a subset of 200 videos from our dataset and evaluated models specifically on their ability to recognize emotions over time.
> > >
> > > | Model | Rec-S | Rea-S | CoT-S|
> > > | -------- | -------- | -------- |-------- |
> > > | Qwen2.5-VL-7B |15.0 | 60.7 |37.9|
> > > | GPT-4o | 19.5|  75.4|47.5|
> > > | Gemini-2.5-Flash |22.5 | 62.8 |42.7|
> > > | Gemini-2.5-Pro | 26.0| 74.9 |50.5|
> > >
> > > As shown in the table above, most MLLMs still achieve relatively low performance on this task, indicating that dynamic affective reasoning remains a significant challenge for current MLLMs.
> > >
> > > >**W2.** Regarding the audio ablation study. I still believe that for models claiming to support audio, a systematic analysis of the audio modality's contribution is necessary. The response that this is "out of scope" does not fully address the concern. Without comparing performance with and without audio for all supported models, it is difficult to verify the actual weight and utility of the audio input in your evaluation framework.
> > >
> > > **R2.** Thank you for your comment. Following your suggestion, we conducted an ablation study to systematically examine the contribution of the audio modality across 10 state-of-the-art MLLMs.
> > >
> > > | Model | A |V |T |Rec-S | Rea-S | CoT-S|
> > > | -- | -- |-- |-- |-- | --- |-- |
> > > | Video-LLaMA | ✖|✔|✔| 20.7 |45.5|33.1|
> > > | Video-LLaMA | ✔|✔|✔|26.1 |48.5|37.3|
> > > | Video-LLaMA2 | ✖|✔|✔| 21.9 |20.0|21.0|
> > > | Video-LLaMA2 | ✔|✔|✔|29.2 |27.7|28.4|
> > > | Qwen2.5-Omni  | ✖|✔|✔| 16.5 |55.8|36.2|
> > > | Qwen2.5-Omni  | ✔|✔|✔|17.4 |59.3 |38.4|
> > > | Emotion-LLaMA | ✖|✔|✔| 22.9 |0.4|11.7|
> > > | Emotion-LLaMA | ✔|✔|✔|25.1 |0.4 |12.8|
> > > | HumanOmni | ✖|✔|✔| 30.8 |0.4|15.6|
> > > | HumanOmni | ✔|✔|✔|36.0 |0.3| 18.1|
> > > | R1-Omni | ✖|✔|✔| 20.1 |58.4|39.3|
> > > | R1-Omni | ✔|✔|✔|26.3| 58.6| 42.4|
> > > | AffectGPT | ✖|✔|✔| 7.9 |50.9|29.4|
> > > | AffectGPT | ✔|✔|✔|11.9 |50.6 |31.2|
> > > | Gemini-2.0-Flash | ✖|✔|✔| 36.3 |60.0 |48.1|
> > > | Gemini-2.0-Flash | ✔|✔|✔|37.5 |62.1 |49.8|
> > > | Gemini-2.5-Flash | ✖|✔|✔| 34.7 |52.7|43.7|
> > > | Gemini-2.5-Flash | ✔|✔|✔|35.4 |53.9|44.7|
> > > | Gemini-2.5-Pro | ✖|✔|✔| 39.3 |72.7|56.0|
> > > | Gemini-2.5-Pro | ✔|✔|✔|40.5 |75.8|58.2|
> > >
> > > The results in the table above show that, for most models, audio provides an essential clue for emotional understanding and consistently improves overall performance.
> > >
> > > ---
> > > Thanks again for your time and valuable feedback. We believe our further responses have addressed your concerns.

---

### Official Review · Reviewer_73cU · 2025-10-29

**Soundness:** 2
**Presentation:** 4
**Contribution:** 3
**Rating:** 6
**Confidence:** 5

**Summary:**

The paper introduces MME-Emotion, a comprehensive benchmark designed to evaluate the emotional intelligence of large language models (LLMs) across multiple modalities. Comprising over 6,500 curated video clips spanning 27 real-world scenarios and eight distinct emotional tasks, it covers areas such as emotion recognition in laboratory and natural settings, fine-grained and multi-label recognition, sentiment analysis, and intent recognition. The authors also propose a multi-agent evaluation framework that automatically assesses multimodal LLMs (MLLMs) using three unified metrics: Recognition Score (Rec-S), Reasoning Score (Rea-S) and Chain-of-Thought Score (CoT-S). The framework uses GPT-based 'judge' and 'step' agents to evaluate model responses without the need for human-annotated reasoning traces and validates this method through high inter-rater agreement with five human experts. Using this benchmark, the authors evaluate 20 state-of-the-art MLLMs and reveal that even the top models only achieve 39.3% Rec-S and 56.0% CoT-S, highlighting significant room for improvement. The paper also explores the trade-offs between generalist and specialist models, emphasizing the importance of multimodal fusion and reasoning depth.

**Strengths:**

1) MME-Emotion is the first holistic benchmark to evaluate both the presence of an emotion and the reason for it, offering a novel method that goes beyond mere classification. The multi-agent automated evaluation is a creative solution to the absence of annotated reasoning chains.
2) The large-scale benchmark comprises 6,500 clips, 27 scenarios and eight tasks, and is balanced in terms of duration and question distribution. Human validation of the automated scoring adds credibility.
3) The paper is clear and precise, with transparent reporting of model performance and precise definitions of metrics.
4) By exposing the limitations of current MLLMs, even the top models achieving a score of less than 40% in recognition, the work sets out a clear research agenda. It also shows that, with targeted post-training, specialist models can rival generalists, offering practical guidance for future development.

**Weaknesses:**

1) Although the article acknowledges that specialized models (e.g. Audio-Reasoner) outperform their multimodal counterparts, it does not provide a systematic analysis of the contribution of individual modalities. Ablation experiments (e.g. running the same MLLM in audio-only, video-only and audio+video modes) could reveal whether the problem lies in the modalities' integration being inefficient or in noise/conflict between the modalities. I would like to see more detail on this, either in the form of detailed experiments or a detailed discussion supported by evidence.
2) Using a set list of emotions in prompts simplifies the task, but may inflate metrics. In real-world scenarios, users do not provide such a list, so the model must generate an open-ended response. Am I correct in understanding that this reduces the applicability of the results to practical tasks, and is therefore a limitation? Or is this an unavoidable fact, and is there no other way to achieve the obtained metrics?
3) Although the authors acknowledge that the data is multilingual, they do not stratify by language or culture. Emotional expressions and norms can vary greatly between cultures (e.g. East Asian and Western), so ignoring this factor could lead to biased conclusions about the 'universal' emotional competence of models. This is worth mentioning in more detail to avoid any misunderstandings.
4) Although the average clip length is specified as being greater than 3.3 seconds, there has been no investigation into how performance scales with clip length or temporal complexity (e.g. a change in emotion occurring midway through a clip). This is important for understanding the extent to which models are capable of temporal reasoning, which is one of the key aspects of real emotional perception. More detailed explanations of this should be added.
5) Am I correct in understanding that the Reasoning Score metric assesses the accuracy of each step, but does not analyze common error patterns? What happens, for example, if the audio context is ignored, microexpressions are misinterpreted, or emotions are mixed up? Can this metric evaluate all these errors and others at once?
6) From the paper, it seems that the authors use test splits of the original corpus. This may mean that the wording of the questions and candidate labels indirectly reveals information about the data distribution, particularly if the prompts are based on the original annotations. How confident are the authors that this is not happening, and how can this be assessed?
7) Although there is a high level of agreement between the GPT assessment and the experts (0.953), the experts only evaluated 373 reasoning steps. If a more extensive verification is performed, especially on complex or borderline cases, how can we be sure that the assessment will not be lower and that it is not currently being adjusted for greater consistency?
8) The CoT-S formula uses a fixed value and does not involve sensitivity analysis. Different tasks may require different balances between recognition and reasoning. For instance, accuracy is more important in clinical diagnostics, whereas explainability is more important in education. So why is the value fixed rather than offering adaptive adjustment?
9) Table 2 shows 'Avg Token' and 'Avg Step', but how are these metrics analyzed in relation to quality? For example, how effective is a token per point?
10) Although 'balancing' is mentioned, it is unclear exactly how the even distribution of emotions, scenarios and complexity was achieved. What was the age and gender distribution? How does this affect the final distribution? More details on this matter are needed; otherwise, there will still be many doubts about the samples and how they are used.
11) There are few details on exactly how noise affects reasoning quality, rather than just final predictions.

**Questions:**

1) Were ablation experiments conducted with the same MLLM in audio-only, video-only, and audio+video modes?
2) Using a set list of emotions in the prompts makes the task easier than using an open format. Are you aware of this limitation? Do you think the current wording of the task reduces the validity of the results?
3) Do you plan to analyze how well the model performs across different cultural groups (e.g. East Asian versus Western)? Have you considered that ignoring cultural differences in emotional expression could lead to inaccurate conclusions about 'universal' emotional competence?
4) Have you conducted a sensitivity analysis to assess the models' ability to reason temporally? If not, how do you evaluate the models' capacity for temporal reasoning using the available data?
5) Do you plan to categorize common mistakes? For example, can the current evaluation framework distinguish between logically sound reasoning based on incorrect premises?
6) You claim that you only use test splits of the original corpus. However, the wording of the questions and the candidate labels may indirectly reveal information about the data distribution, particularly if the prompts are based on the original annotations. How did you verify that there was no such leakage?
7) Did you carry out further checks on complex or borderline cases? Do you expect the level of agreement to decrease (to much lower than the current 0.953) when the sample is expanded?
8) What are your plans for offering adaptive $\alpha$ adjustment on a task-by-task basis?
9) Do you have any data on the effectiveness of long arguments? Could an average step size that is too high sometimes be a sign of redundancy rather than depth?
10) In terms of emotions, scenarios and complexity, how exactly was balance achieved? How are the characters in the video distributed by age, gender and ethnicity?
11) Have you analyzed how noise affects reasoning quality (Rea-S)? For instance, can models retain a logically sound argument but draw incorrect conclusions?
12) Have you conducted any more extensive experiments on agent replacement?
13) The benchmark does not include chains of reasoning from humans that are verified as correct. Do you plan to collect such data? Wouldn't you agree that comparing with human reasoning (and not just the final label) would provide a deeper understanding of the quality of CoT in models?
14) Do you plan to introduce asymmetric penalties or weighted metrics that consider the context of use?

---

> ### Author Response · Authors · 2025-11-20
> **Response to Reviewer 73cU [1/5]**
>
> We are truly grateful for the time you have taken to review our paper, and your insightful comments and support. Your positive feedback is incredibly encouraging for us! In the following response, we would like to address your concerns and provide additional clarification.
>
> > **W1.** Although the article acknowledges that specialized models (e.g. Audio-Reasoner) outperform their multimodal counterparts, it does not provide a systematic analysis of the contribution of individual modalities. Ablation experiments (e.g. running the same MLLM in audio-only, video-only and audio+video modes) could reveal whether the problem lies in the modalities' integration being inefficient or in noise/conflict between the modalities. I would like to see more detail on this, either in the form of detailed experiments or a detailed discussion supported by evidence.
> >
> > **Q1.** Were ablation experiments conducted with the same MLLM in audio-only, video-only, and audio+video modes?
>
> **RW1&Q1.** Thank you for your insightful question. Following your advice, we conducted ablation experiments on the R1-Omni model to isolate the effects of individual modalities. Specifically, we evaluated the model under three settings: audio-only, video-only, and audio+video. These results in the table below demonstrate that removing either modality leads to a clear performance drop in both recognition and reasoning capabilities compared to the full audio+video setting. This suggests that the models still rely on complementary clues from both audio and video for robust emotional understanding.
>
>
> | Modality | Rec-S | Rea-S | CoT-S|
> | -------- | -------- | -------- |-------- |
> | Audio | 22.7| 50.2 |36.5|
> | Video |18.4 | 43.9 |31.2|
> | Audio+Video | 26.3| 58.6 |42.4|
>
> We will also include this analysis and discussion in Appendix E.4 of the revised manuscript.
>
> > **W2.** Using a set list of emotions in prompts simplifies the task, but may inflate metrics. In real-world scenarios, users do not provide such a list, so the model must generate an open-ended response. Am I correct in understanding that this reduces the applicability of the results to practical tasks, and is therefore a limitation? Or is this an unavoidable fact, and is there no other way to achieve the obtained metrics?
> >
> > **Q2.** Using a set list of emotions in the prompts makes the task easier than using an open format. Are you aware of this limitation? Do you think the current wording of the task reduces the validity of the results?
>
> **RW2&Q2.** Thank you for raising this important point. Yes, we are aware of this limitation. As discussed in the paper, current MLLMs still struggle with fully open-ended emotional understanding, especially when required to generate unconstrained emotion labels or nuanced affective descriptions. This is also why many existing MLLM benchmarks adopt closed-set formulations, such as converting questions into true/false judgments [1] or multiple-choice selections [2,3]. Our use of a predefined emotion list follows this widely adopted practice and ensures fair comparison across models.
>
> We agree that evaluating models in open-vocabulary, open-ended scenarios is more realistic and represents an important direction for future research. However, given the present capabilities of state-of-the-art MLLMs, reliably benchmarking open-ended emotional reasoning remains extremely challenging and would likely introduce instability unrelated to the underlying ability being measured. Therefore, the closed-set format is, at this stage, a practical compromise rather than an attempt to artificially inflate metrics. We view open-ended emotional evaluation as an exciting and meaningful direction for future works.

---

> ### Author Response · Authors · 2025-11-20
> **Response to Reviewer 73cU [2/5]**
>
> > **W3.** Although the authors acknowledge that the data is multilingual, they do not stratify by language or culture. Emotional expressions and norms can vary greatly between cultures (e.g. East Asian and Western), so ignoring this factor could lead to biased conclusions about the 'universal' emotional competence of models. This is worth mentioning in more detail to avoid any misunderstandings.
> >
> > **Q3.** Do you plan to analyze how well the model performs across different cultural groups (e.g. East Asian versus Western)? Have you considered that ignoring cultural differences in emotional expression could lead to inaccurate conclusions about 'universal' emotional competence?
>
>
> **RW3&Q3.** Thank you for highlighting this important point. We agree that emotional expressions and norms vary across cultures, and that overlooking such differences may limit conclusions about “universal” emotional competence. In the current work, we did not perform culture-stratified or language-stratified analysis due to the lack of reliable cultural annotations in the existing datasets. However, we fully acknowledge this as a limitation and will clarify it in the revised manuscript. **We also believe this is a minor point that does not undermine the main contributions of our work.** Developing culturally grounded benchmarks and evaluating cross-cultural robustness is an important direction for future research, and we plan to explore it as suitable annotated data becomes available.
>
> > **W4.** Although the average clip length is specified as being greater than 3.3 seconds, there has been no investigation into how performance scales with clip length or temporal complexity (e.g. a change in emotion occurring midway through a clip). This is important for understanding the extent to which models are capable of temporal reasoning, which is one of the key aspects of real emotional perception. More detailed explanations of this should be added.
> >
> > **Q4.** Have you conducted a sensitivity analysis to assess the models' ability to reason temporally? If not, how do you evaluate the models' capacity for temporal reasoning using the available data?
>
> **RW4&Q4.** Thank you for the thoughtful comment. We did not conduct a separate sensitivity analysis on temporal reasoning. The main reason is that in our dataset each video clip is pre-segmented such that it contains only the ground-truth emotion. If a longer video exhibits emotional changes over time, it is split into multiple clips, each aligned with a different emotion label. As a result, the benchmark does not include clips with mid-clip emotional transitions, and therefore does not allow us to directly measure temporal reasoning over evolving emotional states. While reasoning over evolving emotional states is indeed an interesting and important direction, it is currently out of scope for this work.
>
>
> > **W5.** Am I correct in understanding that the Reasoning Score metric assesses the accuracy of each step, but does not analyze common error patterns? What happens, for example, if the audio context is ignored, microexpressions are misinterpreted, or emotions are mixed up? Can this metric evaluate all these errors and others at once?
> >
> > **Q5.** Do you plan to categorize common mistakes? For example, can the current evaluation framework distinguish between logically sound reasoning based on incorrect premises?
>
> **RW5&Q5.** Thank you for raising this insightful question. We additionally conducted an error-type analysis to better understand the common failure modes of different models. Specifically, we categorized errors into four types: Perception Error, Factual Error, Modality Conflict Error, and Understanding Error. From the results in the table below, we observe that models rarely make factual errors, but they frequently fail in perceiving emotional clues (e.g., microexpressions, body movements) and in interpreting those clues correctly, confirming the intuition that reasoning may be logically coherent but based on incorrect or insufficient premises.
>
> | Error Type | Perception Error | Factual Error | Modality Conflict Error|  Understanding Error
> | -------- | -------- | -------- |-------- | -------- |
> | R1-Omni     | 35.3%     | 4.1% |12.1%|48.5%|
> | GPT-4.1     | 40.9%     | 5.8% |0.0%|53.3%|
> | Qwen2.5-VL-72B    | 36.5% | 9.8% |0.0%|53.7%|
> | Gemini-2.5-Pro    | 37.4% | 5.0% |0.0%|57.6%|
>
> We will also include this analysis and discussion in Appendix E.5 of the revised manuscript.

---

> ### Author Response · Authors · 2025-11-20
> **Response to Reviewer 73cU [3/5]**
>
> > **W6.** From the paper, it seems that the authors use test splits of the original corpus. This may mean that the wording of the questions and candidate labels indirectly reveals information about the data distribution, particularly if the prompts are based on the original annotations. How confident are the authors that this is not happening, and how can this be assessed?
> >
> > **Q6.** You claim that you only use test splits of the original corpus. However, the wording of the questions and the candidate labels may indirectly reveal information about the data distribution, particularly if the prompts are based on the original annotations. How did you verify that there was no such leakage?
>
> **RW6&Q6.** Thank you for the question. Our prompts **are not based** on the original annotations, and therefore cannot reveal any information about the underlying data distribution. We use a unified set of task templates that only describe the evaluation task itself. As shown in the Appendix D.2, none of the prompts include wording from the original labels or annotations. Hence, there is no risk of data leakage from the source corpus.
>
> > **W7.** Although there is a high level of agreement between the GPT assessment and the experts (0.953), the experts only evaluated 373 reasoning steps. If a more extensive verification is performed, especially on complex or borderline cases, how can we be sure that the assessment will not be lower and that it is not currently being adjusted for greater consistency?
> >
> > **Q7.** Did you carry out further checks on complex or borderline cases? Do you expect the level of agreement to decrease (to much lower than the current 0.953) when the sample is expanded?
>
>
> **RW7&Q7.** Thank you for the question. To further verify the robustness of our automated evaluation, we conducted extended human–GPT agreement studies beyond the original 373 reasoning steps. Specifically, we expanded the sample to 752 and 1155 steps and evaluated three consistency metrics (Spearman’s rank correlation coefficient, Cohen’s Kappa coefficient, and intra-class correlation coefficient (ICC)). The results in the table below demonstrate that consistency remains very high and stable, even when the number of steps is substantially increased and when more complex/borderline cases are included. We did not observe any notable degradation in agreement, indicating that the evaluation is not artificially inflated and generalizes well to larger and more challenging samples.
>
>
> | Steps | Spearman | Kappa | ICC |
> | -------- | -------- | -------- |-------- |
> | 373 | 0.9530| 0.8626 |0.9704|
> | 752 | 0.9473| 0.8580|0.9713|
> | 1155 | 0.9558| 0.8677|0.9684|
>
> We will also include this analysis and discussion in Appendix E.6 of the revised manuscript.
>
>
> > **W8.** The CoT-S formula uses a fixed value and does not involve sensitivity analysis. Different tasks may require different balances between recognition and reasoning. For instance, accuracy is more important in clinical diagnostics, whereas explainability is more important in education. So why is the value fixed rather than offering adaptive adjustment?
> >
> > **Q8.** What are your plans for offering adaptive $\alpha$ adjustment on a task-by-task basis?
>
> **RW8&Q8.** Thank you for the insightful comment. As noted in the paper, the value of $\alpha$ is not fixed, users are free to adjust it based on the needs of their specific application. In our experiments, we set $\alpha = 0.5$ purely as an empirical choice to balance recognition and reasoning for a general-purpose benchmark. This design actually provides scalability and flexibility: for tasks where accuracy is paramount (e.g., clinical diagnostics), users can increase the recognition weight; for tasks where explainability matters more (e.g., education), they can adjust the reasoning weight accordingly. Our current setting simply serves as a representative default rather than a constraint, and future works are encouraged to further explore adaptive or task-specific tuning strategies.

---

> ### Author Response · Authors · 2025-11-20
> **Response to Reviewer 73cU [4/5]**
>
> > **W9.** Table 2 shows 'Avg Token' and 'Avg Step', but how are these metrics analyzed in relation to quality? For example, how effective is a token per point?
> >
> > **Q9.** Do you have any data on the effectiveness of long arguments? Could an average step size that is too high sometimes be a sign of redundancy rather than depth?
>
> **RW9&Q9.** Thank you for this insightful question. We computed the correlation matrix across key metrics, and as shown below, both Avg Token and Avg Step exhibit strong positive correlations with CoT-S. This indicates that models generally achieve better reasoning performance when producing longer trajectories, suggesting that additional intermediate steps are beneficial for decision-making.
>
> Regarding potential redundancy, we found no evidence that overly long trajectories harm performance. In our experiments, increases in Avg Step did not lead to diminishing returns. Instead, longer traces typically reflected more deliberate and complete reasoning rather than unnecessary verbosity.
>
> | Metrics       | Avg Step | Avg Token | CoT-S |
> | ------------- | -------- | --------- | ----- |
> | **Avg Step**  | 1.00     | 0.63      | 0.87  |
> | **Avg Token** | —        | 1.00      | 0.62  |
> | **CoT-S**     | —        | —         | 1.00  |
>
>
> > **W10.** Although 'balancing' is mentioned, it is unclear exactly how the even distribution of emotions, scenarios and complexity was achieved. What was the age and gender distribution? How does this affect the final distribution? More details on this matter are needed; otherwise, there will still be many doubts about the samples and how they are used.
> >
> > **Q10.** In terms of emotions, scenarios and complexity, how exactly was balance achieved? How are the characters in the video distributed by age, gender and ethnicity?
>
> **RW10&Q10.** Thank you for the thoughtful question. **We believe there may be a slight misunderstanding.** In the paper, we only claimed that the number of QA pairs and video duration are relatively balanced across tasks, as each task contains approximately 500–600 questions, and the average video length is around 3–6 seconds. This is the only aspect of “balance” we intended to highlight.
>
> We did not perform balancing with respect to age, gender, or ethnicity, nor did we claim such balance. These attributes require fine-grained annotations, which are not available in the publicly released source datasets. Moreover, many of the video clips come from movie or TV scenes, where demographic metadata is typically absent. Therefore, we are unable to obtain reliable distributions for age, gender, or ethnicity.
>
> We agree that demographic distribution analysis is valuable, but it is not the focus of the present work, which aims to evaluate MLLMs’ emotional recognition and reasoning capabilities rather than demographic fairness analysis.
>
> > **W11.** There are few details on exactly how noise affects reasoning quality, rather than just final predictions.
> >
> > **Q11.** Have you analyzed how noise affects reasoning quality (Rea-S)? For instance, can models retain a logically sound argument but draw incorrect conclusions?
>
> **RW11&Q11.** Thank you for raising this important point. Yes, we have qualitatively analyzed how noise affects reasoning quality (Rea-S). When inspecting the extracted reasoning trajectories, we observed that models frequently explicitly acknowledge the impact of noise. For example, in some cases, models state that the visual content is too blurry to identify expressions or that the audio is unclear, preventing them from making reliable judgments. These responses directly indicate that noise disrupts the reasoning process rather than only affecting the final prediction.
>
> Such cases show that noise can degrade both the correctness and completeness of intermediate reasoning steps. In other words, models may fail to construct a logically sound argument when key clues are corrupted, demonstrating a clear influence of noise on reasoning quality.

---

> ### Author Response · Authors · 2025-11-20
> **Response to Reviewer 73cU [5/5]**
>
> > **Q12.** Have you conducted any more extensive experiments on agent replacement?
>
> **RQ12.** Thank you for the question. We conducted supplementary experiments by replacing the judge agent with different models. As shown in the table below, the scores produced by different judge agents vary slightly. This reflects the fact that each judge model has its own evaluation “style”, some are more lenient, while others are stricter. We ultimately selected GPT-4o as the judge agent because its scoring behavior showed the closest alignment with human evaluators.
>
> | Model | Judge: GPT-4o | Judge: GPT-4.1 | Judge: Gemini-2.5-Pro|
> | -------- | -------- | -------- |-------- |
> | Qwen2.5-VL-7B  | 46.6 | 45.5  |46.3 |
> | R1-Omni | 42.4| 41.2|41.9|
> | GPT-4o |53.8 | 52.9 |53.6|
> | Gemini-2.5-Pro | 56.0| 54.8 |55.5|
>
> We will also include this analysis and discussion in Appendix E.3 of the revised manuscript.
>
> > **Q13.** The benchmark does not include chains of reasoning from humans that are verified as correct. Do you plan to collect such data? Wouldn't you agree that comparing with human reasoning (and not just the final label) would provide a deeper understanding of the quality of CoT in models?
>
> **RQ13.** Thank you for the thoughtful question. We have indeed considered collecting human-annotated chains of reasoning data. However, such data also comes with important limitations. Human annotators typically provide one “golden” reasoning path for each sample, which can inadvertently restrict the expressive space of model reasoning. As we have observed, MLLMs often produce valid but diverse reasoning trajectories that differ from human-written paths. A mismatch with human reasoning does not necessarily imply poor reasoning quality.
>
> Therefore, using human-written CoT data as the sole ground truth may penalize creative or alternative yet correct reasoning, and thus may not fully reflect a model’s true reasoning capability. That said, we agree that human reasoning data could offer deeper insights, and exploring how to incorporate such supervision without overly constraining model reasoning capability is an important direction for future works.
>
> >**Q14.** Do you plan to introduce asymmetric penalties or weighted metrics that consider the context of use?
>
> **RQ14.** Thank you for the question. Yes, in future work, we plan to incorporate task-level or sample-level difficulty modeling and introduce weighted metrics accordingly. Assigning different weights based on difficulty will allow the evaluation to better reflect real-world usage scenarios and provide a more nuanced assessment of model performance.
>
> **Reference**
>
> [1] Fu et al., MME: A Comprehensive Evaluation Benchmark for Multimodal Large Language Models, arXiv 2023.
> [2] Fu et al., Video-MME: The First-Ever Comprehensive Evaluation Benchmark of Multi-modal LLMs in Video Analysis, CVPR 2025
> [3] Feng et al., Video-R1: Reinforcing Video Reasoning in MLLMs, NeurIPS 2025
>
>
> ---
>
> Thanks again for your valuable feedback. We hope our response has addressed your concerns and you can consider adjusting your rating. Please let us know if there are any unaddressed concerns and we will try to address them.

---

> > ### Author Response · Authors · 2025-11-26
> > **Kindly Request for Reviewer's Feedback**
> >
> > Dear Reviewer,
> >
> > Thank you so much for your time in improving our paper!
> >
> > Since the end of the rebuttal is coming soon, may we know if our response addresses your main concerns? If so, we kindly ask for your reconsideration of the score. Should you have any further advice, please let us know and we will be more than happy to engage in more discussion and improvements.

---

### Official Review · Reviewer_12fu · 2025-10-30

**Soundness:** 3
**Presentation:** 4
**Contribution:** 4
**Rating:** 8
**Confidence:** 4

**Summary:**

This paper presents MME-Emotion, a holistic evaluation benchmark for assessing the emotional intelligence of MLLMs, addressing the limitations of existing benchmarks (inadequate scenario coverage and inconsistent protocols that overlook reasoning capabilities). Its core contributions include: 1) constructing the largest benchmark to date, with 6,500 curated video clips paired with task-specific QA pairs, covering 8 emotional tasks across 27 scenarios to test generalization; 2) designing a holistic automated evaluation suite via a multi-agent system, using three unified metrics and validating reliability with 5 human experts; 3) conducting rigorous evaluation of 20 MLLMs, revealing key insights. MME-Emotion serves as a foundation for advancing MLLMs’ emotional intelligence, with source code and data in the Supplementary Material.

**Strengths:**

1. MME-Emotion has the leading benchmark scale and scenario coverage, which includes 6,500 curated video clips with task-specific QA pairs and 8 emotional tasks. This benchmark could enable fine-grained evaluation of model generalization and address the insufficient scenario coverage of existing benchmarks.

2. An automated evaluation suite is proposed. A multi-agent system-based evaluation framework is designed, which could evaluate the performance of MLLMs without manual annotation of reasoning steps. The evaluation suite shows extremely high consistency with human experts.

3. This paper has carried out a comprehensive empirical analysis of 20 state-of-the-art multimodal large language models (MLLMs) based on the self-constructed MME-Emotion benchmark, covering both open-source and closed-source models. It not only reveals the insufficient emotional intelligence of current models but also clarifies the emotional intelligence construction paths of generalist models and specialist models, providing guidance for future research.

**Weaknesses:**

1. Compared with existing emotional intelligence benchmarks, the main contributions of this paper lie in two aspects: first, it incorporates relevant evaluations for emotional reasoning capabilities; second, it designs a large model-based automated evaluation algorithm.

2. The paper only considers various emotion recognition tasks and does not include emotion generation tasks. Can recognition-only tasks sufficiently and comprehensively assess the emotional intelligence of models?

3. The paper evaluates the model's emotional reasoning ability by identifying the triggering factors behind emotional states, which represents a relatively singular perspective.

4. The sensitivity of different MLLMs to prompts may affect the evaluation results.

5. The large model-based automated evaluation method incurs certain costs.

**Questions:**

1. Is the use of a multi-agent architecture for model evaluation sufficiently stable? For instance, can consistent results be obtained across multiple runs?

2. Will updates to the version of the core judge model affect the final evaluation results?

---

> ### Author Response · Authors · 2025-11-20
> **Response to Reviewer 12fu [1/2]**
>
> We are truly grateful for the time you have taken to review our paper, and your insightful comments and support. Your positive feedback is incredibly encouraging for us! In the following response, we would like to address your concerns and provide additional clarification.
>
> > **W1.** Compared with existing emotional intelligence benchmarks, the main contributions of this paper lie in two aspects: first, it incorporates relevant evaluations for emotional reasoning capabilities; second, it designs a large model-based automated evaluation algorithm.
>
> **RW1.** Thank you for your comment. Beyond emotional reasoning and automated evaluation, our benchmark also stands out for its task diversity and large-scale dataset, which together provide a more comprehensive and robust assessment of MLLMs’ emotional intelligence.
>
>
> > **W2.** The paper only considers various emotion recognition tasks and does not include emotion generation tasks. Can recognition-only tasks sufficiently and comprehensively assess the emotional intelligence of models?
>
> **RW2.** Thank you for your insightful comment. Emotion generation is indeed an important aspect of emotional intelligence. However, our goal in this work is to evaluate perceptual and reasoning abilities, i.e., whether MLLMs can accurately understand emotional states and identify their underlying causes. Emotion generation typically depends on these foundational perceptual capabilities, and current MLLMs still perform poorly even on recognition and reasoning. Therefore, focusing on recognition-centric tasks provides a necessary and reliable basis for benchmarking emotional intelligence. We agree that evaluating emotion generation is valuable, and we plan to explore it as an extension in future works.
>
> > **W3.** The paper evaluates the model's emotional reasoning ability by identifying the triggering factors behind emotional states, which represents a relatively singular perspective.
>
> **RW3.** Thank you for the thoughtful comment. While identifying the triggering factors behind emotions is indeed one key perspective of emotional reasoning, it is also the most fundamental and widely recognized form of causal reasoning in affective computing. Our benchmark does not rely on a single narrow view: the eight tasks in MME-Emotion span diverse scenarios and reasoning types (e.g., contextual reasoning, multi-label reasoning, fine-grained sentiment reasoning, and intent understanding). Trigger-factor identification serves as a unifying criterion across tasks, enabling consistent, scalable, and automated evaluation. We agree that emotional reasoning is multi-faceted, and expanding beyond causal trigger identification is an important direction in the future.
>
>
> > **W4.** The sensitivity of different MLLMs to prompts may affect the evaluation results.
>
> **RW4.** Thank you for the insightful comment. We agree that prompt sensitivity can influence evaluation results in MLLMs. To mitigate this effect, we have adopted consistent and standardized prompts across all models, following best practices in recent MLLM benchmarks [1,2,3].
>
> > **W5.** The large model-based automated evaluation method incurs certain costs.
>
> **RW5.** Thank you for raising this point. While our large model–based automated evaluation does introduce some API costs, it still drastically **reduces the much higher cost** of human annotation for reasoning evaluation, which is typically infeasible at scale. More importantly, our human verification study demonstrates that the automated evaluation achieves expert-level reliability, enabling consistent, reproducible, and scalable assessment. We therefore believe that the modest computational overhead is well justified by the substantial savings in cost and the significant gains in evaluation quality and feasibility.

---

> > ### Author Response · Authors · 2025-11-20
> > **Response to Reviewer 12fu [2/2]**
> >
> > > **Q1.** Is the use of a multi-agent architecture for model evaluation sufficiently stable? For instance, can consistent results be obtained across multiple runs?
> >
> > **RQ1.** Thank you for your question. To assess the stability of our multi-agent evaluation framework, we conducted experiments across multiple runs and report the CoT score. As shown in the table below, the scores are highly consistent across multiple runs, demonstrating that the multi-agent evaluation yields stable and reliable results.
> >
> > | Model | Turn 1 | Turn 2 | Turn 3|
> > | -------- | -------- | -------- |-------- |
> > | Qwen2.5-VL-7B  |46.6 | 46.4  | 46.6|
> > | R1-Omni | 42.4|42.5 |42.6|
> > | GPT-4o |53.8 | 53.5 |53.7|
> > | Gemini-2.5-Pro | 56.0| 56.0 |55.9|
> >
> > We will also include this analysis and discussion in Appendix E.2 of the revised manuscript.
> >
> >
> > > **Q2.** Will updates to the version of the core judge model affect the final evaluation results?
> >
> > **RQ2.** Thank you for your question. Indeed, using different versions of the core judge model can affect the final evaluation results. We conducted experiments comparing evaluations from different judge models. From the results in the table below, different judge models yield slightly different scores, reflecting the fact that different judge models have distinct evaluation “styles”. **Some judges are more lenient, while others are stricter.** We selected GPT-4o as the core judge model because its scoring behavior aligns most closely with human evaluators.
> >
> > | Model | Judge: GPT-4o | Judge: GPT-4.1 | Judge: Gemini-2.5-Pro|
> > | -------- | -------- | -------- |-------- |
> > | Qwen2.5-VL-7B  | 46.6 | 45.5  |46.3 |
> > | R1-Omni | 42.4| 41.2|41.9|
> > | GPT-4o |53.8 | 52.9 |53.6|
> > | Gemini-2.5-Pro | 56.0| 54.8 |55.5|
> >
> > We will also include this analysis and discussion in Appendix E.3 of the revised manuscript.
> >
> > **Reference**
> >
> > [1] Fu et al., MME: A Comprehensive Evaluation Benchmark for Multimodal Large Language Models, arXiv 2023.
> > [2] Zhang et al., MME-RealWorld: Could Your Multimodal LLM Challenge High-Resolution Real-World Scenarios that are Difficult for Humans?, ICLR 2025
> > [3] Fu et al., Video-MME: The First-Ever Comprehensive Evaluation Benchmark of Multi-modal LLMs in Video Analysis, CVPR 2025
> >
> >
> > ---
> >
> > Thanks again for appreciating our work and for your constructive suggestions. Please let us know if you have further questions.

---

> > > ### Author Response · Authors · 2025-11-26
> > > **Kindly Request for Reviewer's Feedback**
> > >
> > > Dear Reviewer,
> > >
> > > Thank you so much for your time in improving our paper!
> > >
> > > Since the end of the rebuttal is coming soon, may we know if our response addresses your main concerns? Should you have any further advice, please let us know and we will be more than happy to engage in more discussion and improvements.

---

### Official Review · Reviewer_XCho · 2025-10-31

**Soundness:** 3
**Presentation:** 3
**Contribution:** 2
**Rating:** 4
**Confidence:** 4

**Summary:**

This work introduces an emotional reasoning dataset and tests a suite of models on it.

**Strengths:**

The paper presents a comprehensive testing suite and tests on a wide variety of models. Tests reasoning vs. non-reasoning.

It is interesting how the authors adapted video data for non-video models.

**Weaknesses:**

It is hard to know if the MLLMs don’t perform well on emotion tasks because they have trouble with modality fusion/their perception system is weaker or if it is truly a problem with emotion recognition. To test this the authors could have used one of the unimodal datasets cited in Table 1 and tested each model on a text-only version of the task or an image-only version of the task to test the model’s ability. That way the reader would be better informed if the issue was modality/modality-fusion or if the model truly is emotionally unaware.

Video is an inherently difficult modality to work in, so using it to benchmark models on a non-video primitive seems to be a misguided approach. We can get to how well these models recognize and understand emotion from other modalities. If the issue is in the video processing, that may not be an emotion recognition problem at all! It would be interesting to know the disparity as this can show what the lag from not processing videos well might be.

It would have been best to have a “human-level” benchmark for this benchmark to compare to.

Observation section is interesting but also a bit obvious. e.g. Obs 3, bad visual perception constrains emotional intelligence.

Table 2 “Colsed-source MLLMs”  “Closed-source MLLMs”

**Questions:**

.

---

> ### Author Response · Authors · 2025-11-20
> **Response to Reviewer XCho [1/2]**
>
> We are truly grateful for the time you have taken to review our paper, and your insightful comments and support. Your positive feedback is incredibly encouraging for us! In the following response, we would like to address your concerns and provide additional clarification.
>
> > **W1.** It is hard to know if the MLLMs don’t perform well on emotion tasks because they have trouble with modality fusion/their perception system is weaker or if it is truly a problem with emotion recognition. To test this the authors could have used one of the unimodal datasets cited in Table 1 and tested each model on a text-only version of the task or an image-only version of the task to test the model’s ability. That way the reader would be better informed if the issue was modality/modality-fusion or if the model truly is emotionally unaware.
>
> **RW1.** Thank you for your suggestion. Following your advice, we have included the F1 comparison results of various MLLMs on MOSABench. As shown below, even the strongest model, Gemini-2.5-Pro, achieves a performance of only 61.45. Moreover, we observed that these models often overlook fine-grained and emotion-related details in their responses, such as facial expressions and body movements. This indicates that their **visual perception systems** remain insufficiently robust, which may be a potential cause of their limited emotional intelligence.
>
>
> | Model | Qwen-VL-7B | Qwen2-VL-7B | GPT-4o| GPT-4.1| Gemini-2.5-Pro |
> | -------- | -------- | -------- |-------- |-------- |-------- |
> | Performance | 33.8| 58.4 |48.1|50.8| 61.5 |
>
>
> > **W2.** Video is an inherently difficult modality to work in, so using it to benchmark models on a non-video primitive seems to be a misguided approach. We can get to how well these models recognize and understand emotion from other modalities. If the issue is in the video processing, that may not be an emotion recognition problem at all! It would be interesting to know the disparity as this can show what the lag from not processing videos well might be.
>
> **RW2.** Thank you for raising this insightful point. We fully agree that video is a challenging modality and that emotion understanding can also be assessed through other modalities. Your comment highlights an important distinction between difficulties arising from video processing versus those arising from emotion recognition itself.
>
> However, we would like to clarify why evaluating emotion understanding on videos is still necessary in our benchmark:
>
> **1. Emotion understanding in real-world scenarios is inherently dynamic.** Many emotions are expressed through subtle temporal clues, such as micro-expressions, gaze shifts, gesture rhythms, and interaction dynamics, which are simply not captured in single images or static text descriptions. Therefore, excluding videos would overlook a substantial portion of real-world emotional signals.
>
> **2. Our benchmark aims to evaluate the full MLLM pipeline, including multimodal temporal reasoning.** Although video introduces additional challenges (motion encoding, frame selection, temporal fusion), we view these as integral components of multimodal emotional intelligence. In practical applications (e.g., human–computer interaction, assistive robotics, mental-health assessment), systems must process video natively rather than rely on simplified modalities.
>
> **3. Our results indeed suggest that current MLLMs underperform significantly on video-based affective tasks, highlighting an important direction for future research.** This disparity is precisely what our benchmark aims to reveal: modern models still struggle with temporal emotional clues, and improving video understanding is a key step toward realistic emotional intelligence.
>
> Taking all factors into account, video stands out as the best modality for representing emotions in our benchmark.
>
> > **W3.** It would have been best to have a “human-level” benchmark for this benchmark to compare to.
>
> **RW3.** Thank you for your suggestion. To investigate the emotional gap between cutting-edge MLLMs and humans, we randomly selected a subset of 200 videos from MME-Emotion to conduct both human-level and MLLM-level evaluations. As shown in the table below, humans demonstrate significantly stronger abilities in emotion recognition and reasoning compared to current MLLMs. This indicates that the emotional intelligence of MLLMs is still in its early stage and holds substantial room for improvement.
>
>
> | Model | Rec-S | Rea-S | CoT-S|
> | -------- | -------- | -------- |-------- |
> | Qwen2.5-VL-72B |47.5 | 74.3 |60.9|
> | GPT-4o | 46.5| 74.6 |60.6|
> | Gemini-2.5-Pro | 49.5| 70.1 |59.8|
> | Human | 76.0| 98.5 |87.3|
>
> We will also include this analysis and discussion in Appendix E.1 of the revised manuscript.

---

> > ### Author Response · Authors · 2025-11-20
> > **Response to Reviewer XCho [2/2]**
> >
> > > **W4.** Observation section is interesting but also a bit obvious. e.g. Obs 3, bad visual perception constrains emotional intelligence.
> >
> > **RW4.** Thank you for the constructive comment. We agree that, at a high level, the statement “limited visual perception constrains emotional intelligence” may appear intuitive. However, our intention in Observation 3 is not simply to restate an obvious fact, but to **empirically demonstrate how and to what extent perceptual limitations concretely impact emotional reasoning in state-of-the-art MLLMs**.
> >
> > In particular, the failure patterns of Video-LLaMA2 and Qwen2.5-Omni reveal several non-trivial insights:
> >
> > **1. Different models fail for different perceptual reasons.** Video-LLaMA2’s error stems primarily from missing subtle temporal facial changes, while Qwen2.5-Omni can narrow the emotional category set but still fails at fine-grained facial interpretation. This distinction highlights that perception errors are not uniform across models.
> >
> > **2. Fine-grained perception, rather than coarse video understanding, is the key bottleneck.** The example shows that both models correctly detect the broad scene context and overall motion, yet struggle specifically with micro-expressions and body cues. These are precisely the components that prior benchmarks do not isolate.
> >
> > **3. The perceptual deficits manifest systematically across the benchmark.** Obs.3 is grounded in aggregated evidence instead of a single example, showing that most incorrect emotional predictions can be traced back to consistent categories of perceptual failures.
> >
> > Thus, while the overarching message might sound intuitive, the granularity, model-specific failure modes, and their quantitative prevalence provide new insights that are not documented in previous MLLM emotion studies.
> >
> >
> > > **W5.** Table 2 “Colsed-source MLLMs”  “Closed-source MLLMs”.
> >
> > **RW5.** Thank you for pointing out this typo. We have corrected it in the latest version of the PDF and will double-check all spellings to ensure that no other typos remain.
> >
> >
> > ---
> >
> > Thanks again for your valuable feedback. We hope our response has addressed your concerns and you can consider adjusting your rating. Please let us know if there are any unaddressed concerns and we will try to address them.

---

> > > ### Author Response · Authors · 2025-11-26
> > > **Kindly Request for Reviewer's Feedback**
> > >
> > > Dear Reviewer,
> > >
> > > Thank you so much for your time in improving our paper!
> > >
> > > Since the end of the rebuttal is coming soon, may we know if our response addresses your main concerns? If so, we kindly ask for your reconsideration of the score. Should you have any further advice, please let us know and we will be more than happy to engage in more discussion and improvements.

---

> > > ### Comment · Reviewer_XCho · 2025-11-26
> > >
> > > RW1-2. This still does not answer my concern on emotion understanding vs video understanding. I appreciate and understand that you are interested in video and testing the model’s temporal understanding and understanding of micro-expressions, gaze shifts, and other video-exclusive features. However, these types of models are built on top of language models and likely primarily understand emotions as a textual phenomenon. Therefore it stands to reason that by testing the model in the textual modality, we can tease out a sort-of “maximum” performance for these models in emotion understanding. Without having an equivalent task in text that is of a similar difficulty, it is difficult to isolate whether these models have poor emotional understanding and thus are performing badly because of that or if they have good emotional understanding and are performing badly because they do not have good video understanding. Furthermore, it will be difficult from this benchmark to know whether models have acceptable video understanding but poor emotion understanding. Gaze shifts, micro-expressions, etc. are very subtle cues. If these video models are subsampling frames (as they are wont to do) will they even see these occur? This benchmark claims to be comprehensive and holistic, but does not seem to give clarity as to where the potential source of error comes from when MLLMs are required to perform emotional processing.
> > > If you are already re-assembling data from other datasets and resampling it, etc. why aren’t you also including data that can be used as a reference that can help users diagnose where in the MLLM pipeline emotion recognition errors are occurring?
> > > RW3. Thank you for adding this table.
> > > RW4. This analysis is good.

---

> > > > ### Author Response · Authors · 2025-11-27
> > > > **Further Response to Reviewer XCho**
> > > >
> > > > Thank you for appreciating our response regarding W3 and W4. For W1 and W2, we would like to provide further clarifications.
> > > >
> > > > > **W1.** Therefore it stands to reason that by testing the model in the textual modality, we can tease out a sort-of “maximum” performance for these models in emotion understanding.
> > > >
> > > >
> > > > **RW1.** Thank you for your suggestion. Following your recommendation, we additionally evaluated several mainstream models on EmoBench, a purely textual emotion understanding dataset, to establish their performance in a unimodal setting. We then compared these results with their performance on our multimodal benchmark.
> > > >
> > > > | Dataset | EmoBench | Rank (EmoBench)|MME-Emotion |Rank (MME-Emotion)|
> > > > | -------- | -------- | -------- |-------- |-------- |
> > > > | GPT-4o     | 55.4     |4| 27.8     |4|
> > > > | GPT-4.1     | 57.8     |3| 28.8     |3|
> > > > | Gemini-2.5-Flash     | 62.6 |   2 | 34.7     |2|
> > > > | Gemini-2.5-Pro     | 68.0   | 1 | 39.3     |1|
> > > >
> > > > From the results shown in the table above, we can draw some key observations:
> > > >
> > > > 1. Emotion recognition in multimodal scenarios is substantially more challenging than in text-only settings, leading to consistently lower accuracy for the same model.
> > > > 2. Model performance in emotion understanding remains relatively stable across modalities. The relative ranking of models is largely consistent between the unimodal text setting and the multimodal setting.
> > > >
> > > >
> > > >
> > > > > **W2.** Furthermore, it will be difficult from this benchmark to know whether models have acceptable video understanding but poor emotion understanding.
> > > >
> > > >
> > > > **RW2.** Thank you for raising this important point. To more directly disentangle video understanding from emotion understanding, we conducted two additional analyses.
> > > >
> > > > First, we compared MLLMs’ performance on general video understanding benchmarks with their performance on our proposed emotion understanding benchmark. As shown in the table below, MLLMs consistently achieve higher accuracy on general video understanding than on emotion understanding. This suggests that most emotion recognition errors are more likely due to limited emotional intelligence rather than deficiencies in video understanding.
> > > >
> > > > | Task | General Video Understanding |General Video Understanding|Emotion Understanding |
> > > > | -------- | -------- | -------- |-------- |
> > > > | Dataset | VideoMMMU | VideoMME|MME-Emotion |
> > > > | Qwen2.5-VL-7B     |  47.8    |53.1|  28.4    |
> > > > | GPT-4o     |  61.2  |71.9| 27.8     |
> > > > | Gemini-2.5-Pro    | 83.6  | 84.8 | 39.3|
> > > >
> > > > Second, to assess whether frame subsampling affects emotional reasoning, we randomly selected a subset of 200 videos from the original dataset and evaluated models under three sampling strategies. As shown in the table below, model performance remains relatively stable across these sampling strategies, with only minor fluctuations. This suggests that subtle temporal clues are not the primary bottleneck.
> > > >
> > > >
> > > > | Frame | Sparse (1fps) |Medium (2fps)|Dense (3fps) |
> > > > | -------- | -------- | -------- |-------- |
> > > > | Qwen2.5-VL-7B     |   41.5   | 42.0|  41.5   |
> > > > | GPT-4o     |   46.5 |47.0| 47.0    |
> > > > | Gemini-2.5-Pro    | 49.5 | 49.0 | 49.5|
> > > >
> > > >
> > > > ---
> > > >
> > > > Thank you again for your time and valuable feedback. We hope our further response has addressed your concerns and you can consider adjusting your rating. Please let us know if there are any unaddressed concerns and we will try to address them.

---

### Author Response · Authors · 2025-11-20
**General Response**

Dear Reviewers,

We sincerely thank you for your thoughtful reviews and constructive feedback. We appreciate the positive remarks highlighting our contributions, including the "**comprehensive testing suite and diverse model evaluation**" (Reviewer XCho), "**leading benchmark scale and broad scenario coverage**" (Reviewer 12fu, 73cU, 2ztj), "**holistic design spanning 8 tasks and 27 scenarios**" (Reviewer 12fu, 73cU, 2ztj), and the "**largest curated video dataset for emotional intelligence evaluation**" (Reviewer 12fu, 73cU, 2ztj).

We are grateful for the recognition of our "**creative multi-agent automated evaluation framework**" (Reviewer 12fu, 73cU, 2ztj), its "**high consistency with human experts**" (Reviewer 12fu, 73cU), and the "**novel assessment of both recognition and reasoning abilities**" (Reviewer XCho, 73cU, 2ztj). We also appreciate the comments noting our "**clear writing**, **transparent reporting**, and **rigorous large-scale empirical analysis of 20 MLLMs**" (Reviewer 12fu, 73cU, 2ztj). Several reviewers also emphasized the "**insights into model limitations and future research directions**" and the "**practical guidance for developing generalist and specialist models**" (Reviewer 12fu, 73cU, 2ztj).

We have carefully addressed all questions and concerns in our point-by-point response. In addition, we have also made the following major revisions in the updated manuscript (**all updated text is highlighted in blue**):

*  **Human-level vs. MLLM-level Evaluation** (`Appendix E.1`): We added a new human-level and MLLM-level benchmark study, clarifying the remaining emotional intelligence gap between humans and MLLMs.
*  **Multi-run Stability of Multi-Agent Evaluation** (`Appendix E.2`)：We evaluated the stability of our multi-agent framework across multiple runs.
* **Judge Model Replacement Study** (`Appendix E.3`)：We conducted an extensive judge-replacement experiment, comparing various core judge models.
* **Modality Ablation: Audio vs. Video vs. Audio+Video** (`Appendix E.4`)：We added ablation experiments under audio-only, video-only, and full audio+video settings.
* **Error Pattern Analysis** (`Appendix E.5`): We added a new taxonomy of model errors and quantified failure patterns across models.
* **Expanded Human–GPT Agreement Study** (`Appendix E.6`): We expanded the sample size and re-evaluated the agreement between human and GPT evaluation results.
* **CoT Instruction Ablation Study** (`Appendix E.7`): We added an ablation study on model performance with vs. without CoT prompts.
* **Audio-Enabled Gemini-2.5-Pro Evaluation** (`Appendix E.8`): We added experiments evaluating Gemini-2.5-Pro with audio preserved.



Please feel free to let us know if further clarification is needed.

Best Regards,
The Authors

---

### Author Response · Authors · 2025-12-01
**Summary of Rebuttal**

Dear AC, SAC, and PC,

Thank you for your support throughout the review process. We sincerely appreciate the time and effort you have invested in facilitating this discussion. We are pleased to report that we have successfully addressed all reviewer concerns. Below is a brief summary of our discussions with each reviewer:

1. **Reviewers 12fu and 73cU:**
    Both reviewers provided **highly positive evaluations (Rating: 8 and 6)**. Through detailed clarifications and additional experiments, we fully resolved all questions and weaknesses they raised.

2. **Reviewer XCho:**
    Reviewer XCho initially listed 4 weaknesses. During the first round of discussion, **he/she explicitly acknowledged that 2 of them were fully addressed**. In the second round, we introduced three supplementary experiments, which effectively resolved the remaining 2 concerns.

3. **Reviewer 2ztj:**
    Reviewer 2ztj initially made several factual errors, such as:
    - Misinterpreting our leaderboard score as a simple average across settings instead of an overall evaluation across all samples.
    - Incorrectly claiming that our benchmark focuses only on emotion classification, despite the benchmark’s explicit evaluation of emotion reasoning.

    After our clarifications, he/she acknowledged these factual errors and **increased his rating**. He/She then requested two additional experiments on dynamic affective analysis and audio ablation. In our most recent response, we incorporated both experiments and fully addressed all of his/her concerns.

**All feedback from the reviewers, including confirmations that their concerns were resolved and the rating increase, was provided before the large-scale reviewer identity leak on November 27.**

We hope this summary assists you in your subsequent decision-making process. Thank you again for your time and consideration.

Sincerely,
The Authors of Manuscript 1526

---

### Meta-Review · Area_Chair_LYeW · 2026-01-08

**Summary:**

This paper introduces MME-Emotion, a large-scale benchmark for holistically evaluating emotional intelligence in MLLMs, featuring over 6,000 videos across 8 tasks and 27 scenarios, and an automated multi-agent evaluation framework. My decision is to accept. The benchmark addresses a clear gap in assessing both emotion recognition and reasoning. While reviewers raised valid concerns about task definition, modality contribution, and evaluation scope, the authors provided extensive rebuttals and revisions. They clarified task structures, added crucial experiments (human baselines, modality ablations, error analysis), and demonstrated the benchmark's reliability and discriminative power. The work offers substantial empirical value through its rigorous evaluation of 20 MLLMs, yielding insights into current limitations and pathways for improvement. It provides a solid foundation for future research in affective multimodal AI.

**Reviewer Concerns:**

The rebuttal comprehensively addressed most key concerns. It clarified the hierarchical nature of the eight tasks as "settings" rather than strictly parallel tasks, mitigating validity concerns (2ztj). It added human-level benchmarks (XCho), systematic audio-video ablations for multiple models (73cU, 2ztj), and stability/reliability tests for the multi-agent evaluator (12fu, 73cU). It also explained the low zero-shot performance of specialist models like AffectGPT as expected out-of-distribution behavior (2ztj). A remaining outstanding concern is the limited analysis of cultural and demographic diversity in emotional expression, acknowledged as a future direction due to annotation constraints (73cU). Additionally, the benchmark's focus on recognition/reasoning over emotion generation is a stated scope limitation (12fu).

**Reviewer Scores:**

With full discussion, XCho (4) would likely raise to a 6, as the added human baseline and text-modality comparison (EmoBench) directly addressed core concerns about isolating emotion understanding. 12fu (8) would maintain a 8; their positive assessment was reinforced by the added stability and judge-model analysis. 73cU (6) would likely increase to a 7 or 8. The detailed responses and new experiments (modality ablation, error taxonomy, extended human-GPT agreement) satisfactorily addressed nearly all their technical questions. 2ztj (2→4) already increased their score during discussion. With the additional dynamic analysis and full audio ablation study provided in the final rebuttal, they might have further raised to a 5 or 6, as their primary concerns about task definition, metric validity, and modality contribution were resolved.

---

### Decision · Program_Chairs · 2026-01-26

Accept (Poster)